# When the hammer drops: Identification of knapping techniques in blade production based on a multi-scale study of knapping traces

Olivier Touzé[1,2]*, Veerle Rots[1,3]

1 TraceoLab, Research Unit AAP, University of Liège, Liège, Belgium, 2 UMR 8068 TEMPS, Nanterre, France, 3 Fonds de la Recherche scientifique-FNRS, Brussels, Belgium

* otouze@uliege.be

## Abstract

As defined by J. Tixier, a knapping technique corresponds to the concrete means used to detach a flake. It involves three essential parameters: the tool(s) used, the mode of force application and the behaviour of the body which includes the knapping gesture. In order to identify the knapping techniques used in prehistory, previous studies have mainly focused on macroscopic features on the blanks, but difficulties have often been encountered, leading to mixed results. We present the results of an experimental study that incorporates the macroscopic and microscopic level to examine and characterize knapping traces and integrates a hierarchical cluster analysis to refine identifications. Microscopic traces prove to be complementary to macroscopic traces and to constitute a key aspect for the identification of prehistoric knapping techniques. By focusing on the mode of force application and the contact tool, we show that each parameter of the knapping technique needs to be identified separately. Based on this principle, we demonstrate that it is possible, on the basis of specific sets of attributes, to identify blades produced by direct and indirect percussion and pressure, as well as to differentiate between the use of harder and softer contact tools, although further characterization of the latter does not seem possible without the identification and analysis of knapping-related residues.

## 1. Introduction

According to J. Tixier's definition, knapping methods correspond to the approaches followed by knappers to achieve a desired goal. It refers to the organisation of removals extracted from the raw material processed. Knapping *techniques*, on the other hand, are the concrete means by which a method – and therefore the detachment of flakes – is carried out [1] (see also [2]: 60, [3]: 30]).

**Data availability statement:** All relevant data are within the manuscript and its Supporting information files.

**Funding:** This research was performed within the framework of a postdoctoral fellowship funded by the Fonds de la Recherche scientifique – FNRS (https://www.frs-fnrs.be/en/) attributed to OT. We also acknowledge support from the European Research Council (ERC Grant Agreement no. 312283 to VR), Fonds de la Recherche scientifique-FNRS (PDR to VR) and the University of Liège in terms of the equipment used for this study. VR acknowledges financial support from the Fonds de la recherche scientifique-FNRS (DR). The funders had no role in study design, data collection and analysis, decision to publish, or preparation of the manuscript.

**Competing interests:** The authors have declared that no competing interests exist.

According to Tixier, a knapping technique involves the combination of three parameters:

- the tool(s) used (hammer, punch, crutch, anvil, abrader, etc.) and its(their) characteristics (raw material, morphology, dimensions, mass, hardness, etc.);

- the mode of force application (MFA in the rest of the text): direct percussion, indirect percussion or pressure;

- all aspects related to the use of the body – what we will refer here as the "body behaviour" – which includes the body position during knapping, the way of holding the core or the flake that is retouched (when no device is used) and the knapping gesture (kinematic, kinetic energy, angle of incidence of the contact tool), etc.

Based on an experimental reference library, it is possible to try to identify the nature of the prehistoric contact tool (= the tool that came in contact with the striking/pressure platform; CT in the rest of the text) and MFA used. Multiple archaeological examples of knapping tools are also documented, providing direct evidence on past knappers' toolkits [e.g., 4–17]. The body behaviour, however, largely requires direct observation and is therefore very difficult to grasp in prehistoric contexts.

Interest in the manufacturing process of prehistoric stone tools and the associated knapping techniques began in the second half of the 19th century and continued throughout the second half of the 20th century. This interest mainly came from modern knappers who were trying to replicate prehistoric and ethnographic artefacts using materials that were compatible with those available to their makers (e.g., [18–25]), as well as from other scholars [26,27] who began to quantify certain features on archaeological pieces, such as bulb height, exterior platform angle, and "bulb angle" (after [28]) which are still used today in technological studies. This pioneering research was conducted in parallel with studies of contemporary flintknapping, both local and non-local (e.g., [29]). Local flintknapping concerns above all the gun flint industry which, although in decline at the time due to the evolution of firearm technology, was still practiced by few craftsmen in England [30–32], France [33–35], or Albania [36]. Although gun flint production is obviously not related to prehistoric flintknapping in many respects, these studies document it in great detail and demonstrate the authors' interest in all aspects of the techniques (*sensu* Tixier) used by these craftsmen.

This dynamic then led to more systematic and comparative testing of a wide range of techniques (e.g., [37–40]). In the second half of the 20th century, the rise of the technological approach, driven in France by A. Leroi-Gourhan, J. Tixier and their teams, and by others elsewhere (e.g., [41–43]), considerably increased interest in prehistoric techniques, as illustrated by the November 1964 conference at Les Eyzies (France) where self-taught knappers F. Bordes, D. Crabtree and J. Tixier, as well as other scholars, met [44,45], and by an increase of the number of knapping experiments reported in the literature [46]. Although experiments with different techniques for accurately reproducing archaeological objects continued (e.g., [47–52]), from the 1970s onwards, knapping experiments gradually moved towards a more in-depth

understanding of stone-knapping. Different research directions therefore begun to be explored, such as fracture mechanics and flake formation (e.g., [53–77]), idiosyncratic knapping behaviours (e.g., [78–88]; see also [89–91] for strictly archaeological cases) and the related topics of skill levels and skill acquisition (e.g., [85,92–105]; see also [106–116] for archaeological and ethnographical cases).

Several researchers became also interested in trying to isolate features on archaeological lithic artefacts made from brittle, isotropic materials (chert/flint and obsidian) that would allow formal identification of the knapping techniques used to manufacture them. Multiple combinations of MFA (direct and indirect percussion, pressure) and CT types (e.g., quartz-ite, limestone, antler, copper) were tested in the context of bifacial flaking and blade production [117–121]. However, few attributes were taken into account during the analyses (e.g., dimensions and weight of the flakes, dimensions of the butt, presence/absence of butt crushing), and although certain patterns were identified, no causal relationship could be established with the techniques tested in most cases, which led the authors to stress the difficulty of identifying knapping techniques, as well as the need to work with large samples when pursuing this goal [121: 57].

K. Ohnuma and C. Bergman, who used blind tests to confirm the reliability of their results, were among the first to identify a set of features which, in their opinion, allow for a reliable distinction between hard stone direct percussion (clear percussion point and cone, pronounced conchoidal fracture marks on the bulb, absence of lip, pronounced bulb) and soft (i.e., organic) direct percussion (vague percussion point and cone, presence of lip, diffuse bulb), while also highlighting the difficulty of discriminating the latter and soft stone direct percussion [122]. However, some of the features they considered to be diagnostic seem questionable, in particular the presence/absence of a lip, since R. Bonnichsen had already demon-strated a few years earlier that there was no direct causal relation between this feature and the nature of the CT [59: 176], a fact that was subsequently confirmed by A. Pelcin [72].

In France, the extensive experimental research of J. Tixier and J. Pelegrin on flintknapping had a major impact by pro-posing qualitative analysis grids for identifying different variants of direct percussion (hard stone, soft stone and organic CT [123], see also [6,124] for some nuances concerning the discrimination of certain variants), indirect percussion [125] (see also [126: 10–15, 127]), and pressure flaking and its various modes [125,128,129] (see also [126: 10–15, [130,131]: 98–101]) in relation to blade production, as well as for several retouching techniques used for the preparation of backed pieces [132]. While this research was initially linked to specific archaeological case-studies, its success with French lithic technologists led to the transfer of its results to flint industries from various other contexts. It also created a framework for applications with materials other than flint, such as obsidian [133] and schist [134], for example (see also [135] for dolerite).

Recent research on bifacial shaping [136–138], blade production [139–142], and blank retouching [143–146] pursued tests of multiple morpho-metric attributes in an attempt to discriminate various MFA and types of CT used under "realistic" conditions (i.e., where a limited set of variables is controlled, generally excluding idiosyncratic knapping behaviours). The variable results obtained confirm the complexity of the task, even more so when coarser, or anisotropic raw materials such as quartz, are considered [147–149].

Beyond the variability of their protocols, almost all of these studies share a common focus on the macroscopic knap-ping features present on the experimental artefacts they produced. This is maybe the result of the analyses being generally carried out with the naked eye, while low-magnification instruments, and even more so high-magnification instruments, were usually not used with a few exceptions (e.g., [144]; see also [150] for a recent example of a high-magnification approach applied to the traces left by the retouching of backed pieces). This choice is not meaningless, since S.A. Semenov, in his seminal work available in English since the mid-1960s [151] (1970 edition consulted), already drew attention to the formation of microscopic traces during manufacturing processes. It seems that lithic technologists (we are referring here to scholars who primarily study the knapping methods and techniques used to transform raw material into functional tools) have largely left aside microscopic observation, which remained in the realm of functional studies' specialists. From this perspective, it appears logical that the latter are to be credited for initiating the analysis of

microscopic wear traces related to knapping techniques, even if, in comparison with the amount of literature produced by technologists on the theme of techniques, these studies remain understandably limited from a strictly quantitative point of view. They nevertheless highlight the research potential of the different types of microscopic features related to knapping, including striations, polishes and residues [152: 25–28, 153–162], which appear on the platforms of flakes, i.e., the areas where the physical interactions between the CT and the striking/pressure platforms occurred. Fracture wings (also known as "gull wings", see [61: 80–84]) are also worth mentioning, as their relationship with crack velocity makes them an interesting proxy for identifying specific combinations of MFA and CT on glassy materials like obsidian [163–166].

To our knowledge however, few attempts have been made to date to test a wide range of MFA and CT in order to characterise and compare the resulting microscopic knapping traces (but see [155,158]). Bearing in mind J. Tixier's reflection about pressure flaking and knapping techniques in general: "*Y a-t-il une clé pour reconnaître le débitage par pression? Non. Comme pour tout ce qui concerne les techniques de taille il y a une série de stigmates plus ou moins caractéristiques*" (Is there a key to recognising pressure flaking? No. As with everything concerning knapping techniques, there is a series of more or less characteristic features [128: 66]; our translation), we believe nonetheless that the wider the range of knapping features taken into account, the greater the chances of reducing the risk of false positives and accurately identifying prehistoric knapping techniques.

Considering the literature accumulated to date, which has highlighted the need to identify techniques, not on the basis of isolated traces, but on the basis of specific associations of several traces, this article has two aims:

1) To take the characterisation of microscopic knapping traces further, building on the earlier work of V. Rots;

2) To combine several scales of analysis (macroscopy and microscopy), to facilitate the identification of trace associations specific to different techniques, using a suitable statistical tool (cluster analysis).

## 2. Review of the knapping features used in the literature and their diagnostic value

### 2.1. Creation of a sample of studies

The high number of studies related to the identification of knapping techniques makes any attempt at synthesis difficult. To provide a general overview of the attributes that have been considered in recognising the use of these techniques in an archaeological context, we have chosen to examine a sample of 29 studies published between the 1940s and the present day (S1a Fig, S1 Table). The observations presented below do not claim to be exhaustive: we have only sought to identify the main trends associated with this research theme.

This sample was selected according to several criteria. Because of the focus of our research, we first gave priority to publications presenting original experimental results (n = 27), or possibly a synthesis of previously obtained results (n = 1; [130]). The experiments reported in these works are mainly (n = 26) "realistic" experiments, in the sense that they were carried out by experimenters knapping freely, without the body behaviour (gestures, position, holding of the core or the bifacial piece) being controlled, or at least strictly controlled. Controlled experiments, i.e., those in which the fracturing of the rock is not carried out by an experimenter, but with the aid of an ad hoc device enabling all the variables involved to be controlled, are rarer (n = 2; [59,72]). This difference between the number of "realistic" and "controlled" experiments is due less to the choices we made in selecting our sample than to the history of research on knapping techniques. Finally, in addition to the experimental research mentioned above, a study based on the analysis of archaeological collections was also included [167], mainly because it mentions a particular knapping trace (lateralized oblique crack) for which we have found no other mention (or at least no explicit mention).

In addition, we have selected studies that use flint/chert (25 cases) and/or obsidian or industrial glass (8 cases). Some of the works (n = 3) included in the sample also concern other types of rock in addition to the above materials, but often anecdotally. Because our research focuses on traces and attributes related to *débitage*, we have selected studies based

on the production of blades (17 cases) or flakes (6 cases), or on the analysis of flakes detached during the shaping of bifacial pieces (10 cases), with some studies combining several of these productions. We did not, however, include studies focusing on the knapping traces formed during the retouching of tool blanks. Finally, we sought to include experimental studies documenting the three MFA, as well as a wide range of tools. The publications included in our sample primarily concern direct percussion (59 cases), of which the variants using antler (n = 18), hard stone (n = 14), and soft stone (n = 12) hammers are the most frequently studied (S1b Fig, S1 Table). Indirect percussion (n = 20) and pressure (n = 19) appear less frequently but are equally represented. The variants with an antler (n = 9) and a wood (n = 5) punch, and those with a crutch fitted with an antler (n = 6) or a copper (n = 4) CT are the best represented, respectively, for these two MFA. Occasionally, publications describe the traces associated with a specific MFA, without describing the nature of the tool used, but this remains a minority case (n = 5 for pressure; n = 1 for direct percussion).

## 2.2. Data processing

An initial examination of the publications selected enabled us to list a total of 185 attributes which, depending on the case, are considered useful or not useful for the recognition of knapping techniques (S2 Table). These attributes can be classified essentially according to the time of their formation in relation to the initiation of the fracture that causes the flake to detach, and according to the location or nature of the features observed (Fig 1). For example:

- The external platform angle (= the angle between the striking/pressure platform and the extraction surface) is an attribute that is defined <u>before</u> the flake is detached.

- The formation of a circular crack on the flake's butt (contact surface) or a pronounced bulb (ventral surface) is directly <u>associated</u> with the detachment.

- "Spontaneous retouch" [168], which would be more accurately described as "spontaneous scars" ("*enlèvements spontanés*" in French [169: 74]) since its formation is not intentional unlike proper retouch, occurs when the conditions under which the core is held cause the flake to come into contact with it, in the fraction of a second <u>after</u> detachment.

Because of the considerable number of attributes identified, and because many of them turned out to be quite similar, we have grouped together attributes of the same order (for example, those relating to the length of the flakes, or those referring to the dimensions of the platform). The correspondence between the initial attributes and the simplified attributes is shown in S2 Table. This grouping enabled us to reduce the number of attributes considered to 82 (S3 Table).

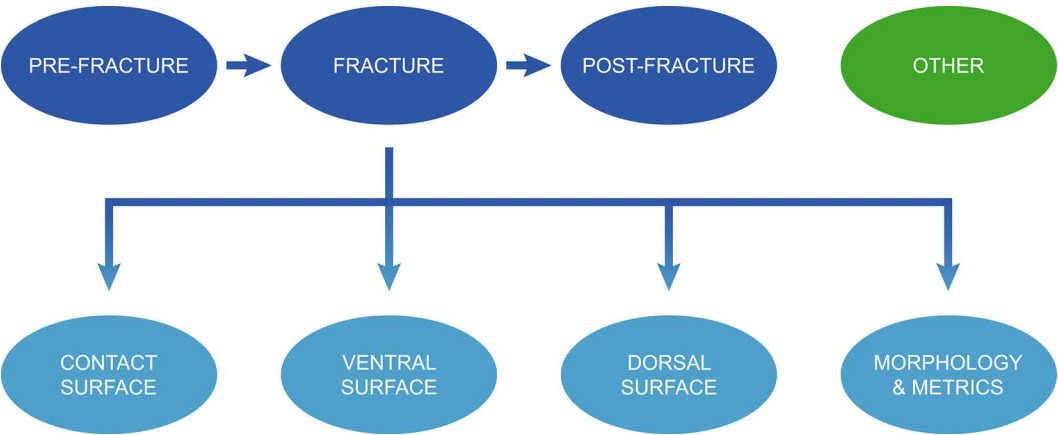

**Fig 1. Classification of attributes in the studies examined.**

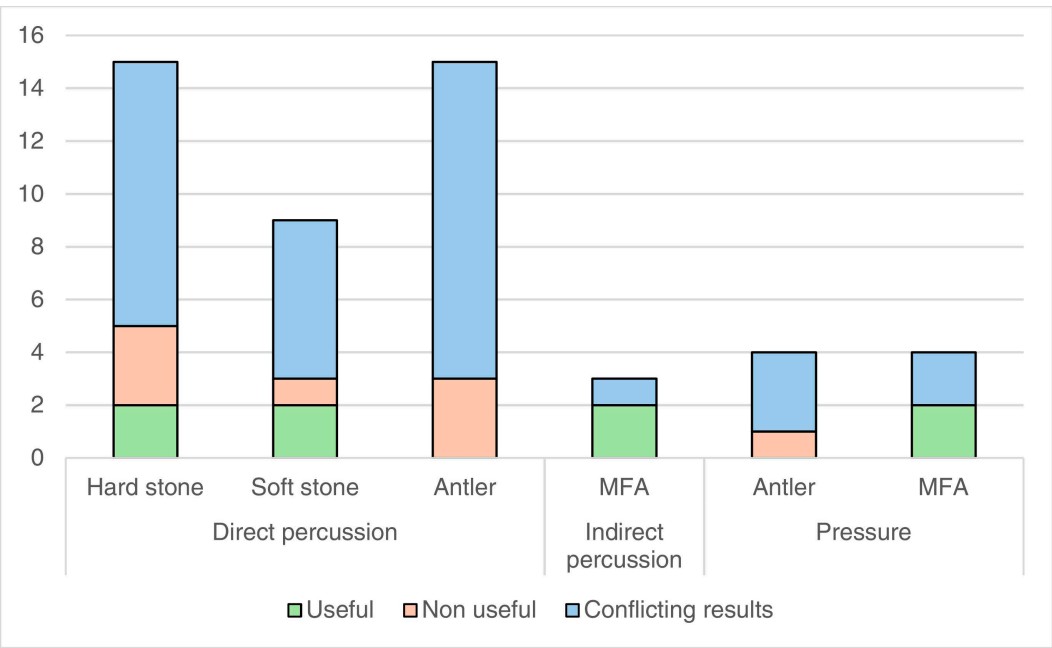

**Fig 2. Synthesis of the usefulness of attributes for the identification of knapping techniques based on the studies examined.**

Most of these 82 attributes appear in only one (n = 44) or two (n = 15) studies, but more than a quarter of them (n = 23) appear in at least three publications, including four in 10 publications or more: flake thickness (10 publications for a total of 20 recorded uses), flake morphology (11 publications for 22 uses), lip (15 publications for 37 uses) and bulb size (17 publications for 37 uses; S1c, S1d Fig, S2 Table). Only the eraillure scar was used more frequently than some of these, despite a slightly lower number of publications (9 publications for 23 uses; S3 Table).

In S3 Table, we have used a colour code to highlight the convergences and divergences observed in the estimated usefulness of each attribute for the identification of the different techniques considered. Convergences were defined on the basis of a minimum of three studies presenting converging results, which are not contradicted by other work. Finally, S4 Table simplifies the visualisation of the results a little further, by offering a count of the cases of convergence and divergence observed for the attributes that appear in at least three different publications. Because of this simplification S4 Table does not refer to the publications used, but this information is available in S3 Table.

### 2.3. Results

The most numerous and therefore instructive results concern direct percussion in its hard mineral (15 results), soft mineral (9 results) and antler (15 results) variants. For pressure knapping, the results only concern the antler variant (4 results) and the MFA in general (4 results). The three results available for indirect percussion only concern the MFA. While it is likely that the inclusion of a larger number of studies could compensate for these imbalances, several trends can be identified.

Of the attributes used in the selected studies, only six are unanimously considered useful (Fig 2), across all techniques. This represents a quarter of the 23 simplified attributes appearing in at least three studies, and around 7% of all 82 simplified attributes. Besides this small number, the usefulness of these "useful" attributes needs to be qualified. In the case of direct percussion, the formation of a point of impact and a cone of percussion, and the presence of a circular crack are

considered diagnostic of the hard and soft stone variants, so it can be estimated that these attributes do not really make it possible to distinguish between the two.

The usefulness of the external platform angle in identifying indirect percussion deserves to be considered even more cautiously, because the values proposed differ between authors (e.g., between 80 and 95° according to [125]; between 80 and 90° according to [127]; between 75 and 85° if the punch is organic according to [161]), and especially because these values overlap with those defined for other techniques (e.g., 60–90° and 70–85° for direct hard mineral and soft mineral percussion according to [123]; angle below 90° for pressure debitage according to [125]). Similarly, the values defined for the thickness of the platform of products obtained by indirect percussion (3–4 mm for blades approximately 20 cm long according to [125]; 3.3 mm on average according to [127]), turn out to be close to those proposed for other techniques (a few millimetres for direct organic percussion according to [123]; 3–4 mm or less for pressure according to [125]).

The morphology and thickness of the products obtained by pressure, which are the two most effective attributes for identifying the use of this MFA, can also be discussed. The studies we examined often state that pressure knapping, particularly when used to knap blades, produces thin products with regular morphology and thickness. However, a few authors have observed that indirect percussion can also produce regular blades when they are no longer than around 20 cm [125], or if the *débitage* is conducted using an antler punch [141]. Similarly, products with a regular thickness can be obtained by direct organic percussion [123], indirect organic percussion [161], or indirect percussion when the blades do not exceed 20 cm [125]. In addition, direct wood percussion and indirect percussion can produce thin blades [38].

There are 5 attributes unanimously estimated to be unhelpful for the identification of knapping techniques, including the mass and general dimensions of the products (direct percussion with hard stone and antler), the eraillure scar (direct percussion with soft stone and pressure with antler), the width of the butt (direct percussion with antler) and the width of the products (direct percussion with hard stone).

In most cases, however, the literature reviewed reveals contradictions in the assessment of the usefulness of attributes. In fact, 14 simplified attributes (platform size; platform thickness; crushed platform; point of impact; cracks; lip; bulb; bulb size; eraillure scar; morphology, length, width, thickness and curvature of the products) out of the 23 that we defined and retained for this analysis present such contradictory evaluations in 34 cases, 28 of which concern direct percussion, 5 pressure and 1 indirect percussion (Fig 2; S4 Table).

## 2.4. Synthesis

Despite its non-exhaustive nature, some conclusions can be drawn from this review of the literature on the identification of knapping techniques.

Firstly, and as has been noted on multiple occasions, while some attributes are more useful than others in identifying certain techniques, none of them is truly diagnostic. In other words, there is no attribute whose observation makes it possible to identify with certainty the use of a specific combination of MFA and CT, without any risk of confusion with other combinations.

Secondly, because the same causes produce the same effects, the high number of contradictions that we have noted in the literature must be caused by the influence of variables related to the design of the protocols or the execution of the experiments that have not been taken into account, or only to a limited extent. Because most of these experiments were carried out under "real" conditions, i.e., without it being possible to control certain aspects of the body behaviour, it seems credible to us that inter-individual behavioural variations constitute a large part of the explanation. Differences, however subtle, in the nature of the materials used (e.g., more or less homogeneous varieties of flint, or with fine or coarser grain; soft stone hammers of varying hardness, etc.) may also have played a significant role.

Thirdly, the correlation established between the use of direct mineral percussion (hard or soft) and, on the one hand, the formation of a point of impact and a percussion cone and, on the other hand, that of a circular crack, is instructive. Generally speaking, stone is in fact the hardest of the materials used as knapping tools in the literature examined, with the

exception of steel used in direct percussion in a single study [72]. This suggests that the hardness differential between the knapping tool that transmits the energy, and the material that fractures as a result of the mechanical stress thus created, is a key aspect in the recognition of techniques. We will come back to this point later.

## 3. Materials and methods

### 3.1. Creation of the experimental reference collection

**3.1.1. Protocol of the experiment.** Three MFA, each associated with different CT were included in the experiment: direct percussion, indirect percussion, and pressure with an abdominal crutch (see Table 1).

A single experienced knapper produced the entire experimental collection analysed in this study, using fine-grained flint collected in a quarry at Harmignies (Province of Hainaut, Belgium). Aged 57 at the time of the experiment, he had over 10 years' experience in blade production and had practised this activity between one and two days a week, on average, during the year preceding the experiment. He also has experience with all the techniques included in the study but has more training in direct percussion and pressure with an abdominal crutch (experience of over 10 years in both cases), than in indirect percussion (experience of between 5 and 10 years). According to his own perception, he feels more comfortable with the first two techniques. He is right-handed and learnt flintknapping by practising with a more experienced knapper.

For each combination of CT and MFA, a minimum of 11 blades had to be produced. These blades should preferably be complete, and it was imperative that their platforms be preserved. In addition, for this experiment, the blades were defined as elongated flakes:

• the length of which is equal to or greater than twice their width;

• at least 6 cm long (no upper limit has been defined);

• obtained during the laminar reduction of a core, i.e., a reduction process whose primary aim is to produce blades.

The knapper has received no instructions on how to proceed with the reduction process, other than to:

• try to extract the longest blades possible according to the characteristics of the core and his personal abilities;

• avoid using a specific reduction process associated with a particular archaeological culture;

• use a maximum of two hammers to prepare and maintain the core, in addition to the CT used to extract the blades;

• use (and maintain) flat striking/pressure platforms.

**Table 1. Combinations of MFAs and CTs used with the number of blades produced with each.**

| MFAs | CTs categories | CTs materials | N blades |
|---|---|---|---|
| Direct percussion | Hard stone | Quartzite | 11 |
| | Soft stone | Sandstone | 11 |
| | Organic | Antler (*Cervus elaphus*) | 10 |
| | | Bone (*Bos taurus*) | 13 |
| | | Wood (*Buxus*) | 11 |
| Indirect percussion | Organic | Antler (*Cervus elaphus*) | 11 |
| | | Bone (*Bos taurus*) | 4 |
| | | Wood (*Buxus*) | 11 |
| Pressure (abdominal crutch) | Organic | Antler (*Cervus elaphus*) | 11 |
| | | Wood (*Buxus*) | 14 |
| | Metallic | Copper | 13 |
| **Total** | | | **120** |

The reason for the last instruction is that flat platforms can record different types of traces during contact with the CT, whereas platforms with protruding areas (e.g., dihedral or faceted platforms) are more likely to record only the crushing of the latter [158], which is not as informative about the technique used.

During the experiment, the knapper worked seated on a chair when using direct percussion or indirect percussion and standing when using pressure with an abdominal crutch. In the latter case, an ad hoc wooden device was used to hold the core in place. Each blade detachment was documented with a Photron FASTCAM NOVA S12 camera facing the knapper at a 90° angle to the trajectory of his knapping gestures, at a speed of 16000 frames per second. The number of percussions or pressures applied to the core to detach a blade was not recorded, and in cases where several attempts were made, only the video recording of the one that succeeded in detaching the blade was kept.

The knapper only worked with tools selected from his personal kit. All the CTs used had a spherical or pointed active part. For indirect percussion, the knapper used a boxwood hammer with a flat active part together with the punches, and a hazelwood abdominal crutch for pressure knapping designed to allow the insertion of a dynamometer load cell. The characteristics of the CTs used are shown in Table 2. The mass of the crutch indicated in this table takes into account the mass of the load cell. Finally, a total of six quartzite and sandstone hammers (length between 59 and 135 mm, width between 17 and 105 mm, thickness between 12 and 60 mm, mass between 149 and 1176 g) were used to shape and maintain the cores during the experiment, but as mentioned above, only two were used for each *chaîne opératoire*.

**3.1.2. Composition of the experimental collection.** For each combination of CT and mode of force application, a core was exploited to produce at least 11 blades meeting the above-mentioned criteria. If this number could not be reached, another core was exploited using the same knapping tools. At the end of a reduction sequence, each blade was placed in a separate minigrip bag to avoid friction against other flint pieces prior to analysis. Its order of extraction during the reduction process was also noted on the bag. The core, as well as flakes larger than 3 cm with preserved, non-cortical butts, were also placed in separate minigrip bags for later analysis. All other flakes and debris were kept together in a single bag.

During the analysis of the experimental material, two blades, one extracted by direct percussion using an antler hammer and one extracted by pressure with a copper CT, had to be set aside because their butt were missing, which was not noticed during the experiment. As these pieces were therefore unusable for the analysis, only 10 blades extracted with

**Table 2. Characteristics of the tools used.**

| MFAs | Tool types | CT | Material | Length (mm) | Width (mm) | Thickness (mm) | Mass (g) |
|---|---|---|---|---|---|---|---|
| Direct percussion | Hammer | Yes | Quartzite | 83 | 73 | 47 | 374 |
| | Hammer | Yes | Sandstone | 120 | 59 | 38 | 406 |
| | Hammer | Yes | Antler | 275 | 59 | 37 | 582 |
| | Hammer | Yes | Bone | 340 | 70 | 42 | 413 |
| | Hammer | Yes | Boxwood | 280 | 30 | 52 | 421 |
| Indirect percussion | Punch | Yes | Antler | 155 | 39 | 36 | 163 |
| | Punch | Yes | Bone nº1 | 106 | 28 | 15 | 48 |
| | Punch | Yes | Bone nº2 | 84 | 30 | 21 | 50 |
| | Punch | Yes | Boxwood | 190 | 42 | 47 | 224 |
| | Hammer | No | Boxwood | 300 | 48 | 50 | 660 |
| Pressure | Crutch indenter | Yes | Antler | 98 | 14 | 13 | 35 |
| | Crutch indenter | Yes | Boxwood | 93 | 15 | 14 | 16 |
| | Crutch indenter | Yes | Copper | 92 | 5 | 13 | 53 |
| | Crutch (+ load cell) | No | Hazelwood | 710 | 260 | 35 | 2200 |

the antler hammer and 13 extracted with the copper CT were studied. In addition, the combination of indirect percussion and bone CT proved ineffective: two different bone CT (*Bos taurus*) were used successively, but the first cracked after extraction of the third removal, while the second was damaged after its first removal to the point where it could no longer be used. Furthermore, each of these four removals in total did not meet the 6 cm length threshold set for the experiment. We conclude from this result that the physical properties of bone do not allow it to withstand the mechanical stress associated with indirect percussion, at least as far as the production of blades is concerned. The four removals were analysed, but due to the failure of the technique and their small number, they were removed from the subsequent statistical analysis. Apart from the above, all the other techniques are documented by a minimum of 11 blades, up to 14 blades. A total of 120 blades were analysed (Table 1) and 116 were included in the statistical analysis (120 minus the 4 removals produced by indirect percussion with a bone punch). These blades are generally complete (78.4%) and, when fragmented, all fragments have usually been collected. However, in a few cases (n = 5; 4.3%), we were only able to collect part of the fragmented blades, usually their proximo-mesial part (n = 4), and in one case, their proximal part.

The collection is incorporated in the TRAIL experimental reference library for wear traces and residues housed in the TraceoLab at the University of Liège and can be consulted on request.

### 3.2. Analysis

**3.2.1. Visual analysis of the experimental collection.** The blades were analysed in three stages. They were first observed with the naked eye to describe their macroscopic characteristics. Next, each blade was examined using a Zeiss Discovery V12 stereomicroscope (magnification up to 120x) and a Zeiss Vario Scope.A1 microscope (magnification up to 500x), both before and after cleaning the knapping residues. The cleaning procedures were adapted according to the nature of the residues present on the platforms and usually involved the use of chemicals (3% hydrochloric acid [HCl] solution) to remove organic residues. As the latter sometimes exhibited very pronounced adhesion, certain pieces had to be subjected to more prolonged and intensive cleaning, sometimes involving the use of a 10% HCl solution. In some cases, certain residues were analysed in more detail with a JEOL IT300 SEM-EDS (EDS detector JEOL ex-230) by Dries Cnuts. Images and elemental spectra of the residues were acquired *in situ* on the tool surface in low vacuum (LV) mode (100 Pa) using the backscattered electron detector (BED) at 20.0kV with a probe current (PC) of 60.0.

Photos were acquired using a Hirox HRX-01 microscope equipped with a Zoom Revolver MXB-2500REZ lens and processed using the associated Hirox software, and a Zeiss V16 Axio Zoom microscope. The colorimetry of some photos has been modified using the DStretch plugin (version 8.41) of the ImageJ software to help the reader locate certain knapping traces, in which case the source photo is also provided. CAD illustrations were created using Affinity Designer 2 software. Statistical analyses were carried out using Jamovi software (version 2.3.28).

Data from the analysis of the 120 blades were recorded in a Microsoft Excel database (S5 Table). The data were recorded using 201 attributes describing the conditions under which the blades were produced (name of the knapper, raw material, MFA, tools used, etc.; 20 attributes in total), the cleaning procedures used to remove the knapping residues (30 attributes), and finally the macro- and microscopic traces observed (151 attributes). Due to the small number of descriptions currently available, the different types of microscopic traces observed are described in detail (see section *4.1. The microscopic knapping traces*; S1 Text). Macroscopic traces, on the contrary, have already been extensively analysed and discussed in the literature (see section *2. Review of the knapping features used in the literature and their diagnostic value*), so another description would not provide any new information. Appropriate attributes associated with macroscopic and microscopic traces were used together in the hierarchical cluster analyses (see section *3.2.3.1. First stage: attributes selection*).

**3.2.2. Types of microscopic traces observed.** Several types of microscopic traces have been examined. By "microscopic", we refer to traces that most often require low or high magnification to be examined. The distinction between macroscopic and microscopic traces remains partly arbitrary, however, since the latter can sometimes be perceived with

the naked eye (e.g., knapping residues). For the purposes of this work, we have included the following traces into the microscopic category:

- **Residues:** exogenous material deposited on the blade when it is detached. These residues mainly include material deposited by the CT on the platform of the blade, but they can have another origin as well (e.g., mineral residue deposited on the ventral face of the blade following a counter-shock with the core).

- **Cracks:** initiated on the surface of the platform, ring cracks can be more or less complete, of varying dimensions, and located in different places on the platform. For the purposes of this work, they have been classified into several types according to these different criteria (Fig 3).

- **Polishes:** a modification of the microtopography of the flint that is still visible after chemical cleaning of the surfaces and which, when observed under a reflected-light microscope, results in a clear reflection of light. Although the possible causes of polish formation have been debated for a long time, we suspect that knapping-related polishes mainly result from an attritional levelling of the higher parts of the microtopography, which may itself be caused by the abrasion that probably occurs when the CT comes into contact with and rubs against the striking/pressure platform (e.g., [170,171]).

- Many of the knapping polishes observed in this work have a distinctly linear aspect that results from the frictional movement of the CT against the striking/pressure platform. Instead of using the term "striations" which seems to encompass linear traces of various nature [153], we chose to use the term "linear polishes" here to designate these specific traces, as they do not have the appearance of grooves and they do not seem to differ in nature from the non-linear polishes, only in morphology.

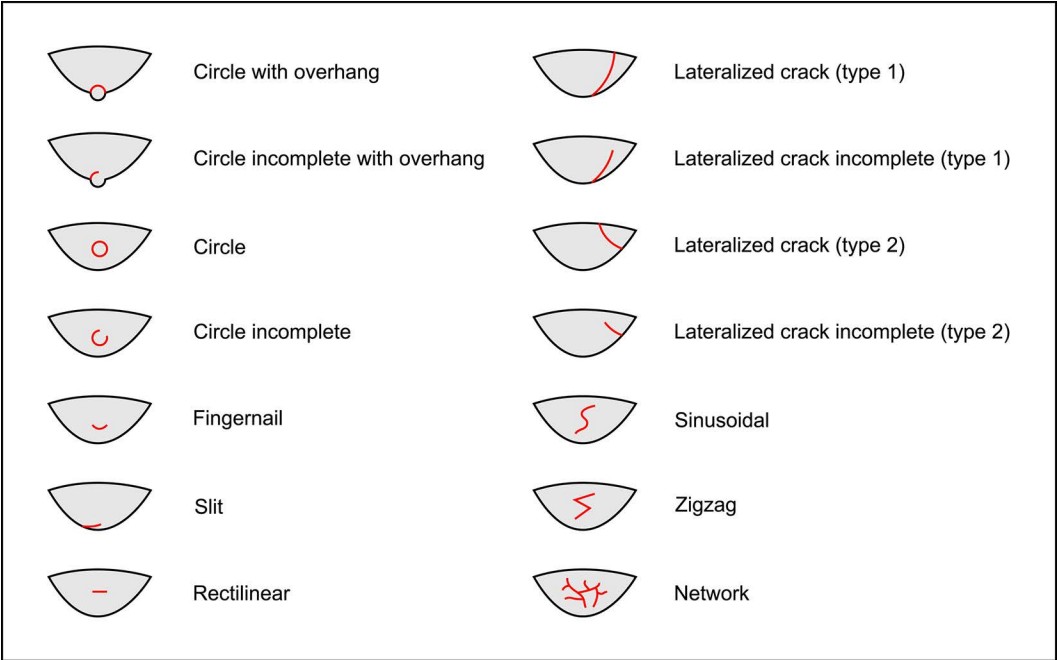

**Fig 3. Types of cracks.**

- **Incisions or grooves:** narrow linear traces, generally associated with linear polishes, but which, unlike the latter, are formed as a result of material being scratched off and therefore present a dark appearance when observed under a reflected-light microscope.

- **Scars:** negatives of small removals or chips that detach along the edges of the blade, because of the blade hitting external elements shortly after extraction (e.g., counter-shock against the core, shocks against the core's holding device if pressure knapping is used, contact with the ground when it falls, etc.).

3.2.3. **Hierarchical cluster analysis.** The hierarchical cluster analyses were carried out in several stages using Jamovi's snowCluster - Multivariate Analysis module (version 7.1.7) with the goal to determine whether knapping techniques can be identified from the recorded attributes, and, if so, to which extent. Hierarchical cluster analysis is an interesting tool for testing such an assessment, as it groups blades "blind" (i.e., without any knowledge of the techniques used) according to their degree of similarity defined based on the attributes they possess, or do not possess.

3.2.3.1. *First stage: attributes selection*: First, we chose to discard 84 attributes whose recording was relevant to the documentation of the experimental collection, but whose inclusion in the cluster analysis was not appropriate to the objectives pursued. These attributes include, for example, cracks and polishes resulting from abrasion of the edge of the striking/pressure platform *prior* to blade detachment, cracks associated with previous detachment attempts (type 2 lateralized cracks; Fig 3), attributes that do not relate to knapping traces *per se*, or attributes whose link with the MFA and CT was considered too weak or indirect to be useful for the identification of knapping techniques. The list of rejected attributes and the reasons for their rejection are detailed in S6 Table.

We therefore retained a total of 67 attributes out of the 151 used to describe knapping traces. We then tried to determine which of these 67 attributes could be useful in the following situations:

- Identification of the MFA, regardless of the CT used

- Identification of the CT, regardless of the MFA used

- Identification of the CT when the MFA is direct percussion

- Identification of the CT when the MFA is indirect percussion

- Identification of the CT when the MFA is pressure

The three last situations were chosen because of the composition of the experimental collection. Since each MFA was used with at least two CTs (in the case of indirect percussion, since the few laminar flakes obtained with bone were excluded from the cluster analysis) and at most five CTs (in the case of direct percussion), it was interesting to see how the cluster analysis would perform in CT identification when the MFA is already known. On the other hand, because four out of six CTs were used with only one MFA (quartzite, sandstone and bone with direct percussion; copper with pressure), testing situations where the CT is already known would have been of less interest, as it would also have led to the identification of the MFA in most cases. Furthermore, given the results presented below (see section 4.2.5. *Identification of the CT without prior identification of the MFA*), we believe that the order in which the MFA and the CT are identified is important and that the identification of the MFA should probably be prioritised.

To select the attributes used in each situation, Fisher's exact tests were performed for nominal attributes, and Kruskal-Wallis tests were performed for quantitative attributes (S6 Table). Attributes with a p-value ≤ 0.05 were selected. When Fisher's exact test failed to provide a result, we considered that the link between the attribute and the parameter (MFA or CT) under consideration could not be determined (the conditions were not met to apply a chi-square test instead), and we excluded the attribute from the hierarchical cluster analysis.

Finally, due to a bias in the formation of polishes on several blades knapped by direct percussion with antler (see section *4.1.1.3 Polishes*), the attributes associated with the description of polishes were excluded when these blades were included in the analysis. These are seven attributes for the identification of the MFA.

The hierarchical cluster analyses were performed with the following numbers of attributes:

- Identification of the MFA, regardless of the CT used: 22 attributes

- Identification of the CT, regardless of the MFA used: 21 attributes

- Identification of the CT when the MFA is direct percussion: 18 attributes

- Identification of the CT when the MFA is indirect percussion: 8 attributes

- Identification of the CT when the MFA is pressure: 41 attributes

The attributes used in each situation are indicated in S6 Table.

**3.2.3.2. *Second stage: attributes coding*:** As most of the 67 attributes we selected were initially recorded in nominal form, they were then transcribed in continuous form so that they could be used in the cluster analysis (S7 Table). The different possible states of a nominal attribute were coded using one-hot encoding, so that all data are coded '0' or '1'. In cases where an attribute has more than two states, columns were added so that each state could be coded using these numbers. Where possible, we also reduced the number of columns associated with a single attribute to simplify the database. For example, the attribute "Edge scarring – Location" has seven possible states (proximal, proximo-mesial, mesial, mesio-distal, distal, entire length, and NA if no edge scars are observed). We therefore coded this attribute using only three columns labelled 'proximal', 'mesial' and 'distal'. If a scar is present at one of these locations, the code used is '1', and if no scar is observed, the code used is '0'. In cases where scars are present or absent in several locations, '1' or '0' was entered in each of the appropriate columns. Thus, if scars are present, for example, along the entire length of a blade, '1' was coded in all three columns. In cases where an attribute has only two states, we kept a single column and simply coded each state using '0' and '1'. For example, the attribute "Edge scarring – Presence" has the possible states 'presence' and 'absence', so the former was coded '1' and the latter '0'.

Metric data, such as the width or thickness of a blade's platform, was normalised to reduce the risk of undue influence on the clustering results. To do this, we applied a min-max scaling so that all values are also between 0 and 1, while preserving the relative distribution of the data. Ordinal data, which concern only two attributes in the database, underwent the same procedure. For example, the attribute "Polishes – Density" has three ordinal states (low, intermediate, and high, as well as NA if no polish was observed), which were coded between 0 and 1 while respecting their order.

The results of the cluster analyses are presented in section *4.2. Identifying knapping techniques using hierarchical cluster analysis*.

## 4. Results

### 4.1. The microscopic knapping traces

#### 4.1.1. Direct percussion.

**4.1.1.1. *Residues*:** The five CTs used in direct percussion nearly always leave residues on the platforms of the blades. Only one blade detached with bone does not have any residue, while the presence of residues on a few blades detached with boxwood and antler could not be formally confirmed (Figs 4 and 5; Table 3). This observation confirms previous experimental results which had already shown that *débitage* and retouching are major causes of residue deposition [172]. Quartzite, which is the hardest CT used in the experiment, regularly leaves residues concentrated in a specific area of the butt (n = 4 cases out of 11; Fig 4a), but this phenomenon is less frequently observed with the other, softer CTs (only 2 cases observed in total with antler and bone). Moreover, these residues are mostly found in close spatial

association with cracks in the case of quartzite (n = 6/11). When the cracks are circular in morphology, quartzite residues are commonly located within these cracks exclusively, with little or no overlap with their contours (Fig 4a). With the other CTs, this association is partial most of the time, and only direct percussion with sandstone gives a few cases (n = 2/11) of associations comparable to what is observed with quartzite. Conversely, the platforms of three blades knapped by direct percussion with bone show a total absence of spatial links between the residues and the cracks.

The knapping residues generally take the form of deposits of variable shape, and it is uncommon that they have a clear linear morphology testifying to a brief friction of the CT against the striking platform (n = 3/11 with quartzite, 1/10 with antler, 1/13 with bone). This directional friction is however noticeable from the grooves or incisions that can often be observed within the residues, particularly when these are organic. These incisions are indeed more frequently observed with boxwood (n = 6/11) and antler (n = 4/10; Fig 5a) residues, but slightly less common with bone (n = 3/13), sandstone (n = 2/11) and quartzite (n = 1/11) residues. It is conceivable that these incisions are created by silica particles detached during impact and trapped between the CT and the surface of the striking platform. These particles would then be rubbed against the latter and would incise the residues. In the case of organic CTs, it is also possible that some silica particles could become stuck on their surface after several detachments and cause these incisions to form.

In this work, we did not attempt to quantify the degree of adhesion and resistance of the knapping residues. Nevertheless, we found that there were significant differences in these aspects depending on the CT used in direct percussion. The

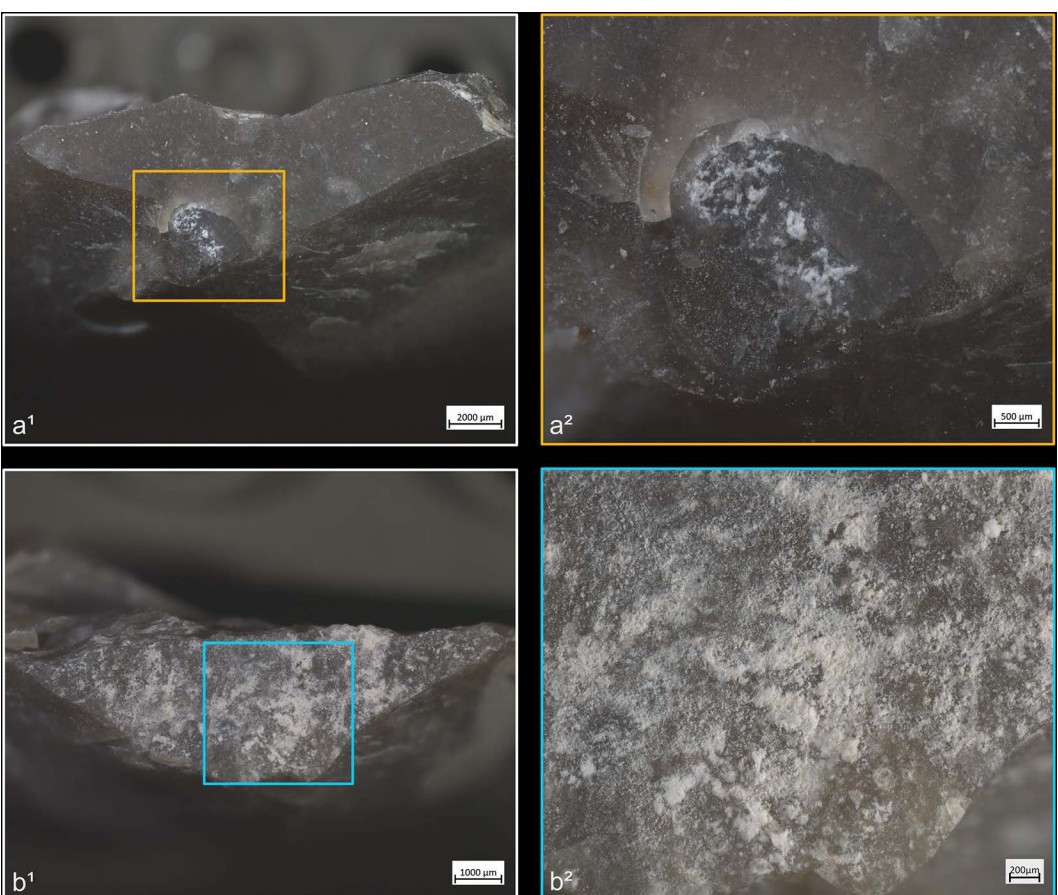

**Fig 4. Residues associated with the use of direct percussion and a mineral CT.** a: quartzite; b: sandstone.

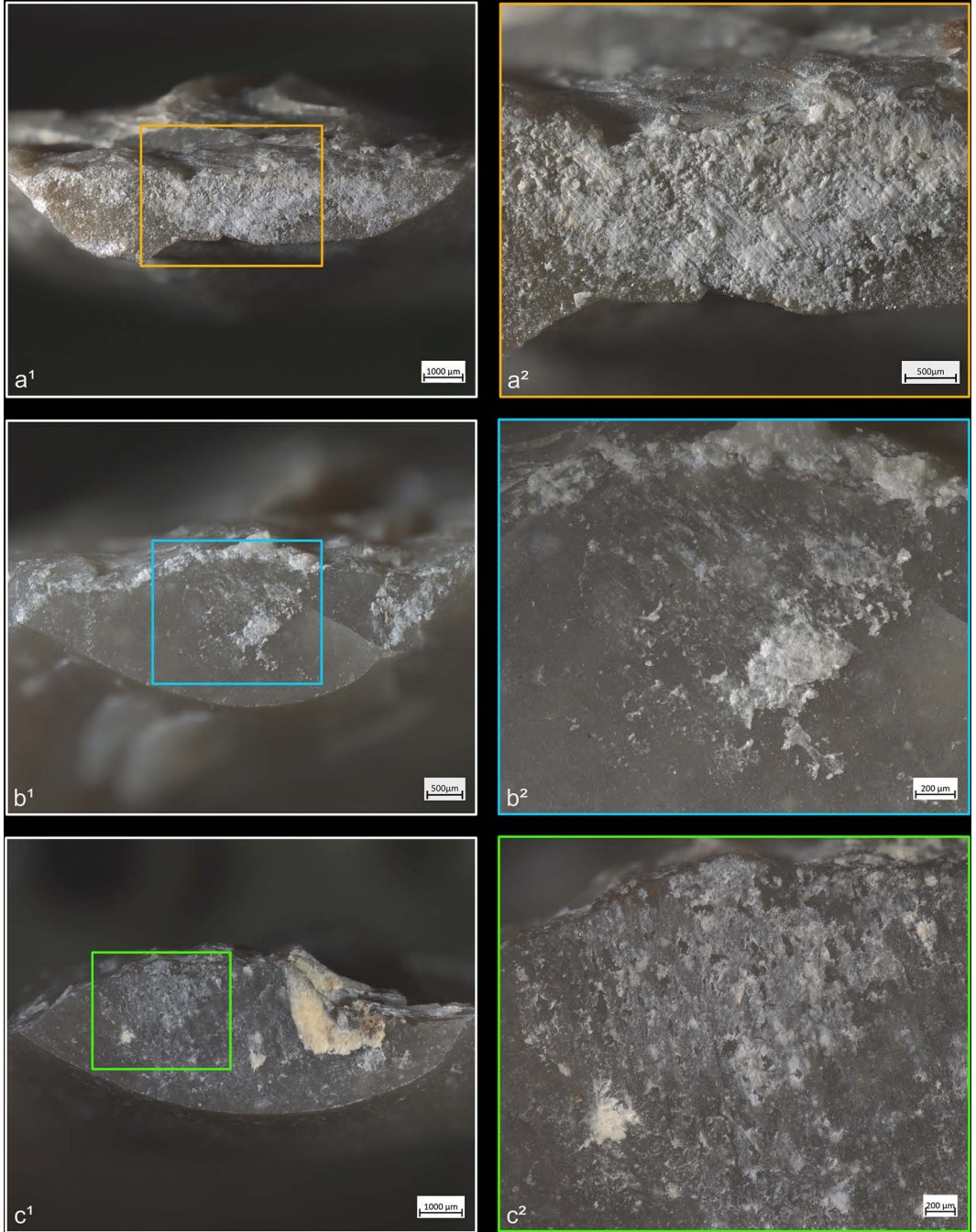

**Fig 5. Residues associated with the use of direct percussion and an organic CT.** a: antler; b: bone; c: boxwood.

cleaning of knapping residues related to the use of organic CTs showed that these residues can be very resistant. Antler residues required the most aggressive procedures to be removed, combining exposure to a 10% hydrochloric acid solution over periods of up to two hours in some cases, and the use of an ultrasonic bath. Previous observations suggested a broad variation in adhesion for hydroxyapatite-rich residues depending on the cause of deposition (knapping, use, hafting,

**Table 3. Count of the types of microscopic traces observed on the blades for each combination of MFA and CT used.**

| Location | Trace | Direct percussion | | | | | Indirect percussion | | | Pressure | | | Total |
|---|---|---|---|---|---|---|---|---|---|---|---|---|---|
| | | Antler (n=10) | Bone (n=13) | Box-wood (n=11) | Quartz-ite (n=11) | Sand-stone (n=11) | Antler (n=11) | Bone (n=4) | Box-wood (n=11) | Antler (n=11) | Box-wood (n=14) | Cop-per (n=13) | |
| Platform | Residue | 9 | 12 | 10 | 11 | 11 | 7 | 3 | 10 | 9 | 5 | 13 | **100** |
| Platform | Incision with residue | 4 | 3 | 6 | 1 | 2 | 7 | 2 | 7 | 1 | 2 | 1 | **36** |
| Platform | Polish | 5 | 12 | 10 | 10 | 9 | 11 | 4 | 9 | 10 | 3 | 10 | **93** |
| Platform | Incision with polish | 5 | 12 | 10 | 10 | 9 | 9 | 4 | 9 | 4 | 3 | 4 | **79** |
| Platform | Crack (all types) | 10 | 11 | 8 | 10 | 10 | 10 | 4 | 8 | 8 | 5 | 13 | **97** |
| Platform | Crack – circle w/ protrusion | 0 | 0 | 0 | 2 | 3 | 1 | 0 | 0 | 0 | 0 | 0 | **6** |
| Platform | Crack – circle w/ pro-trusion incomplete | 0 | 2 | 0 | 1 | 3 | 1 | 0 | 0 | 0 | 0 | 4 | **11** |
| Platform | Crack – circle | 2 | 1 | 0 | 5 | 2 | 1 | 1 | 0 | 2 | 0 | 0 | **14** |
| Platform | Crack – circle incomplete | 3 | 0 | 0 | 5 | 0 | 3 | 1 | 1 | 0 | 0 | 2 | **15** |
| Platform | Crack – fingernail | 4 | 1 | 0 | 1 | 0 | 3 | 2 | 4 | 0 | 0 | 7 | **22** |
| Platform | Crack – slit | 1 | 5 | 6 | 0 | 2 | 2 | 1 | 4 | 2 | 3 | 0 | **26** |
| Platform | Crack – lateralized type 1 | 1 | 6 | 3 | 0 | 1 | 2 | 1 | 2 | 1 | 0 | 1 | **18** |
| Platform | Crack – lateralized type 1 incomplete | 0 | 0 | 0 | 0 | 0 | 0 | 0 | 0 | 0 | 0 | 1 | **1** |
| Platform | Crack – rectilinear | 0 | 0 | 0 | 1 | 0 | 0 | 0 | 0 | 0 | 0 | 1 | **2** |
| Platform | Crack – sinusoidal | 1 | 0 | 0 | 0 | 0 | 1 | 0 | 0 | 1 | 0 | 1 | **4** |
| Platform | Crack – zigzag | 1 | 0 | 0 | 0 | 0 | 0 | 0 | 0 | 0 | 0 | 0 | **1** |
| Platform | Crack – network | 3 | 0 | 1 | 1 | 4 | 2 | 0 | 0 | 3 | 1 | 4 | **19** |
| Platform posterior edge | Residue strip | 3 | 8 | 10 | 5 | 5 | 11 | 4 | 11 | 11 | 14 | 13 | **95** |
| Platform posterior edge | Scar | 0 | 1 | 1 | 0 | 0 | 0 | 0 | 1 | 7 | 4 | 1 | **15** |
| Platform anterior edge | Polish | 7 | 13 | 11 | 11 | 10 | 11 | 4 | 11 | 11 | 14 | 4 | **107** |
| Platform anterior edge | Crack | 9 | 12 | 9 | 10 | 10 | 11 | 4 | 11 | 9 | 14 | 12 | **111** |
| Dorsal face | Residue | 0 | 2 | 1 | 0 | 0 | 1 | 0 | 1 | 11 | 14 | 13 | **43** |
| Ventral face | Residue | 0 | 1 | 0 | 0 | 0 | 3 | 0 | 4 | 1 | 0 | 1 | **10** |
| Edges | Scar | 8 | 9 | 8 | 8 | 5 | 11 | 3 | 11 | 11 | 14 | 13 | **101** |

etc., cf. [173]). Production-related residues were found not to adhere very strongly when using an ultrasonic bath, but only a small number of blanks were tested (n=3; [173]). Here, an ultrasonic bath was only used in a subsequent phase of cleaning due to the adhesion of production-related organic residues, which can indeed be important according to this larger dataset. This adhesion could be explained by the amount of kinetic energy involved in direct percussion which likely causes strong compression of the residues on the striking platform. As this compression increases, the adhesion of the residues also increases [173] which makes them less vulnerable to little or moderately aggressive cleaning procedures. Mineral residues proved much easier to remove by comparison, with simple exposure to the ultrasonic bath for around ten minutes often being sufficient. Although the adhesion of knapping residues should be more precisely studied and quantified in the future, it is nevertheless interesting to note their very pronounced adhesion in certain cases, particularly when

the CT is organic. Provided that taphonomic conditions are favourable, this suggests that such residues could often be preserved in archaeological contexts, which would facilitate the identification of the CTs used.

Blades knapped by direct percussion often (n = 31/56) show of a more or less continuous trace along part of the posterior edge of their platform, which can be likened to a polish due to its reflective nature under the microscope. This trace also appears with indirect percussion and pressure knapping (see sections *4.1.2.1.* and *4.1.3.1.*). The SEM-EDS analyses carried out by D. Cnuts (TraceoLab/ULiège) on a sample of blades knapped by indirect percussion with a boxwood CT (Fig 6) and pressure with a copper CT (Fig 7) have shown that this trace corresponds to a residue of organic matter, whose presence both on top and in-between the higher parts of the microtopography suggests that it has undergone very strong compression. This also explains why the cleaning procedures employed, although they generally reduced the density of these residues, never resulted in their complete disappearance. Because they also appear with pressure knapping with copper, we assume that they correspond primarily to organic material unintentionally deposited by the knapper himself on the active part of the CT before blade detachment. Nevertheless, the nature of the CT used probably also influences the formation of these residues, as the SEM-EDS analyses indicate a more marked presence of carbon in the case of indirect percussion with boxwood than in the case of pressure with copper (Figs 6 and 7).

**4.1.1.2. *Cracks*:** Cracks are extremely common with direct percussion (Fig 8). Boxwood is more often associated with platforms with no crack (n = 3/11) than other materials (n = 2/13 for bone, 1/11 for sandstone and quartzite, 0/10 for antler). Cracks may be single or multiple, and more or less concentrated or dispersed over the surface of the platform depending on the number of contact points, which is influenced by the microtopography of the CT and also by the possible repetition of detachment attempts made in the same area of the striking platform. In most cases, cracks are in contact with the posterior edge of the platform, but it is not uncommon, especially with quartzite (n = 4/11) and antler (n = 3/10), to observe cracks located both in contact with and further away from this edge. When cracks are present on the platforms of blades knapped with boxwood, they are always in contact with the posterior edge (Fig 8c). In addition, on three blades knapped with quartzite, some cracks are associated with small, crushed areas (Fig 8a). This phenomenon corresponds to the "frosted" contact points described by J. Pelegrin for direct percussion with hard stone [123]. Like this author, we did not observe it with other CTs.

From a morphological point of view (see Fig 3 for the different crack morphologies considered in this work), circular cracks are observed with rather hard or moderately hard CTs, especially mineral CTs, but never with boxwood. Circular cracks associated with a protrusion (*débord*) of the posterior edge of the platform appear both with sandstone (n = 3/11 for complete as well as incomplete circular cracks), and with quartzite (n = 2/11 for complete cracks and 1/11 for incomplete cracks), and sometimes with bone (no case of complete crack, but 2/13 for incomplete cracks). Circular cracks without protrusion are mainly observed with quartzite (n = 5/11 for complete as well as incomplete cracks), but they also form with sandstone (n = 2/11 for complete cracks, no case of incomplete crack), with antler (n = 2/10 for complete cracks and 3/10 for incomplete cracks), and with bone (n = 1/11 for complete cracks, no case of incomplete crack).

Conversely, non-circular cracks tend to be more often observed with moderately hard or rather soft CTs, whether organic or mineral (sandstone), and are only exceptionally documented with quartzite. Fingernail cracks are mainly observed with antler (n = 4/10; Fig 8b) and to a lesser extent with quartzite (n = 1/11) and bone (n = 1/13). Type 1 lateralised cracks are observed mainly with bone (n = 6/13) and boxwood (n = 3/11), while rare cases were found with sandstone (n = 1/11) and antler (n = 1/10). Slit cracks show a similar pattern: they are more present with boxwood (n = 6/11; Fig 8c) and bone (n = 5/13), but they also appear occasionally with sandstone (n = 2/11) and antler (n = 1/10). Finally, network cracks are mainly encountered with sandstone (n = 4/11) and antler (n = 3/10), and more rarely with quartzite (n = 1/11) and boxwood (n = 1/11). Network cracks are probably created because of repeated attempts at detachment in the same area of the striking platform, by CTs with irregular microtopography that creates multiple contact points, or by a combination of these two factors.

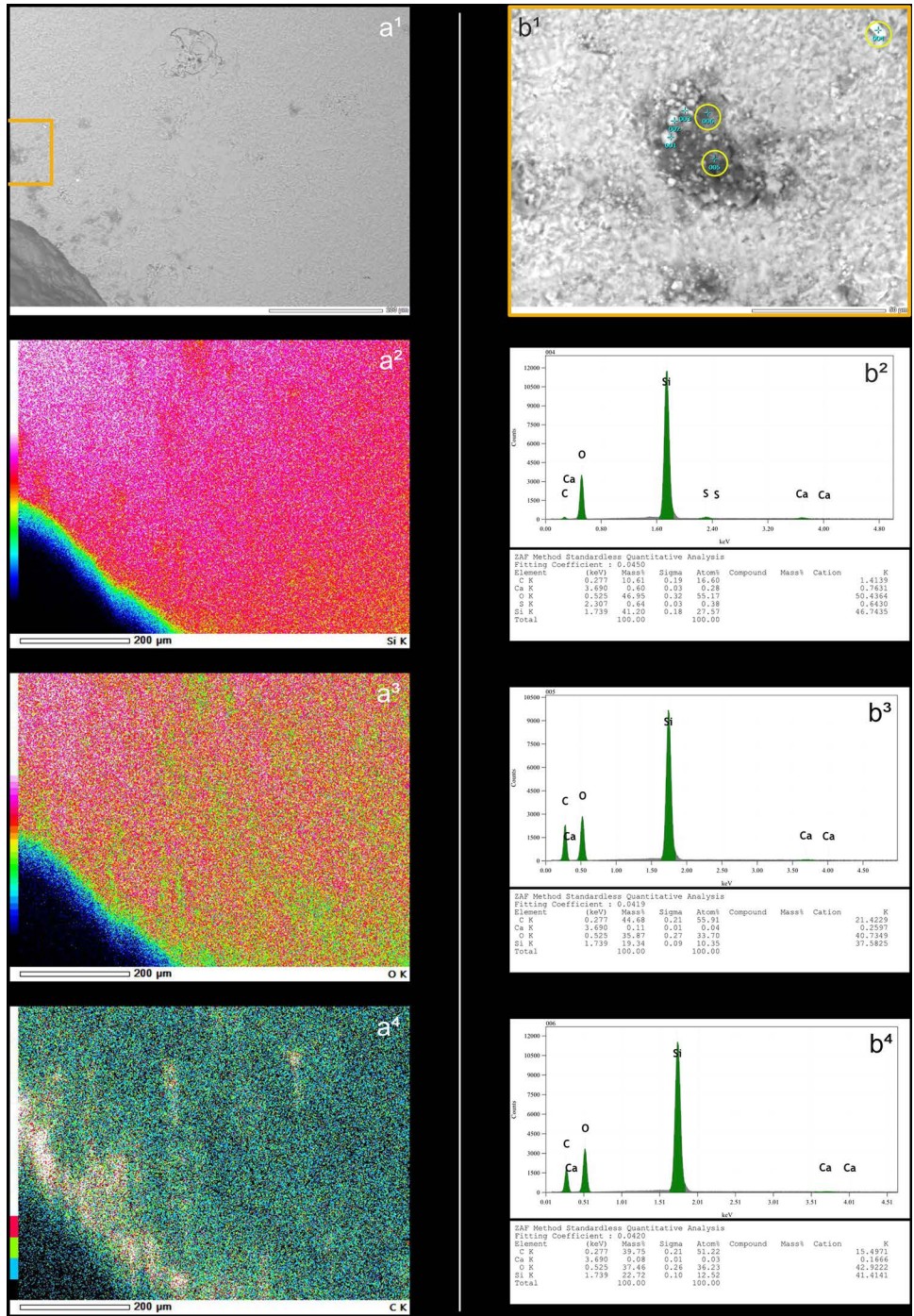

**Fig 6. Mapping of the elemental characterisation of a section of the platform of a blade knapped by indirect percussion with a boxwood CT (experimental piece Exp122-93).** On the left, SEM micrograph showing the analysed platform area, with part of its posterior edge visible in the lower-left corner (a[1]), and the elemental mapping of silica (Si; a[2]), oxygen (O; a[3]), and carbon (C; a[4]). Note that carbon is concentrated along the posterior edge of the platform and indicates the deposition of organic matter in this specific location. On the right, SEM micrograph (b[1]) of a specific area of image a[1], showing the locations (yellow circles) where the elemental spectra in b[2] (sample 004 shown in b[1]), b[3] (sample 005) and b[4] (sample 006) were obtained. Note that the spectra in b[3] and b[4] come from the dark area shown in b[1] and have high peaks for carbon (C), indicating that this area is rich in organic matter. In contrast, the spectrum in b[2] shows high peaks only for oxygen (O) and silica (Si) and therefore corresponds to the flint matrix (SEM analysis: D. Cnuts; CAD: O. Touzé).

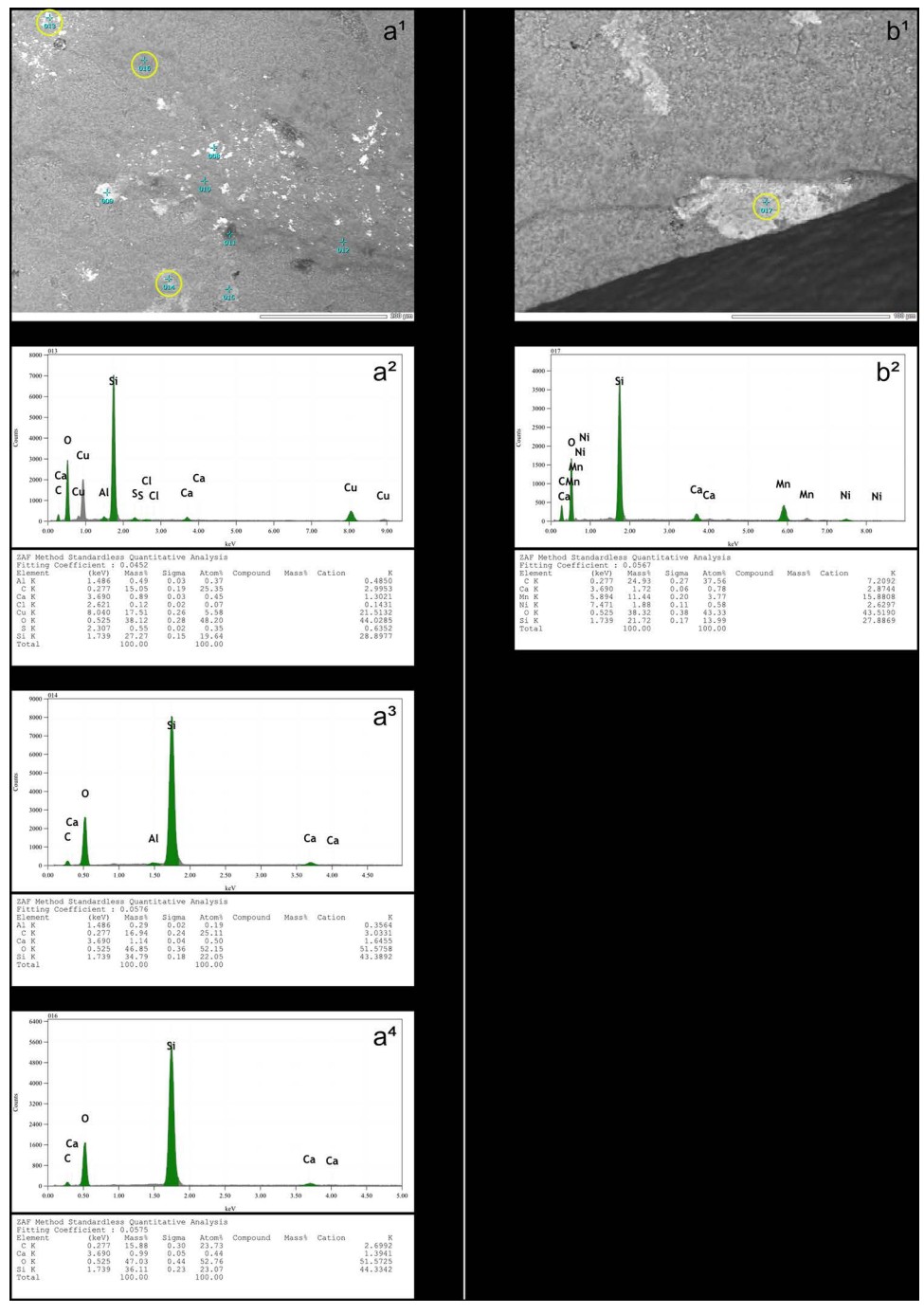

**Fig 7. On the left, SEM micrograph (a¹) of a section of the platform of a blade knapped by pressure with a copper CT (experimental piece Exp122-230), showing the locations (yellow circles) where the elemental spectra in a² (sample 013 shown in a¹), a³ (sample 014) and a⁴ (sample 016) were obtained.** The spectrum in a² comes from what was expected to be a copper residue and, in addition to high peaks for oxygen (O) and silica (Si), also shows moderate peaks for copper (Cu) and carbon (C), the latter indicating the presence of organic matter. In the spectrum in a³, carbon is also represented by a moderate peak. The spectrum in a⁴ comes from an area expected to contain only flint and logically shows high peaks for oxygen and silica, but carbon is still present. On the right, SEM micrograph (b¹) of a section of the same platform, with part of its posterior edge visible in the lower part of the image, showing the location (yellow circle) within a white area where the elemental spectrum in b² (sample 017 shown in b¹) was obtained. The spectrum in b² displays a moderate peak for carbon, indicating the presence of deposited organic matter in this area (SEM analysis: D. Cnuts; CAD: O. Touzé).

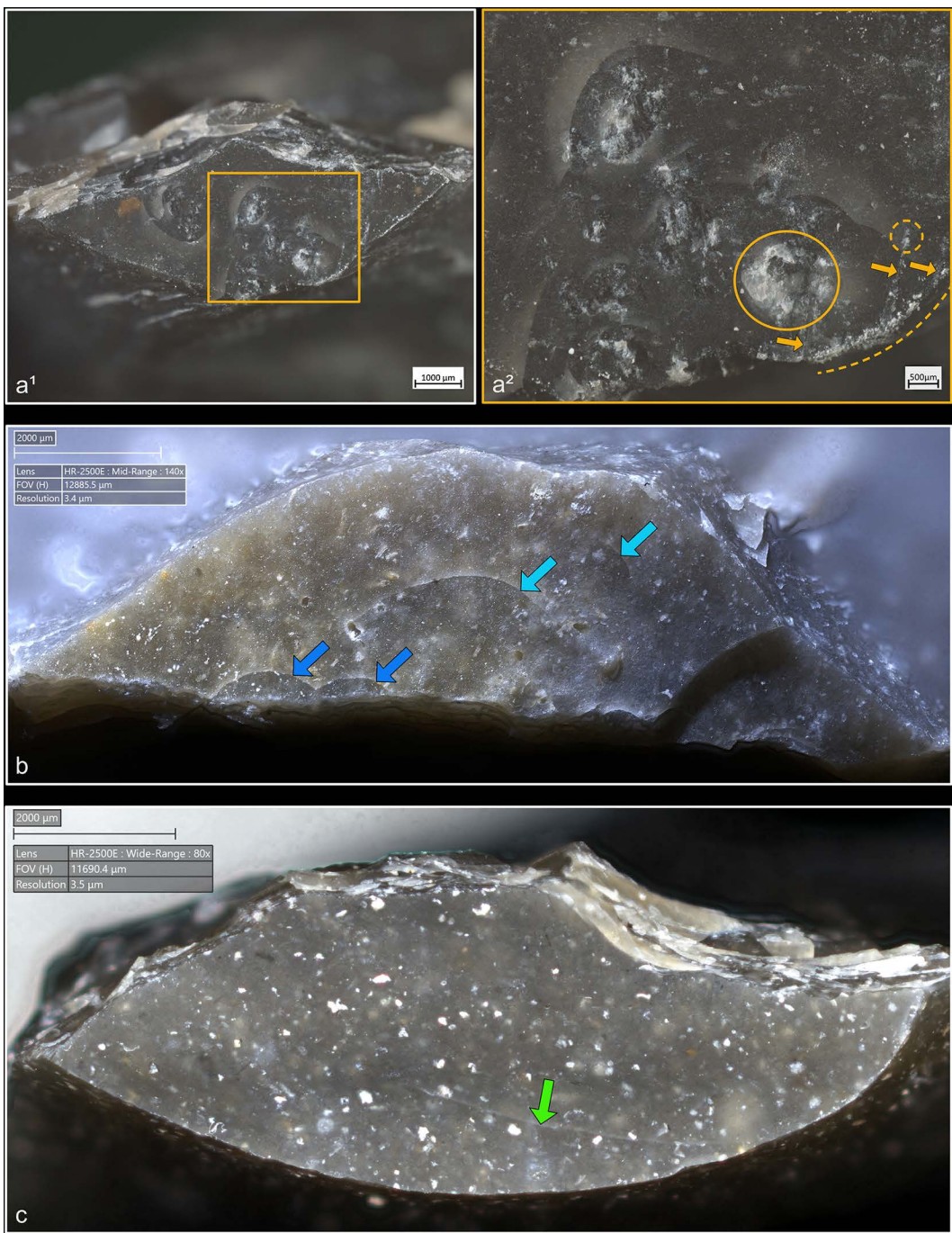

**Fig 8. Cracks associated with the use of direct percussion.** a: quartzite (complete and incomplete circle cracks); b: antler (light blue arrows: finger-nail cracks due to the detachment of the blade; dark blue arrows: cracks maybe due to the preparation of the edge of the striking platform); c: boxwood (slit crack). In image a², a small crushed area is visible (solid orange circle), along with silica dust particles located along the posterior edge of the platform (dotted orange curved line). Several striations extend from this same edge (orange arrows) and were probably created by silica particles being rubbed against the platform by the CT, as suggested by the location of some of these particles at the end of a striation (dotted orange circle).

**4.1.1.3. *Polishes*:** Polish observations were made only after complete or nearly complete removal of the residues through cleaning. Direct percussion almost always results in the formation of polishes (Figs 9–11), but the grain of the flint plays a role. In total, we counted 10 blades whose platforms show no polish: 5 of them were knapped with antler – half the number of blades obtained with this CT –, 2 with sandstone, and 1 with quartzite, bone and boxwood. Most of these blades (n = 7) were detached either from a rough and weathered non-cortical surface, or from a surface totally or partially characterised by a coarse grain. A Fisher's exact test (p-value = 0.031) confirms the existence of a link between the presence of polish and the grain size of the flint observed on the platform of the blades knapped by direct percussion. We can therefore deduce that the more weathered or coarser-grained the flint, the lower the probability that it will record the formation of knapping polishes. This corresponds to what has been noted previously for use-wear polishes (e.g., [154: 27,28, 174: 89, 175]). It should be noted, however, that the abrasiveness of the CT is also a factor to be considered. Of the 11 blades knapped with sandstone, all were extracted from a striking platform prepared in a coarse-grained area of the core, but only two of them have no polish as mentioned above.

The polishes develop only on the upper parts of the microtopography, but we did not observe any significant differences in their appearance or their brightness, which can vary greatly from one blade to another, regardless of the CT used (Fig 12). It seems likely that the fact that knapping polishes are formed in a very short time span, of the order of a second, limits or prevents the appearance of polishes with well-differentiated characteristics with regards to the CTs used, contrary to what is observed for polishes related to tool use. Although this remains speculative, we suspect that knapping polishes are created by an erosive levelling – i.e., a loss of material in the form of silica dust – of the higher parts of the microtopography, caused by the friction of the CT against the striking platform.

Several "spots" or small groups of polishes are generally present on the platforms (e.g., Fig 9), and it is rare for there to be only one (n = 1/13 with bone, 1/11 with boxwood). Their density in a given area is low to moderate with most of the CTs, except with quartzite where it is often moderate to high. This can probably be explained by a difference in abrasiveness, even though dense polishes were also observed on a blade knapped with boxwood. When several spots of polish are present, they are most often scattered in several places on the platform and rarely concentrated in a specific area (n = 2/11 with quartzite, 2/11 with sandstone, 1/11 with boxwood). The spots of polishes are also located at varying distances from the posterior edge of the platform, regardless of the CT used. In a few cases, the spots were located exclusively in contact with, or close to, this edge (n = 2/13 with bone, 1/11 with sandstone). Only one blade knapped with boxwood has polishes all located at a distance from the posterior edge.

The spatial association of polishes and cracks is also an interesting aspect of the distribution of polishes. Although this association is only partial most of the time, mineral hammers (n = 3/11 for sandstone, 2/11 for quartzite) can lead to a strict association of the two types of traces, which is more rare with organic hammers (n = 1/10 for antler, 1/13 for bone, no case for boxwood): in this case, the polishes are located inside (if cracks are circular), or in the immediate vicinity of the cracks. Conversely, the cases where the polishes are clearly dissociated from the cracks are mainly observed with organic hammers (n = 4/13 with bone, 3/11 with boxwood) and rarely with mineral hammers (n = 1/11 with sandstone).

The presence of incisions within the polishes is systematic. Visually, incisions appear to be the result of a deep removal of material, unlike the polishes which are formed on the most prominent parts of the microtopography. As we envisaged for the incisions associated with residues (see above), those associated with polishes are almost certainly created by highly abrasive particles [153] of silica already present on the surface of the CT, as supported by the SEM-EDS identification of a silica deposit on the contact surface of the copper indenter used for pressure knapping during the experiment (Fig 13), or detached during impact, which, under the combined effect of the compression exerted by the percussion and the friction of the CT against the striking platform, incise the surface of the latter (i.e., that of the butt of the blade) in depth.

In most cases, the polishes take the form of rectilinear lines of varying width, isolated or grouped (Figs 9d–11), as well as "patches" or isolated points with no particular morphology (Fig 9a–9c). In some cases, the polishes can be exclusively non-linear (n = 4, all CTs combined), or linear (n = 7, all CTs combined), but this is independent of the CT used. The linear

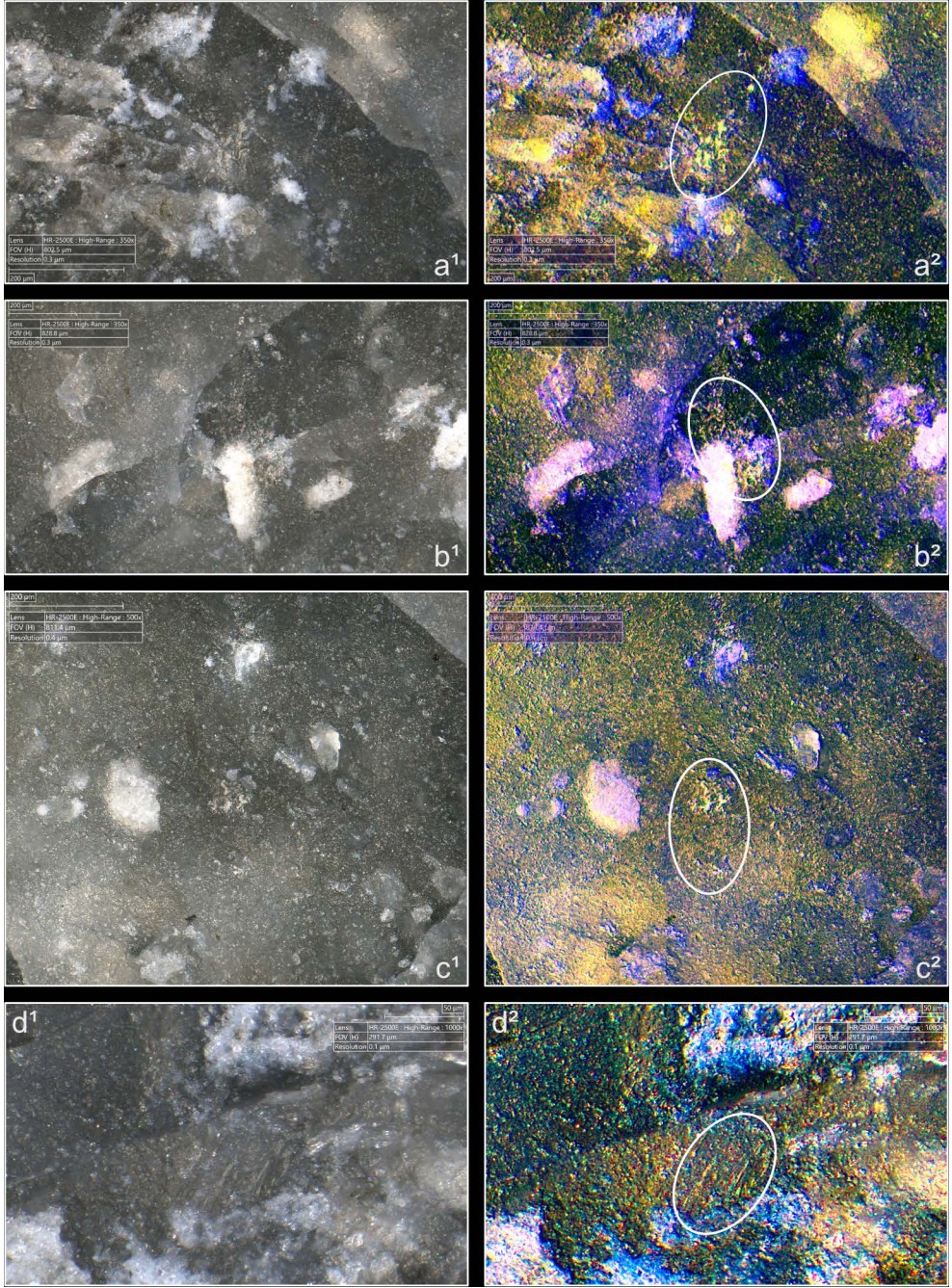

**Fig 9. Polishes associated with the use of direct percussion and a quartzite CT (images a², b², c² and d² modified with DStretch).**

nature of many polishes is a clear indication of the friction movement between the CT and the striking platform which occurs immediately after impact, and which is linked to the (more or less) tangential nature of the percussive gesture. Linear polishes are also often characterised by their discontinuous nature, their 'tracks' being made up of zones of polish more or less distant from each other (e.g., Fig 11c). However, continuous linear polishes can sometimes be formed, although this does

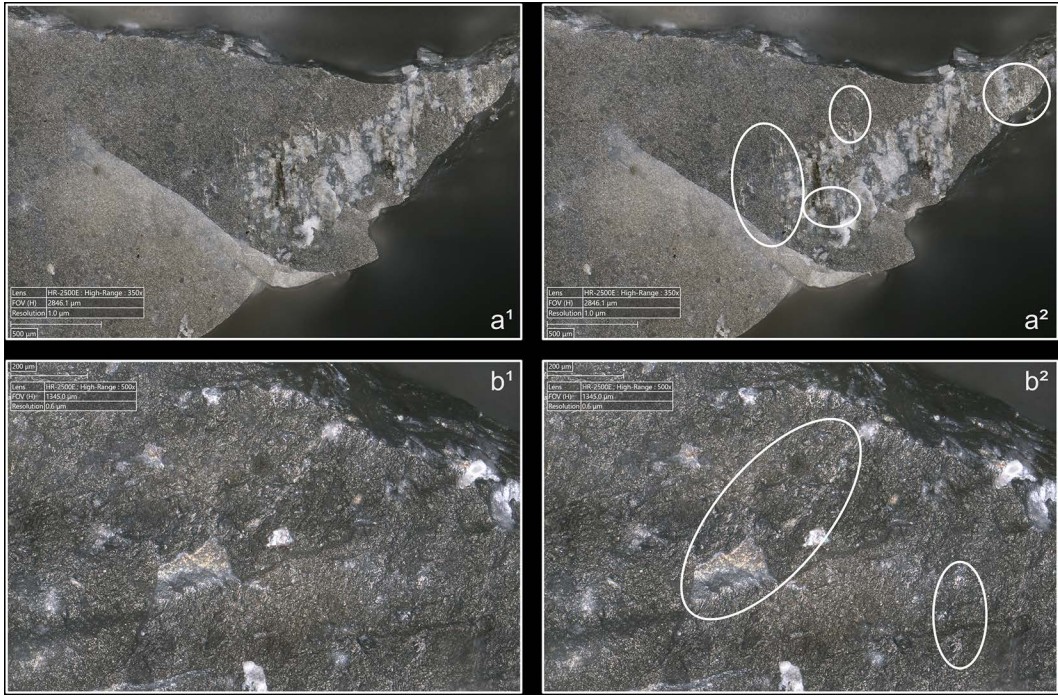

**Fig 10. Polishes associated with the use of direct percussion and a bone CT.**

not seem to be linked to the abrasiveness of the CT, as continuous linear polishes were observed with boxwood (n = 3/11), quartzite (n = 2/11) and sandstone (n = 2/11). Finally, it can happen that the orientations of the linear polishes are secant rather than parallel: this phenomenon indicates that repeated attempts at detachment were made, since the slightest change in the trajectory of the percussion gesture, or in the orientation of the striking platform mechanically leads to the formation of linear polishes of different orientations. In the experimental collection, polishes with secant orientations are most common with quartzite (n = 6/11), which suggests that the knapper had slightly more difficulty extracting the blades with this CT.

### 4.1.2. Indirect percussion.

**4.1.2.1. *Residues*:** The deposition of residues on the surface of the platform is common with indirect percussion (Fig 14), but cases where residues are absent are more frequent than with direct percussion (n = 4/11 with antler, 1/11 with boxwood, 1/4 with bone). When they are present, residues are generally scattered over the platform and rarely concentrated in a particular, limited, area (n = 1/11 with antler as well as with boxwood, 0/4 with bone), and it is exceptional for them to be located within a crack (n = 1/11 for antler), unlike what is observed with direct (mainly mineral) percussion. The shapes of the residue deposits are varied and non-specific, but a few cases of residues with a partly linear distribution were observed with boxwood (n = 2/11). The presence of incisions within the residues is common (Fig 14a, 14b), but a few exceptions were also observed (n = 3/11 with boxwood, 1/4 with bone).

The knapping residues associated with indirect percussion can be resistant, but a cleaning procedure combining exposure to a 3% hydrochloric acid solution combined with the use of an ultrasonic bath for around ten minutes has often been sufficient to remove them, or at least to remove most of them. By comparison, organic residues proved to adhere more strongly in the case of direct percussion.

In our experimental collection, indirect percussion is also associated with the systematic presence of a more or less continuous trace along the posterior edge of the platform (Fig 15a), similar to that observed on most blades knapped by

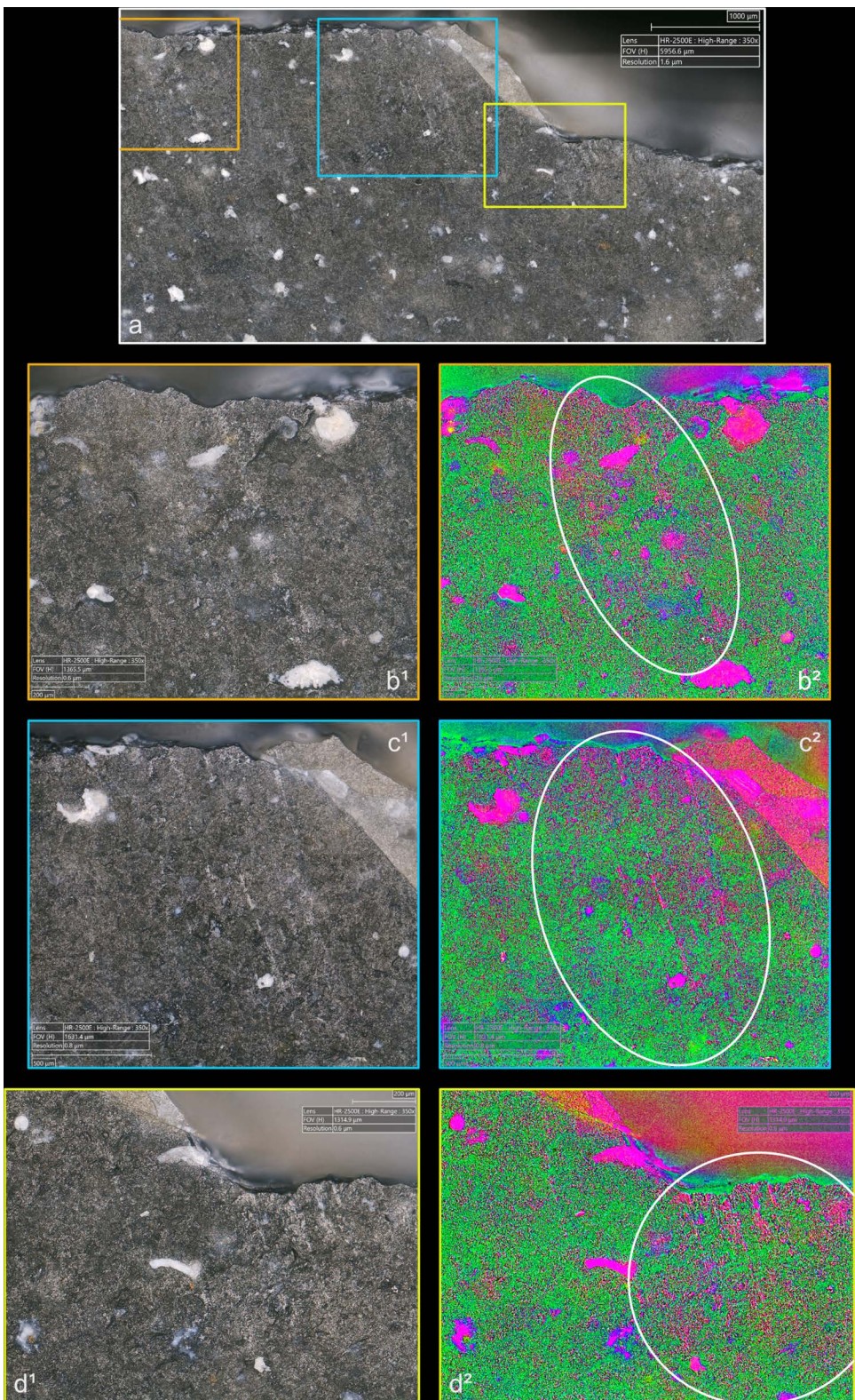

**Fig 11. Polishes associated with the use of direct percussion and a boxwood CT (images b², c² and d² modified with DStretch).**

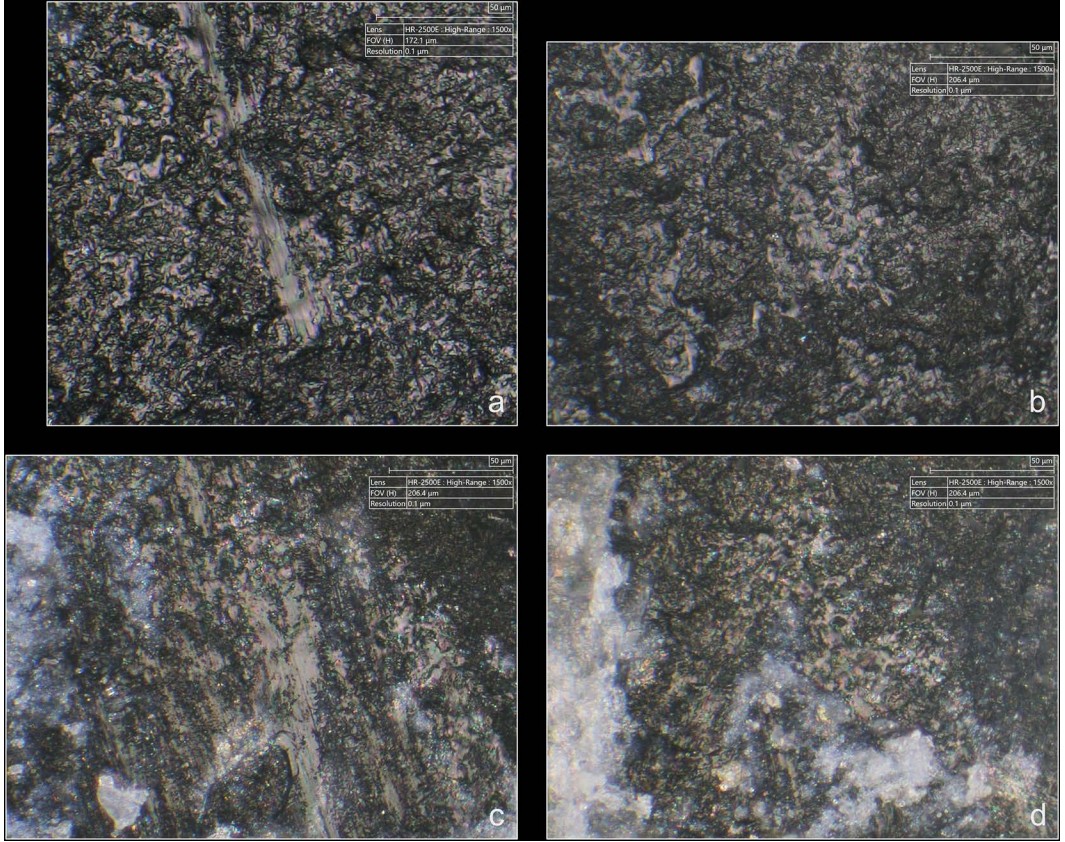

**Fig 12. Different aspects of polishes observed on the platform of two blades associated with the use of direct percussion at magnification x1500.** a-b: boxwood (experimental piece Exp122-56); c-d: quartzite (experimental piece Exp122-44). The polishes in the images on the left have a continuous aspect because they have formed with little interruption on the upper and lower parts of the microtopography. They also have a distinctly linear appearance and are associated with narrow incisions. The polishes in the photos on the right have formed mainly on the upper parts of the microtopography and are more discrete in the lower parts, hence their more "fragmented" appearance. Their linear nature is less obvious.

direct percussion and on all blades knapped by pressure. As mentioned above, SEM-EDS analysis has shown that this trace corresponds to a residue of organic material, potentially deposited by the knapper on the active part of the CT before the blade was detached.

**4.1.2.2. *Cracks*:** The cracks present on the platform of blades knapped by indirect percussion (Fig 16) show some differences depending on whether the CT is antler or boxwood. Despite the inefficiency of the bone punch, the characteristics of the cracks associated with this material are fairly similar to those associated with antler.

Only boxwood is associated with platforms without cracks, although this situation is not frequent (3 cases out of 11). When cracks are present, boxwood is generally associated with only one (n = 6/11; compared with 2 cases of multiple cracks: Fig 16c), whereas antler (3 cases of single cracks and 7 cases of multiple cracks; the presence of cracks on the last blade could not be determined; Fig 16a, 16b) and bone (1 case of single crack and 3 cases of multiple cracks) are more often associated with several. The number of cracks observed is probably related to the number of detachment attempts made to detach the blades. When several cracks are present, they are more often scattered over the platform (n = 5/11 with antler; 2/11 with boxwood) than concentrated in a specific area, except sometimes with antler (n = 2/11) and especially with bone (n = 3/4). As with direct percussion, cracks associated with the use of indirect percussion are either in contact with, or at varying distances from the posterior edge of the platform. In two cases, the boxwood punch created

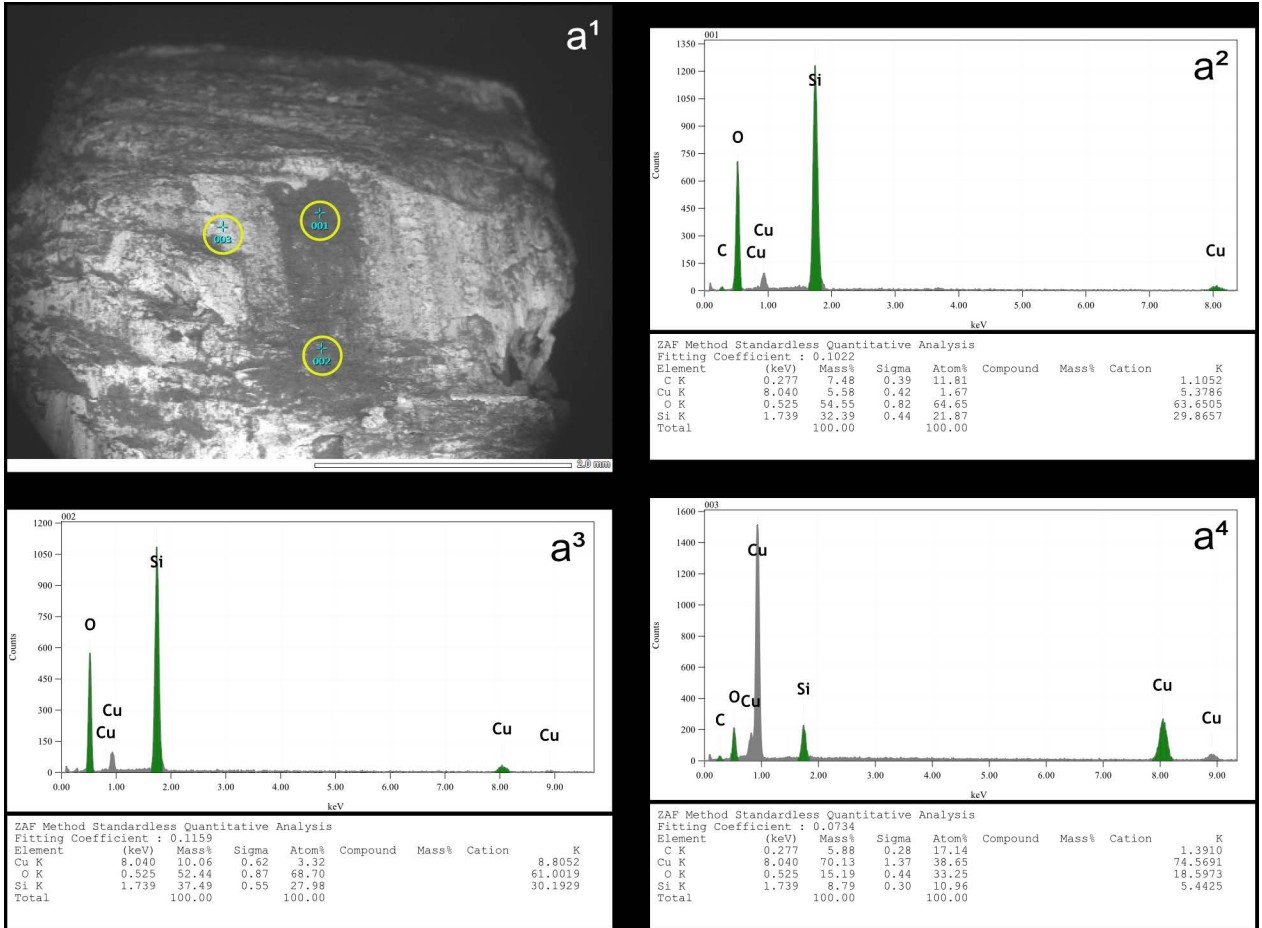

**Fig 13. SEM micrograph (a¹) of the contact surface of the copper indenter used for pressure knapping, showing the locations (yellow circles) where the elemental spectra in a² (sample 001 shown in a¹), a³ (sample 002) and a⁴ (sample 003) were obtained.** The spectra in a² and a³ come from the dark area shown in a¹ and display high peaks for oxygen (O) and silica (Si), as well as low peaks for copper (Cu), indicating that this dark area corresponds to a silica deposit. In contrast, the spectrum in a⁴ comes from a location slightly away from the dark deposit and shows a high peak for copper and lower peaks for oxygen and silica (SEM analysis: D. Cnuts; CAD: O. Touzé).

cracks close to the posterior edge, but not in contact with it. The cracks created by indirect percussion with the three CTs are never associated with crushed areas, contrary to what is sometimes observed with direct percussion with quartzite.

Cracks caused by the boxwood punch are always small in relation to the surface area of the platform, whereas those caused by the antler punch can be extensive (n = 6/11). Extensive cracks are also possible with bone (n = 1/4; compared with 3 cases of small cracks). Circular cracks are often observed with antler and bone, but rarely with boxwood. Circular cracks with a protrusion of the posterior edge are rare and only observed with antler (n = 1/11 for complete as well as incomplete cracks; Fig 16b), while circular cracks without a protrusion are also rare when they are complete (n = 1/11 with antler, 1/4 with bone) but more frequent when they are incomplete (n = 3/11 for antler; 1/4 for bone; 1/11 for boxwood; Fig 16a). In contrast to circular cracks, fingernail and slit cracks are a little more frequent with boxwood (n = 4/11 for both types; Fig 16c), but they are not uncommon with the other CTs either (n = 3/11 and 2/11 respectively with antler; 2/4 and 1/4 respectively with bone). Type 1 lateralized cracks are observed with all CTs (n = 2/11 with antler as well as with boxwood; 1/4 with bone), while network cracks are infrequent and are only observed with antler (n = 2/11).

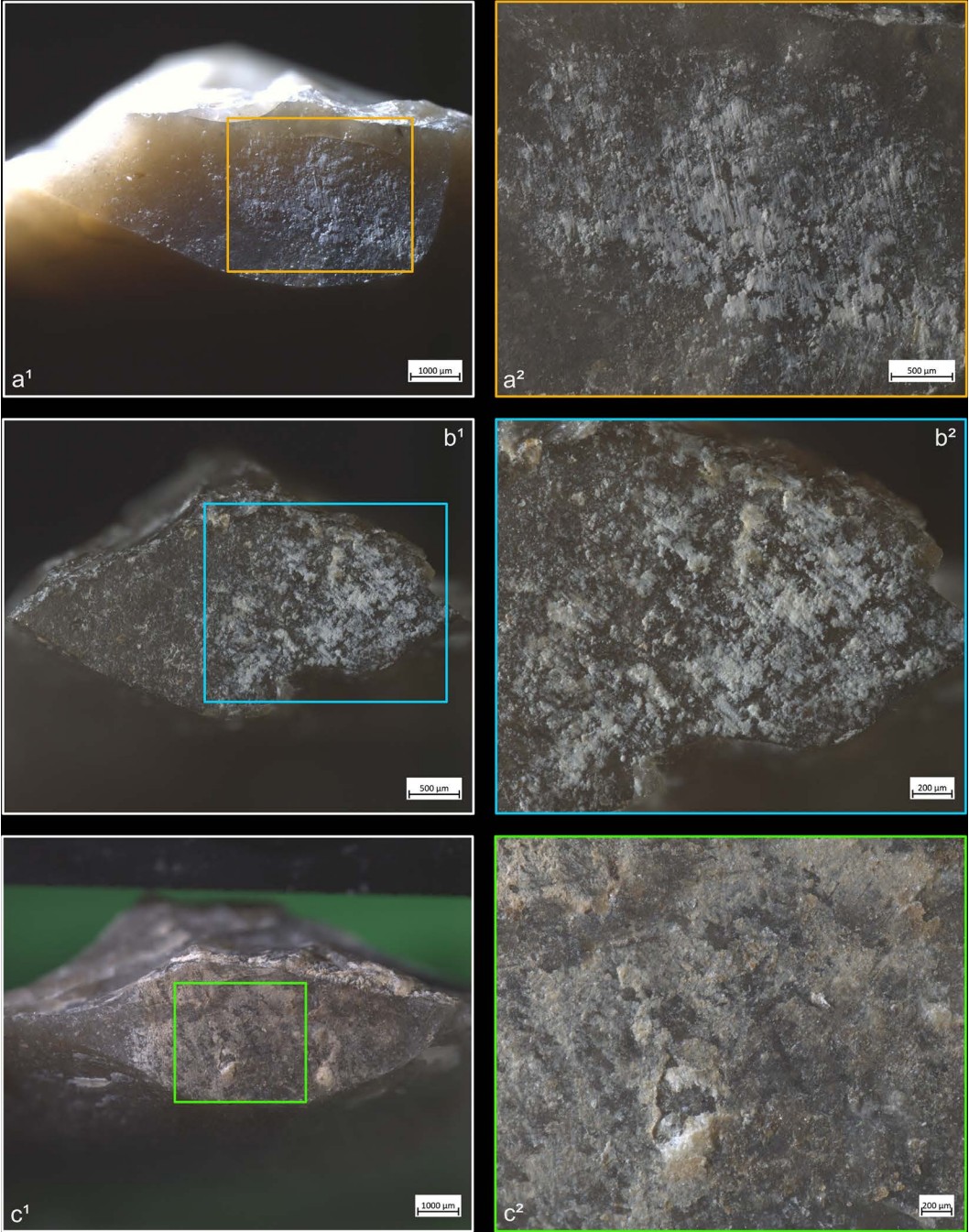

**Fig 14. Residues associated with the use of indirect percussion.** a: antler; b: bone; c: boxwood.

 **4.1.2.3. *Polishes*:** The friction of the CT against the striking platform almost always leads to the formation of polishes (Figs 17–19), except in the case of boxwood where they may be sometimes absent (n = 2/11). When polishes are present, several spots can be observed. Their density in a given area is low or moderate, but high density can very occasionally be observed with antler (n = 1/11). Polishes are generally scattered over the platform and, conversely, they are rarely

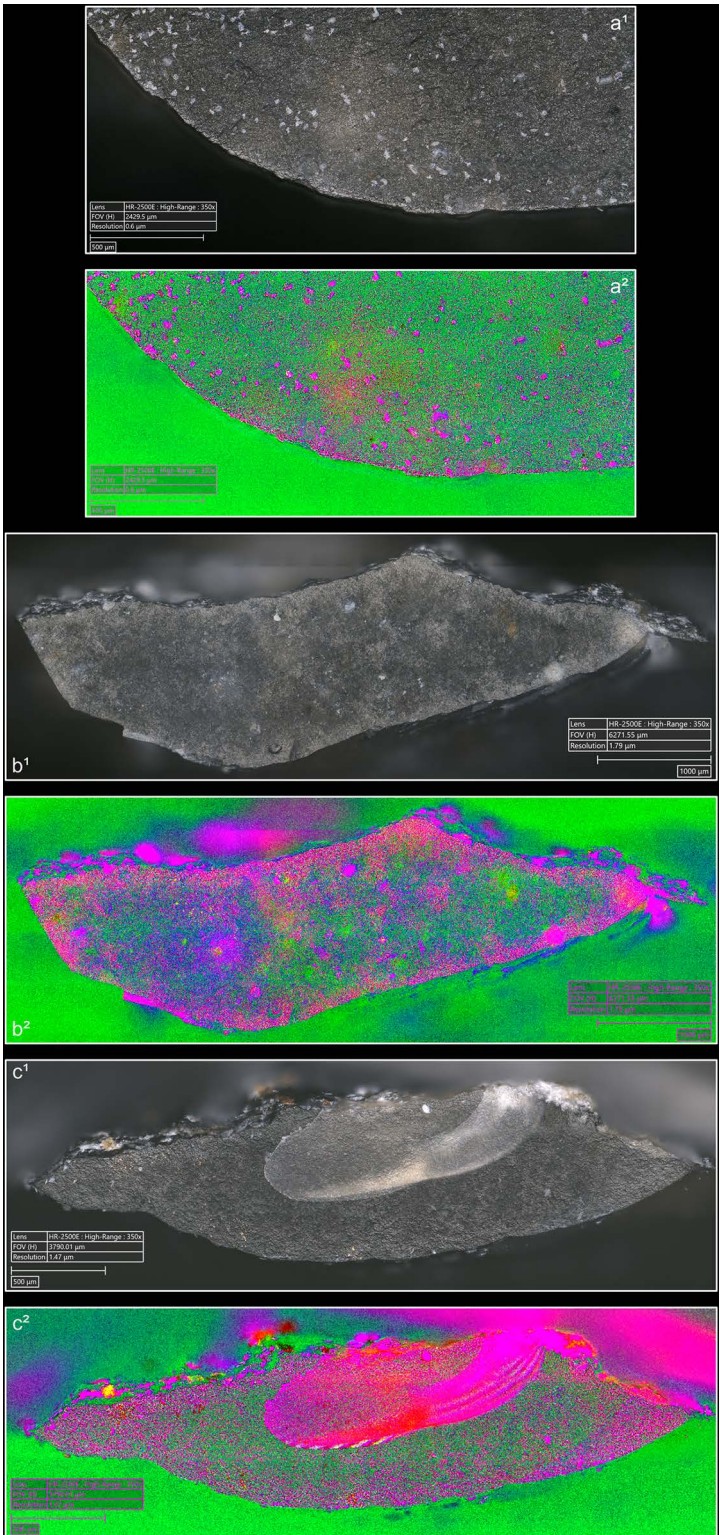

**Fig 15. Organic residues located along the edges of the platform.** a: indirect percussion with antler; b: pressure with antler; c: pressure with copper (images a², b² and c² modified with DStretch). On images a, the residues are located along the posterior edge; on images b and c, they are located along both edges. On images modified with DStretch, areas with residues appear in pink.

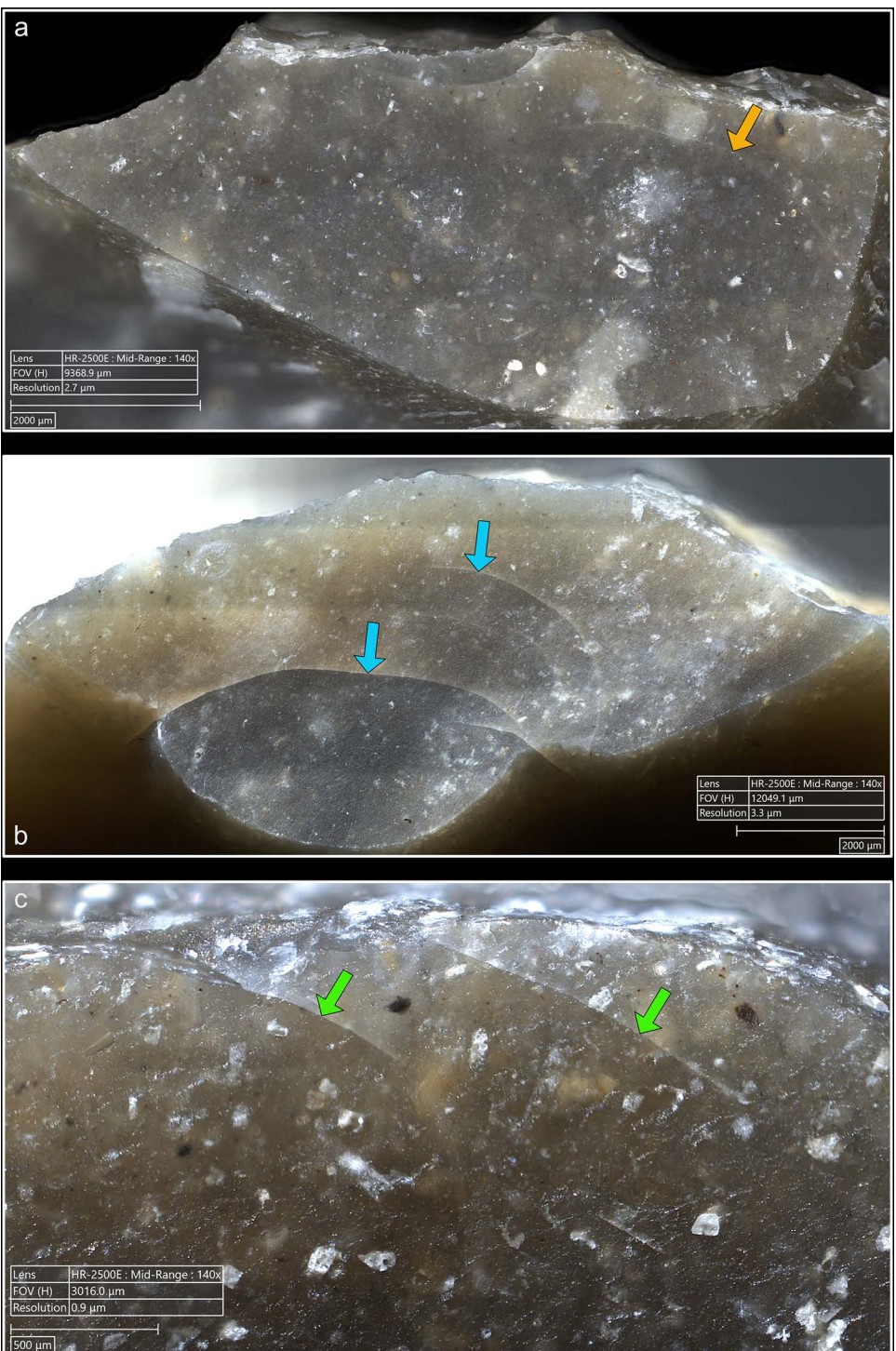

**Fig 16. Cracks associated with the use of indirect percussion.** a: antler (incomplete circle crack); b: antler (complete circle crack with protrusion and incomplete circle crack); c: boxwood (slit cracks; in this case, the posterior edge of the platform is on top of the image).

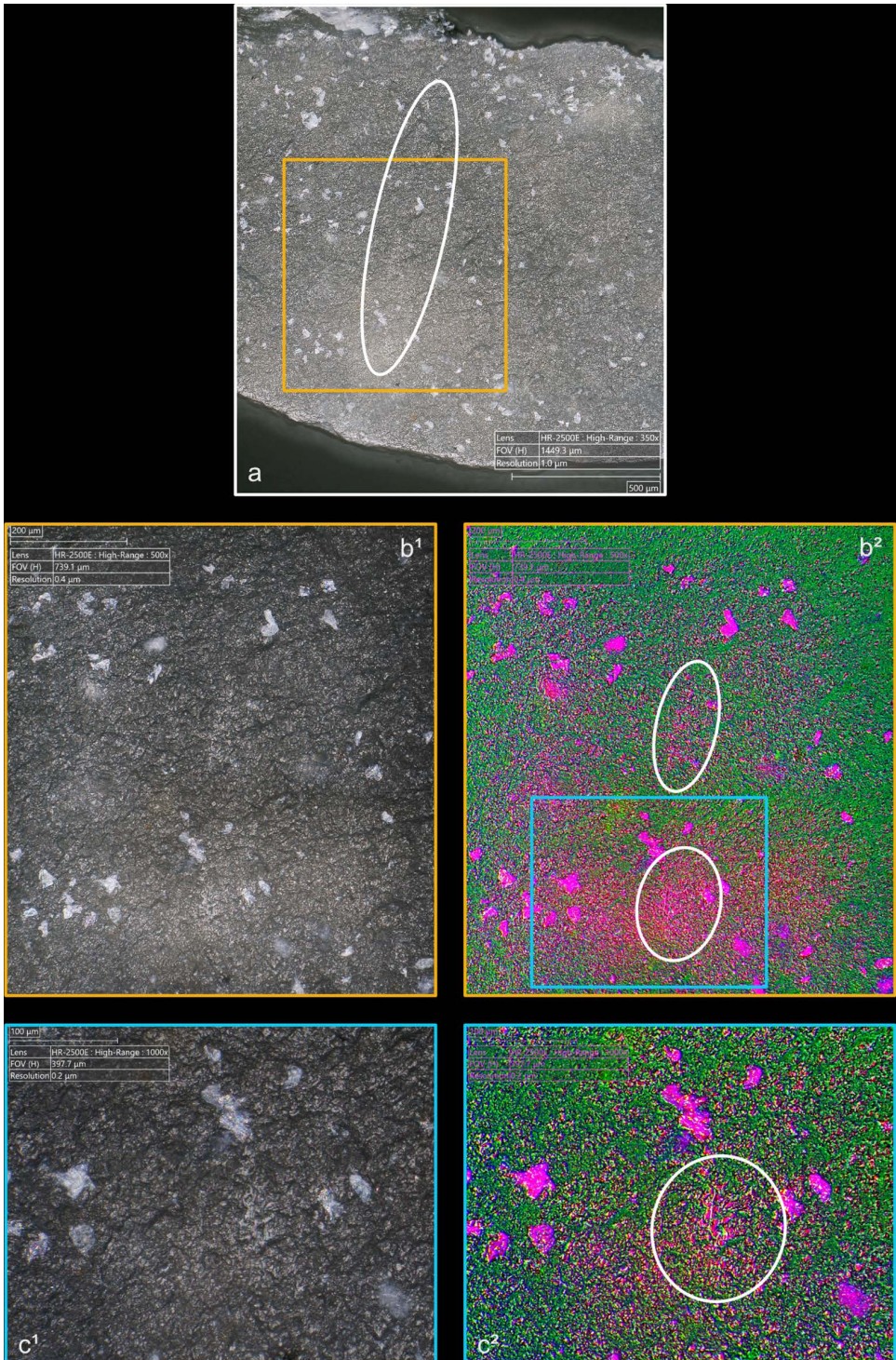

**Fig 17. Polishes associated with the use of indirect percussion and an antler CT (images b² and c² modified with DStretch).**

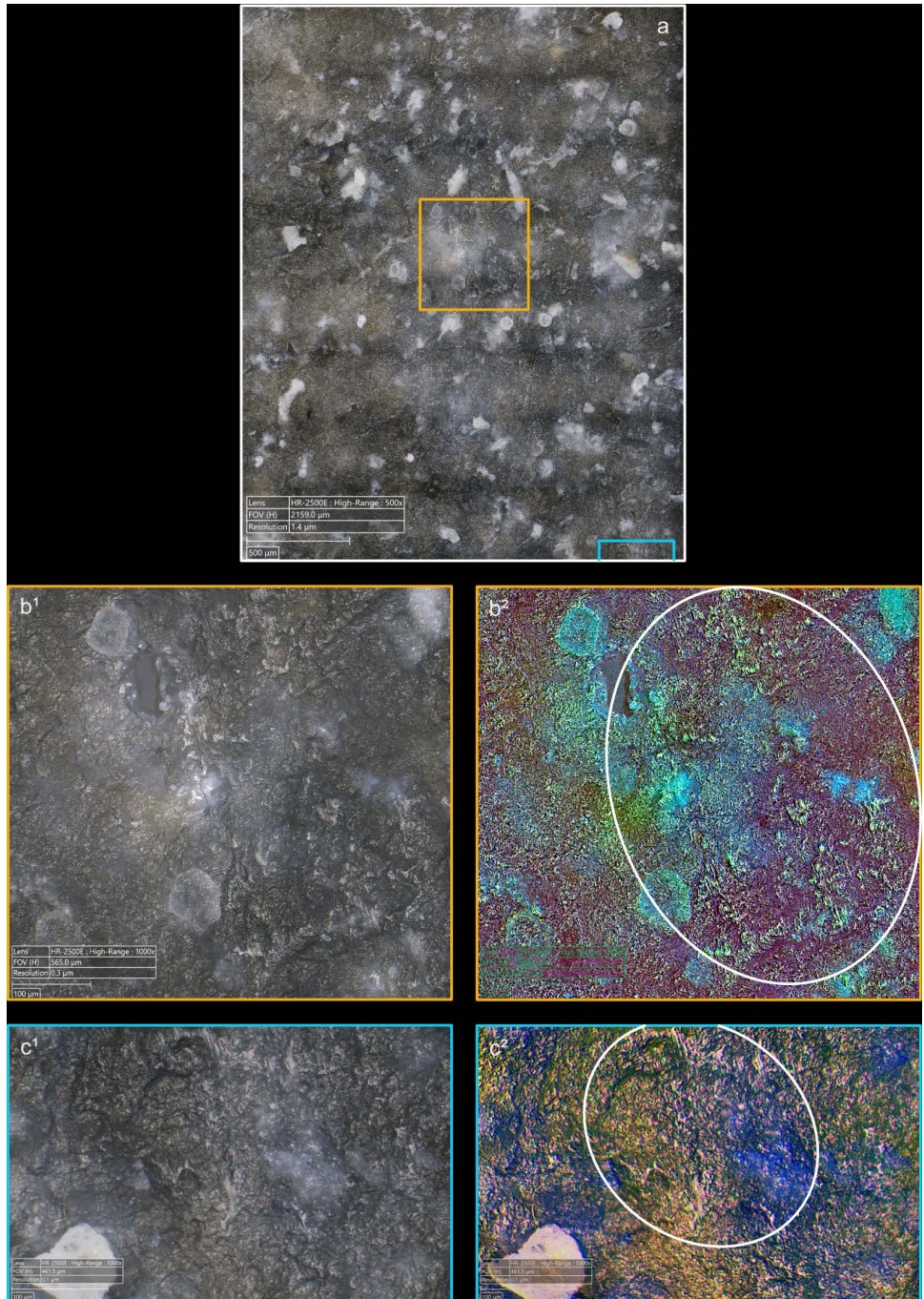

**Fig 18. Polishes associated with the use of indirect percussion and a boxwood CT (images b² and c² modified with DStretch).**

concentrated in a specific area (only 1 case observed for each CT). They are also always located at varying distances from the posterior edge and are never located solely in contact with, or further away from it. As with direct percussion, the spatial association between polishes and cracks is only partial most of the time. However, antler and bone can

occasionally create polishes directly associated with cracks (n = 2/11 and 1/4 respectively), whereas these two categories of traces are sometimes completely dissociated when the CT is boxwood (n = 2/11).

The presence of incisions within the polishes is extremely common, as only two blades knapped with antler do not have any.

The platforms of blades knapped by indirect percussion show both linear and non-linear polishes, and it is very rare for the former to be absent (only 1 case with antler). Linear polishes are almost always discontinuous (e.g., Fig 18b; only 1 case of continuous linear polish was observed with antler) and rectilinear. However, we observed a linear polish that was curved instead of rectilinear on a blade knapped with antler (Fig 19), which could be the result of a slight slippage of the

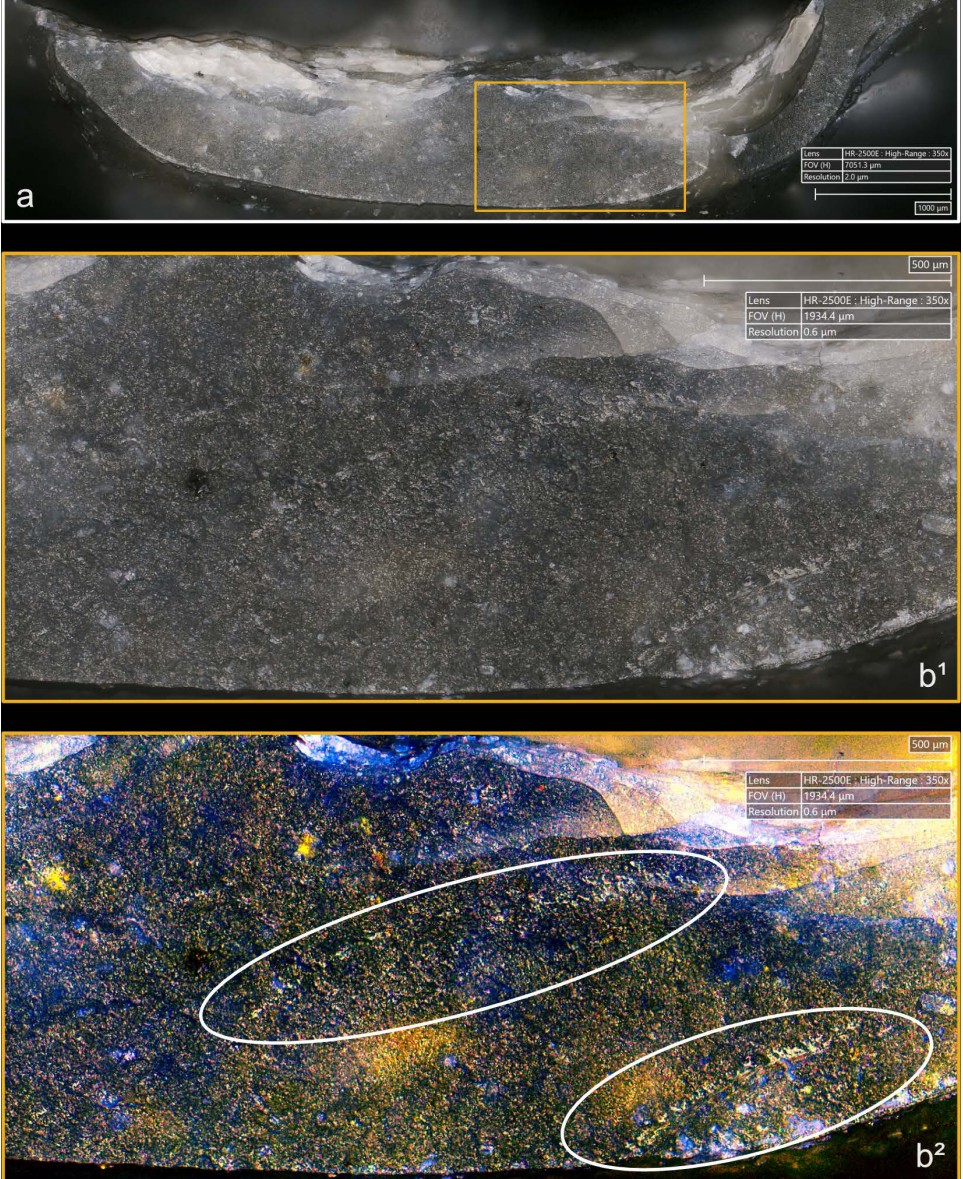

**Fig 19. Linear curved polishes associated with the use of indirect percussion, probably created by a slight slippage of the antler CT on the striking platform (image b² modified with DStretch).**

CT during its contact with the striking platform. The orientations of the linear polishes are slightly more often secant than parallel with boxwood (n = 5/11 and 4/11 respectively; the other two blades have no polish), contrary to antler (n = 4/11 for linear polishes with secant orientations and 6/11 for linear polishes with parallel orientations; the last blade has no linear polish) and bone (n = 2/4 for each orientation), which seems to indicate that the knapper had a little more difficulty extracting blades with the boxwood punch, given that secant orientations indicate multiple detachment attempts.

### 4.1.3. Pressure.

**4.1.3.1. *Residues*:** The deposition of knapping residues is common when a blade is detached by pressure (Fig 20), but the nature of the CT is important, unlike what is observed with direct and indirect percussion: while the presence of residues is systematic with copper and very frequent with antler (2 blades out of 11 have no residues), boxwood stands out with a high proportion of platforms showing no residues (n = 9/14). This difference, which is supported by a Fisher's exact test (p-value < 0.001), could be explained by the less abrasive nature of boxwood and the nature of the contact between the CT and the core's platform. In the case of pressure, this contact is characterised by limited friction because the tip of the crutch remains in contact with a precise point of the platform (unless an accidental slippage happens), located close to its edge. Reduced friction with a little abrasive material like boxwood is therefore less likely to cause the deposition of residues on the butts of the blades.

Residues associated with pressure with antler and boxwood CTs are often concentrated in a specific area of the platform (n = 7/11 and 4/14 respectively; Fig 20a). Copper, on the other hand, mainly leaves scattered residues (n = 11/13, compared with 2 cases of concentrated residues). We have noticed that when preparing to extract a blade using pressure, the knapper placed the CT in different places until he found the ideal one. This behaviour probably partly explains the greater dispersal of copper residues, as the copper CT leaves residues more easily than the organic CTs. The frequent scattering of copper residues, and the less frequent scattering of organic residues can also be the result of successive attempts to detach a blade from the same area of the pressure platform.

The distribution of residues also shows a recurrent spatial association, complete or partial, with cracks in the case of copper (n = 2/13 for complete association and 9/13 cases for partial association; Fig 20c) and antler to a much lesser extent (n = 1/11 for complete association and 2/11 for partial association). When residues and cracks are present on blades knapped with boxwood, these two types of traces are dissociated (n = 2/14; the other blades show no residue or crack), a situation that is also observed sometimes with the other CTs (n = 2/13 with copper; 1/11 with antler).

Morphologically, the residues always appear as patches of variable shape, but never in the form of linear deposits, which can be explained by the limited friction induced by pressure knapping. In addition, incisions are also rarely observed (n = 1/13 for copper residues; 1/11 for antler residues; 2/14 for boxwood residues; Fig 20a).

Our attempts to eliminate copper residues were limited and mostly unsuccessful, but we were generally able to remove completely organic residues by combining the use of a 3% hydrochloric acid solution and an ultrasonic bath for around ten minutes. The adhesion of organic residues therefore appears to be weaker than that of organic residues associated with the use of direct percussion, or even indirect percussion. In the case of pressure knapping, the knapper applies a continuous thrust until the combination of the force exerted and the angle of incidence reaches the threshold necessary to initiate the fracture, whereas in the case of direct and indirect percussion, the fracture is initiated by an impact, which means that the force applied cannot be adjusted over a certain period of time. Furthermore, in the case of pressure with an abdominal crutch, knappers use a significant portion of their body mass, rather than just one of their arms as in the case of direct or indirect percussion, which should result in greater force. We therefore suspect that the fact that residues associated with pressure knapping show less adhesion and are easier to remove than residues associated with percussion is not solely due to the amount of force exerted, but also to whether the fracture is initiated by an impact or a continuous contact of the CT against the striking/pressure platform.

As with indirect percussion, the platforms of the blades knapped by pressure always show a deposit of organic matter along their posterior edge, and sometimes also along their anterior edge (Fig 15b, 15c). This deposit may or may not be

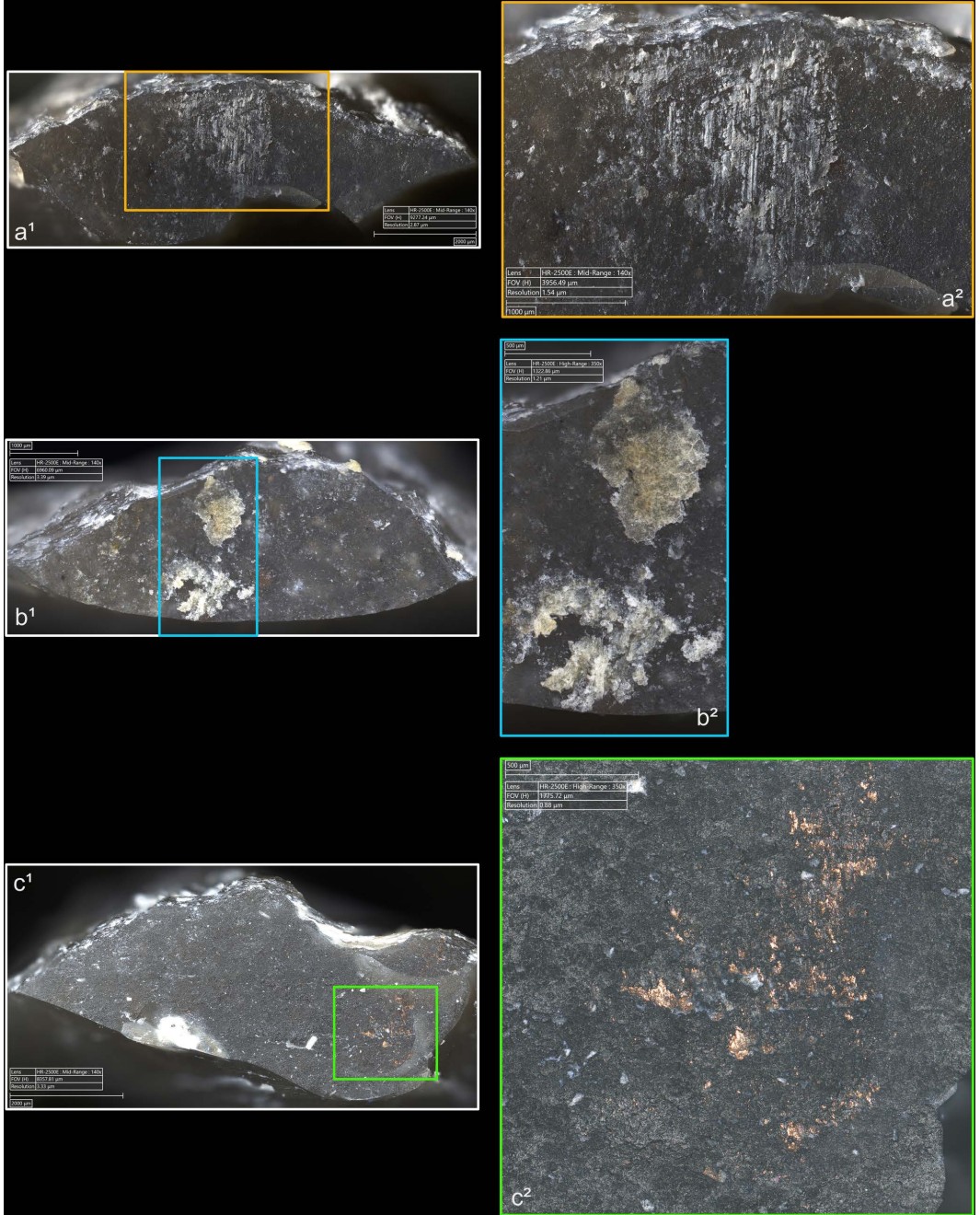

**Fig 20. Residues associated with the use of pressure.** a: antler; b: boxwood; c: copper.

continuous and is often well developed laterally, at the junction between both edges of the platform. The cleaning procedures used have usually reduced the density of these residues but have not made them disappear completely. These organic deposits appear both with the organic and with the copper CTs, suggesting that they result from handling of the active part of the indenter by the knapper, which probably takes place shortly before the detachment of each blade, when the knapper places the crutch on the pressure platform.

Finally, pressure knapping is characterised by the systematic presence of residue on the dorsal face of the blades. There are two reasons for this. Firstly, once extracted, the blade is not retained by the wood device used to hold the core. The blade therefore tilts forward and can accumulate organic residues when it hits the device. Secondly, blades knapped with copper often show a combination of copper and organic residues on their dorsal face (n = 7/13, compared with 5 cases of organic residues only and 1 case of copper residues only). These blades therefore prove that the tip of the crutch, which is initially placed close to the edge of the pressure platform, may slip and rub against the dorsal face of the blade shortly after fracture initiation, and contribute to the deposition of residues. In comparison, the presence of residues on the dorsal face of the blades is anecdotal with direct percussion (n = 3/56) and indirect percussion (n = 2/26) and is almost always attributable – the residues being mineral and not organic in both cases – to the accidental fall of the blades onto the ground, where knapping waste had previously accumulated.

**4.1.3.2. *Cracks*:** With copper, cracks appear systematically, whereas with antler exceptions are possible (3 blades out of 11 have no cracks). With boxwood, the softest material we used, the absence of cracks is the most common situation (n = 9/14). Cracks are generally multiple with copper (n = 11/13), multiple or single with antler (n = 4/11 for each situation), and mainly single with boxwood (n = 4/14, and only 1 case of multiple cracks).

When they are multiple, the cracks are most often grouped in a specific area of the platform, which corresponds to the area where the CT was placed on the pressure platform. When the cracks are not grouped (n = 3/13 with copper and 1/11 with antler), their scattering is probably the result of successive detachment attempts, the knapper having moved the position of tip of the crutch slightly between each attempt. The cracks are usually in contact with the posterior edge of the platform and are rarely just close to it (n = 1/11 with antler), or at a distance from it (n = 2/13 with copper). As with indirect percussion and direct percussion with all CTs except quartzite, there is no crushing associated with the cracks.

Cracks associated with the use of pressure tend to be small relative to the dimensions of the platforms, but some extensive cracks are sometimes observed with copper (n = 4/13; Fig 21b) and to a lesser extent with antler (n = 1/11). Morphologically, circular cracks are not observed with boxwood, whose lower hardness combined with the absence of impact probably reduce the possibility of their occurrence. Circular cracks with protrusion are never complete, but incomplete cracks can appear with copper (n = 4/13). Circular cracks without protrusion are associated with copper (n = 2/13 for incomplete cracks, no case for complete cracks), as well as with antler (n = 2/11 for complete cracks, no case for incomplete cracks). Among non-circular cracks, fingernail cracks are frequent with copper (n = 7/13) and are not observed with the other CTs, whereas slit cracks are found with both organic CTs (n = 3/14 for boxwood, 2/11 for antler; Fig 21a). Type 1 lateralized cracks are rare and do not occur with boxwood (n = 1/13 for complete as well as for incomplete cracks with copper; 1/11 for complete cracks with antler), as are sinusoidal cracks (1 case with copper and antler) and rectilinear cracks (1 case with copper). Finally, network cracks are observed with all three CTs, but is rare with boxwood (n = 4/13 for copper; 3/11 for antler; 1/14 for boxwood; Fig 21b).

**4.1.3.3. *Polishes*:** The presence of polishes is common with pressure with antler (n = 10/11; Fig 22) and copper (n = 10/13; Fig 23) but not frequent with boxwood (n = 3/14). These polishes are always distributed in several zones, generally scattered over the surface of the platform and rarely concentrated in a specific area (1/13 for copper and n = 1/14 for boxwood). The density of polishes is low to moderate but can occasionally be high with copper (n = 2/13). These spots are always located at varying distances from the posterior edge and are only partially associated with cracks. In a few cases, however, polishes and cracks are strictly associated (n = 1/13 with copper and 1/14 with boxwood) or dissociated (n = 1/13 with copper).

Polishes on blades knapped by pressure are often non-linear, but cases of at least partially linear polishes are not uncommon (n = 4/11 for antler, n = 4/13 for copper, n = 3/14 for boxwood). In addition, linear polishes are always associated with incisions regardless of the CT used, unlike the non-linear ones which are never associated with them. Linear polishes with incisions clearly reflect significant friction between the CT and the pressure platform, whereas the non-linear polishes were probably created under conditions involving less friction and/or force.

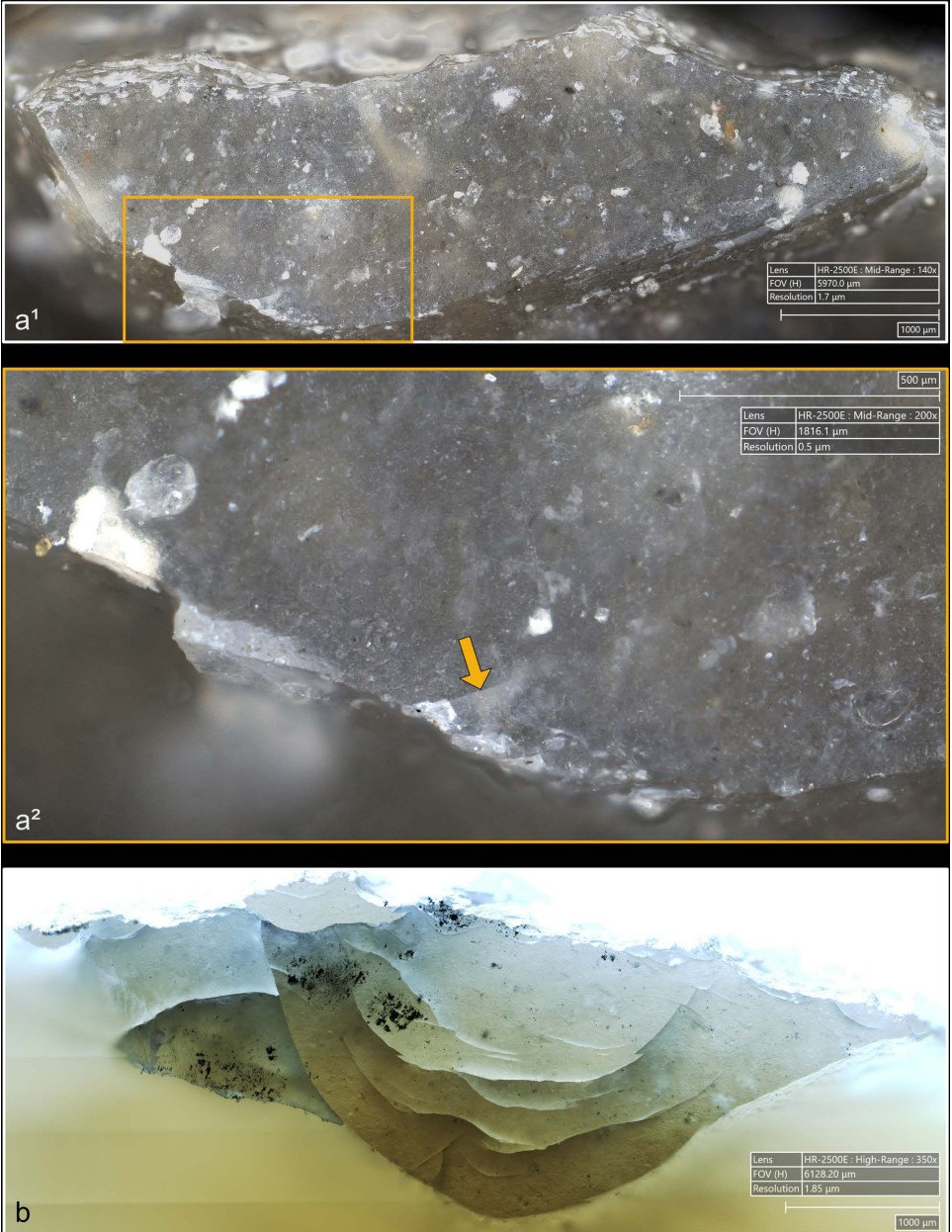

**Fig 21. Cracks associated with the use of pressure.** a: antler (slit crack); b: copper (network crack; copper residues appear as black spots due to the positioning of the light source in front of the piece and not above it).

When they are linear (Figs 22 and 23), the polishes created by copper and antler can be discontinuous or continuous, and this in equivalent proportions (2 cases recorded for each possibility with both CTs). With boxwood, we only observed discontinuous linear polishes. Furthermore, although linear polishes are in principle rectilinear, a blade knapped with antler has a curved polish, which likely indicates that the CT has slipped slightly during its contact with the pressure platform. Finally, as with direct and indirect percussion, if several detachment attempts are made in the same area of the pressure platform before the blade is successfully extracted, the linear polishes associated with pressure knapping will

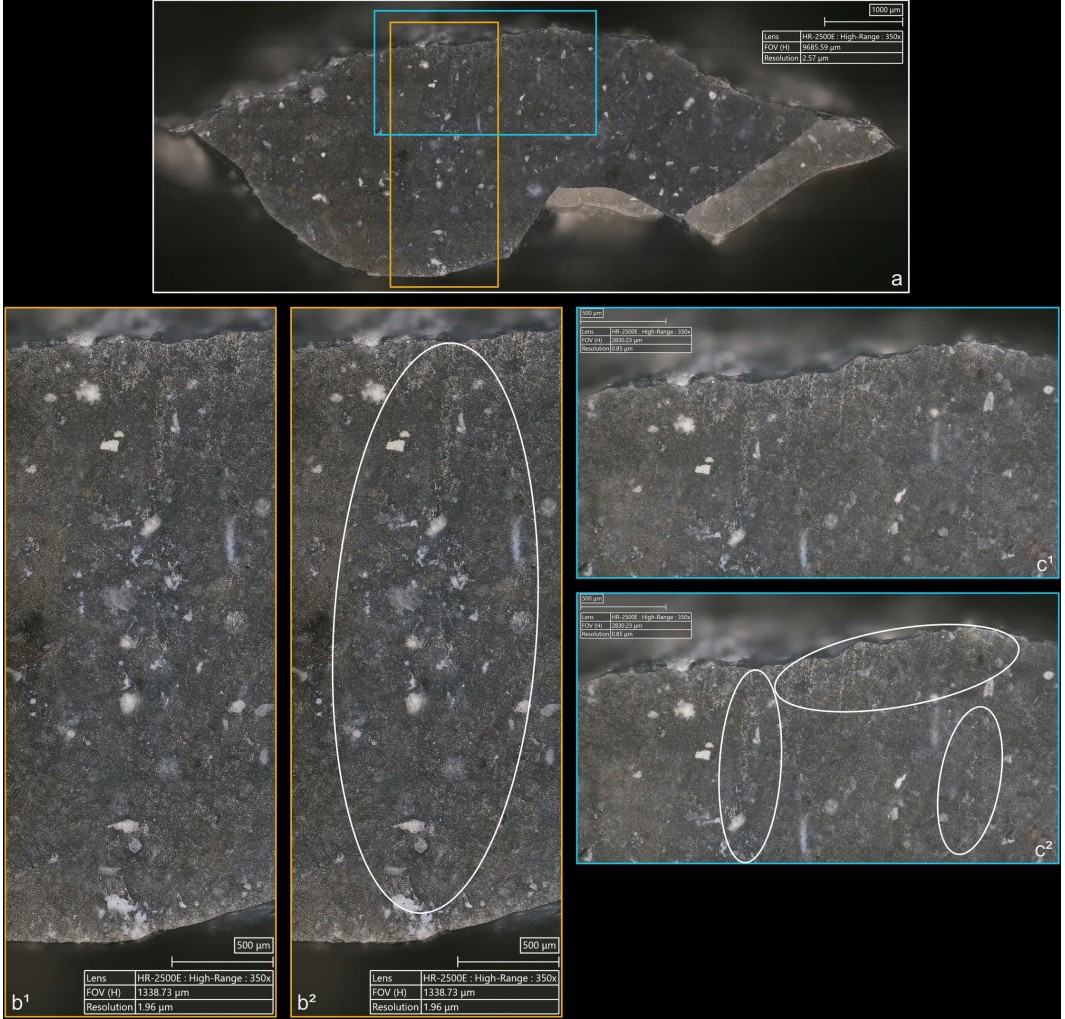

**Fig 22. Polishes associated with the use of pressure and an antler CT.**

show different orientations (Fig 23). In our experimental collection, this phenomenon was observed with all CTs (n = 3/11 for antler, 2/13 for copper, 1/14 for boxwood).

**4.1.4. Traces associated with the preparation of the edge of the striking/pressure platform.** Before a blade is detached, the edge of the striking or pressure platform has almost always been prepared by the knapper by means of an abrasion directed towards the flaking surface. This abrasion causes small semicircular cracks, which can be seen along the anterior edge of the butt with which they are in contact (Fig 8b).

In addition to cracks, abrasion of the edge of the striking/pressure platform can also lead to the appearance of polishes (Fig 24), the development of which depends on how intensive the abrasion is. In our experimental collection, only 3 blades knapped by direct percussion with antler, and especially 9 blades knapped by pressure with copper do not show any polish on the anterior edge of the butt, which indicates that the abrasion remained superficial in these cases.

**4.1.5. Post-fracture traces.**

**4.1.5.1. *Spontaneous scars on the edges*:** During the creation of our experimental collection, the detachment of blades often led to the appearance of scars on their lateral edges. These scars correspond to the "spontaneous retouch"

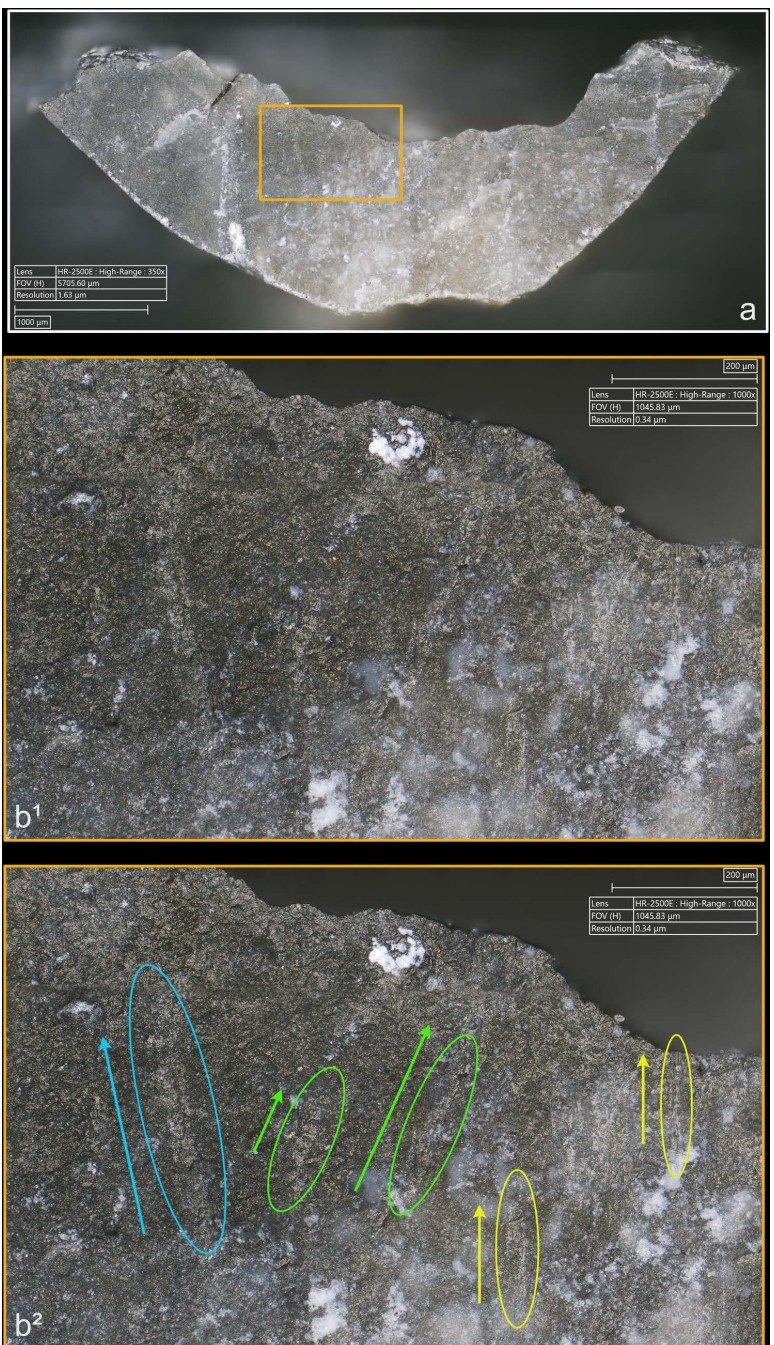

**Fig 23. Polishes associated with the use of pressure and a copper CT.** The different orientations of the linear polishes highlighted in image b² are due to several detachment attempts; the space between the linear polishes circled in green is almost devoid of polishes, probably due to the irregular topography of the CT.

described by M.H. Newcomer [168] (see also [152]: 25]). As we mentioned above, it would be more accurate to refer to them as "spontaneous *scars*" (after [169]: 74]), as their formation is accidental, unlike a retouch. These traces are interesting because their appearance, as well as some of their characteristics, are the result of the conditions in which the core was held and therefore make it possible to reconstruct these conditions.

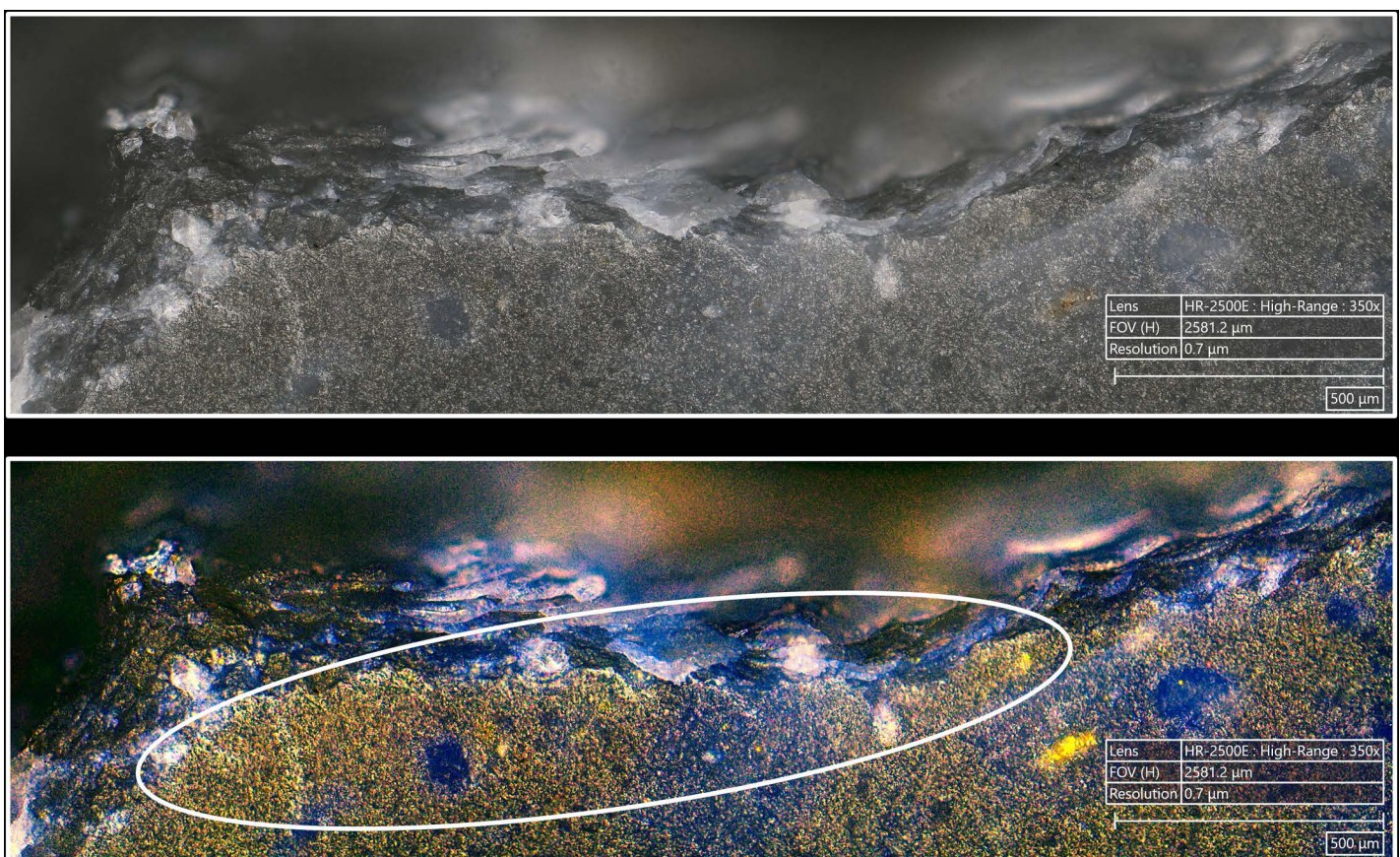

**Fig 24. Polishes located along the anterior edge of the platform of a blade knapped by direct percussion with a bone CT.** These polishes probably result from the abrasion of the edge of the striking platform with a mineral CT that preceded the detachment of the blade.

In direct percussion, the presence of spontaneous scars (Fig 25a, 25b) along the edges is common, although not systematic: it concerns most blades extracted with antler (n = 8/10), boxwood (n = 8/11), quartzite (n = 8/11) and bone (n = 9/13). The frequency of this type of trace is lower for blades knapped with sandstone (n = 5/11). While knapping by direct percussion, the knapper generally took care to hold the core in such a way that, once detached, the blade remained held against it rather than falling to the ground (with the risk of it breaking). The video recording of the knapping process shows that the blades nevertheless move far enough away from the core that, blocked by the knapper's hand, they moved back towards the core and undergo a counter-shock that causes small chips to detach and the scars to form. This explains why most spontaneous scars associated with direct percussion are located on the dorsal face of the blades (n = 33/56), rather than on the ventral face (3 cases), or on both faces (2 cases; scars are absent in 18 cases). The way in which the core and the blade are held also influences the position and lateralization of the scars, but this highly depends on the morphology of the core and how the knapper adapts to it. This is why the distribution of scars is not homogeneous from one series of blades to another, and often also within the same series, even if cases of scars located exclusively on the proximal part are rare in general (n = 3/56; Fig 25a). As for the characteristics of the scars, they are highly variable, making it difficult to detect any patterns, especially in terms of pattern of their arrangement, their morphologies or their terminations. However, it can be noted that their initiations are most often deep (20 cases) and mixed – i.e., they correspond neither entirely to cone initiations, nor entirely to bending initiations. The denticles are also often intact (26 cases), which

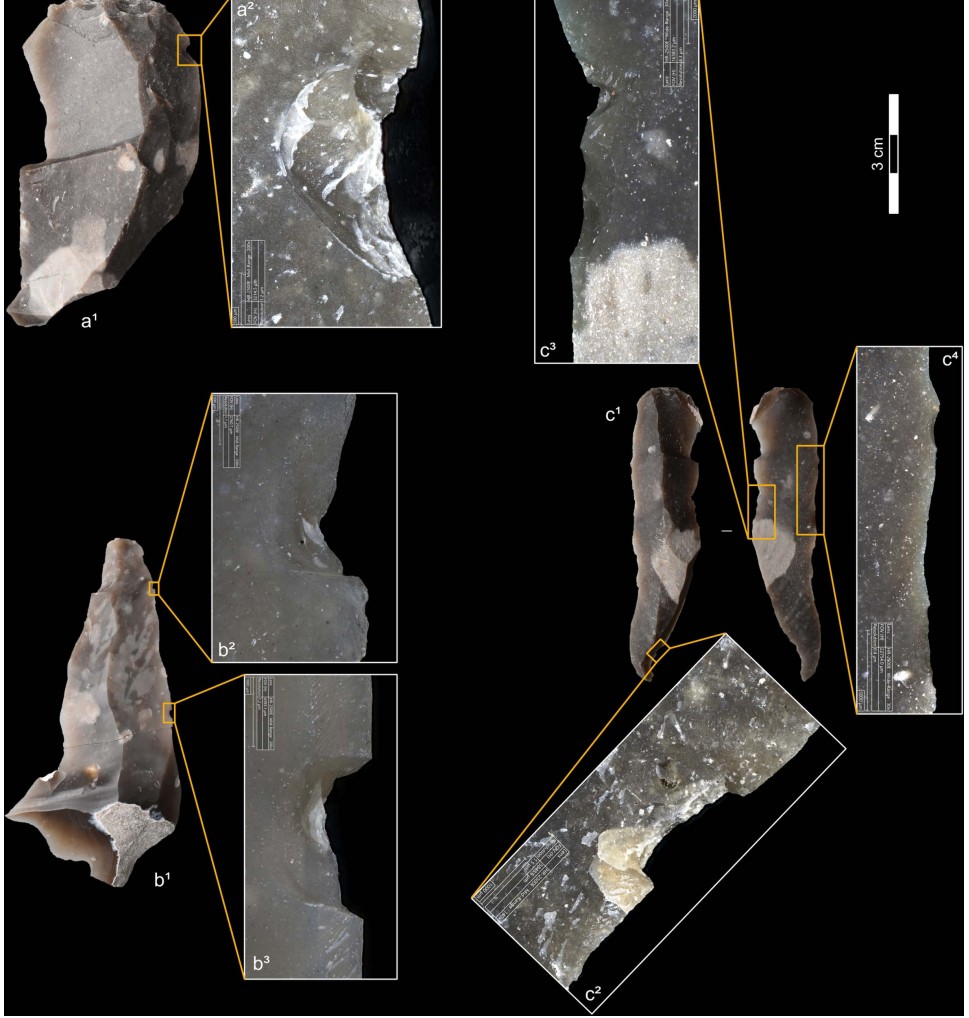

**Fig 25. Scars associated with the use of direct percussion and pressure.** a-b: direct percussion with quartzite; c: pressure with copper. Scars associated with direct percussion are located on the dorsal face of the blades only, while scars associated with pressure are located on both faces.

is consistent with a phenomenon of counter-shock without prolonged contact between the blade and the core. Sometimes, however, all or part of the denticles show signs of crushing. In this situation, which mainly concerns blades extracted with antler (n = 6/10), the contact was longer and may have involved some friction.

Spontaneous scarring is extremely frequent with indirect percussion (Fig 26), and only one of the four elongated flakes detached with bone does not show any. Their formation mechanism is similar to that of direct percussion, in that it originates in a counter-shock between the blade and the core. However, unlike direct percussion in which the knapper held the hammer in one hand and the core in the other (unless the latter was too heavy, in which case it was maintained against the thigh with the hand that remained free), with indirect percussion the knapper had to hold a tool in each hand (the hammer in one and the punch in the other). The knapper therefore held the core between his thighs, with the flaking surface placed against one of them so that blades could not fall to the ground once detached. As a result, the main characteristics of the scars are quite logically similar to those of the scars associated with direct percussion. In indirect percussion, the scars are found on the dorsal face (n = 23/26, compared with 1 blades with ventral scars and 1 blade with scars

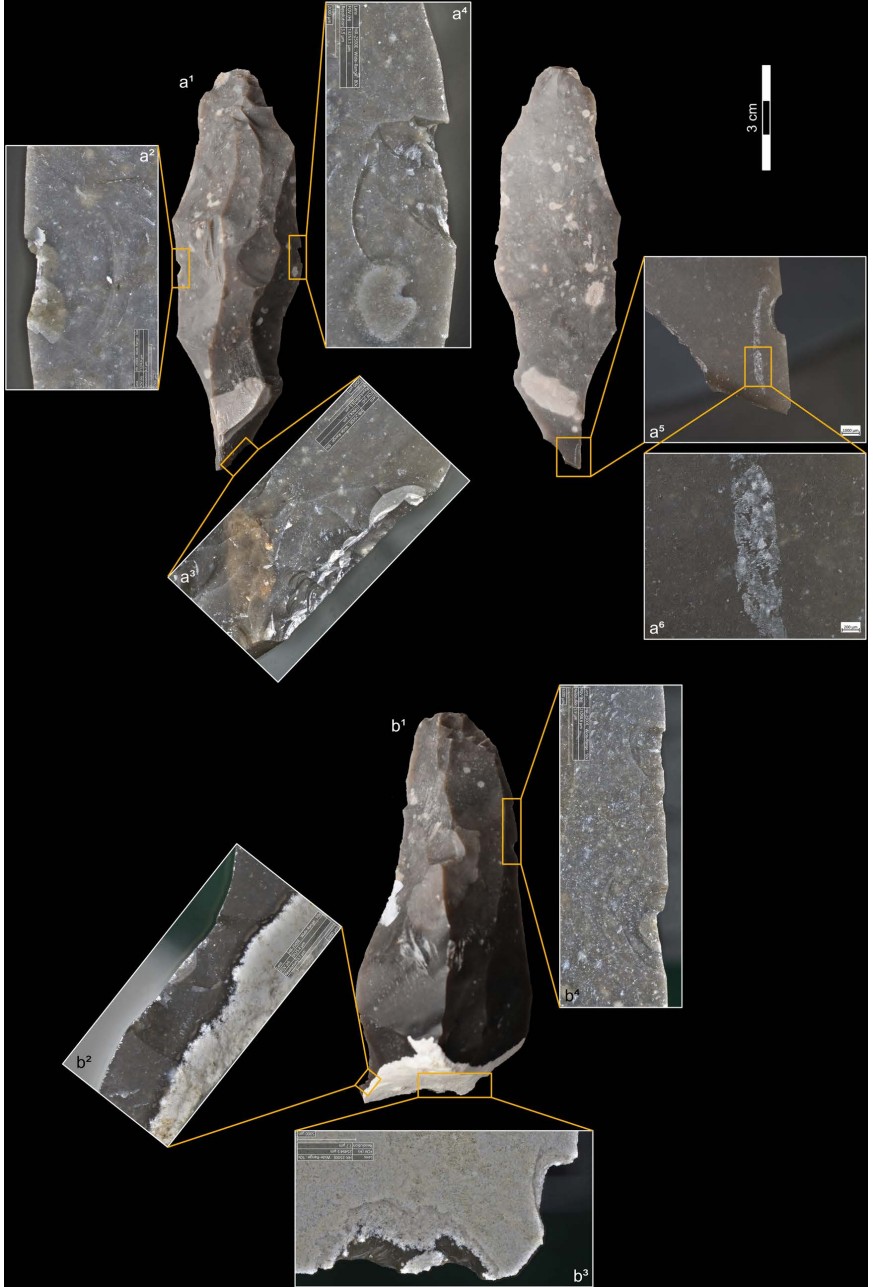

**Fig 26. Scars associated with the use of indirect percussion.** a: antler; b: boxwood. Most of the scars have intact denticles resulting from a brief counter-shock of the blades against their respective cores. However, the blade knapped with an antler CT shows scars at the distal end of its left edge, whose denticles have been crushed by prolonged contact and friction against the core (image a³) due to the way the core was held by the knapper. Images a⁵ and a⁶ provide further evidence of this friction, as they show a linear mineral (flint) residue located at the distal end of the blade's ventral face.

located on both faces), and they are mostly located in the mesio-distal or the distal parts (n = 19/26; Fig 26), sometimes in the mesial part (n = 4/26) and only exceptionally in the proximal part (1 blade) or along the entire length of the blade (1 blade). Initiations are mainly mixed and deep (n = 15/26) or of varying depth (n = 8/26). The condition of the denticles varies

according to the CT used: they are generally intact with boxwood (n = 10/11), but more often partially (n = 7/11) or totally (n = 3/11) crushed with antler. This difference results from the varying way in which the cores were held. In the case of antler, the knapper held the base of the flaking surface against his thigh, while tilting the core so that the striking platform was positioned at around 30° to the ground. As a result, once extracted, the proximo-mesial part of the blade was free of any constraint and tilted forward while the distal part remained blocked by the knapper's thigh and was therefore compressed against the core. This compression eventually causes the proximo-mesial part of the blade to return towards the core, resulting in a counter-shock. Because of this mechanism, the denticles of the proximo-mesial scars are generally intact (counter-shock without prolonged contact), while the denticles of the distal scars are crushed (prolonged compression and friction against the core; Fig 26a; S2 Fig, S1 Video). When he knapped with the boxwood punch, the knapper chose on the other hand to maintain the entire flaking surface against his thigh, so that the striking platform was practically horizontal to the ground. Once extracted, the blade was therefore blocked along its entire length, which prevented its proximo-mesial part from tilting forward. The distal part of the blade is therefore not subject to any significant compression and friction, and the denticles of the scars remain intact (Fig 26b; S3 Fig, S2 Video). This variation in core support also explains why distal or mesio-distal scars are more frequent with antler (n = 10/11) than with boxwood (n = 6/11), and why some of these scars are often located at the distal end of the blade with the former CT (n = 6/11) and rarely with the latter CT (n = 1/11). It is possible that the support of the core also explains why blades knapped with boxwood sometimes only have one scar (n = 4/11), whereas those knapped with antler always have several. We can therefore hypothesise that the limited amplitude of the post-fracture movement of blades knapped with boxwood results in less mechanical stress during counter-shock and that, as a result, the number of detached chips is more limited. Finally, the morphology of the scars is always variable, and no trend could be identified, either with regard to the CTs used, or for indirect percussion in general.

Pressure knapping systematically leads to the formation of spontaneous scars (Fig 25c), and these are generally present in different locations along the edges of the blade (n = 35/38, compared with 3 blades with scars in a single location). The position of the scars differs significantly from that observed in direct percussion and indirect percussion because the blades, unlike with the latter two MFAs, were not held against the core once extracted. They could therefore fall freely and hit various objects, including the core holding device and the core itself, before hitting the ground. It is therefore important to underline that the frequency and/or the position of the scars would have been altered if we had placed something on the ground to cushion the impact of the blades, or if we had used a system to hold them against the core immediately after their detachment.

In our reference collection, the post-fracture shocks suffered by a blade extracted with pressure can occur in different directions, which explains why spontaneous scars most often occur on both faces (n = 30/38; Fig 25c) and not just on the dorsal face (only 1 blade; the other 7 blades have scars only on the ventral face). Scars also normally affect both edges (n = 31/38), rather than one (n = 5/38 for the right edge and 2/38 for the left edge). We note that the scars are mainly located in the proximo-mesial (n = 21/38), or only the proximal (n = 6/38) part. This suggests that the blades probably tend to tilt forward once extracted, with their proximal half experiencing most of the impacts against other objects. On the other hand, it is uncommon for the scars to be situated in the mesial part (n = 6/38) or to be distributed along the entire length of the blade (n = 4/38), and exceptional that they appear in the mesio-distal part (1 blade: Fig 25c; no cases of scars exclusively located in the distal part were observed). Moreover, due to the absence of compression of the distal part of the blade (contrary to what was observed with indirect percussion with antler), even when scars are present distally, none of them are located at the distal end of the blade.

Apart from a few exceptions, the initiations of the scars are intermediary between proper cone and bending initiations (= mixed initiations), and their depths are more or less pronounced, which is probably the result of the specificities of each shock suffered by the blades after detachment. The variable nature of these shocks (in terms of angle of incidence and kinetic energy for instance) also explains why the same blades have scars with both feather and hinge, or even step terminations (n = 27/38), and less often only feather terminations (n = 11/38). Nevertheless, these shocks do not normally

involve prolonged contact between the blade and the objects hit, so that the denticles are usually intact (n = 32/38), with cases of crushed or partially crushed denticles being limited (n = 6/38). As with direct and indirect percussion, the morphology of the scars is always variable, and we did not notice any particular pattern in this respect with regard to the different CTs used.

**4.1.5.2. *Spontaneous scars on the posterior edge of the platform***: Spontaneous scars occur mainly on the edges of the blades, but they can also form on the butt from its posterior edge, certainly as a result (in some cases at least) of this edge hitting the core during a counter-shock (Fig 27). Spontaneous scars located on the posterior edge are common with pressure (n = 11/38), but rare with direct percussion (n = 2/56) and indirect percussion (n = 1/26). In the case of pressure, such scars are found on most of the blades knapped with antler (n = 7/11), but they are less present on blades knapped with boxwood (n = 4/14). No cases were found for blades detached with copper. We can therefore assume that, in one way or another, antler pressure is correlated with frequent counter-shocks of the posterior edge of the butt in our experimental collection, although the nature of the CT used probably has no direct influence on this.

The initiations of the scars are generally shallow (n = 10/14, all MFAs and CTs combined), and their denticles are totally or partially crushed (n = 8/14), which could indicate counter-shocks followed by friction against the core. The denticles may also be intact (n = 4/14; the 2 other cases are undetermined), in which case it is more likely that the blades have been subjected to a counter-shock without subsequent friction; this has only been observed with pressure. The terminations of the scars vary from one blade to another, and no pattern was detected (n = 5/14 for feather terminations, 4/14 for step terminations, 3/14 for hinge terminations, and 2/14 for variable terminations).

If several spontaneous scars have been observed on the posterior edge of the butt, its anterior edge may also have this type of traces. However, in our experimental collection, only one blade has such a feature. It is interesting to note that this blade was knapped by pressure with copper and that this scar was probably detached because of the blade tilting forward and hitting an object, something that would be difficult to achieve with direct or indirect percussion, given how the knapper held the core when using these two MFAs.

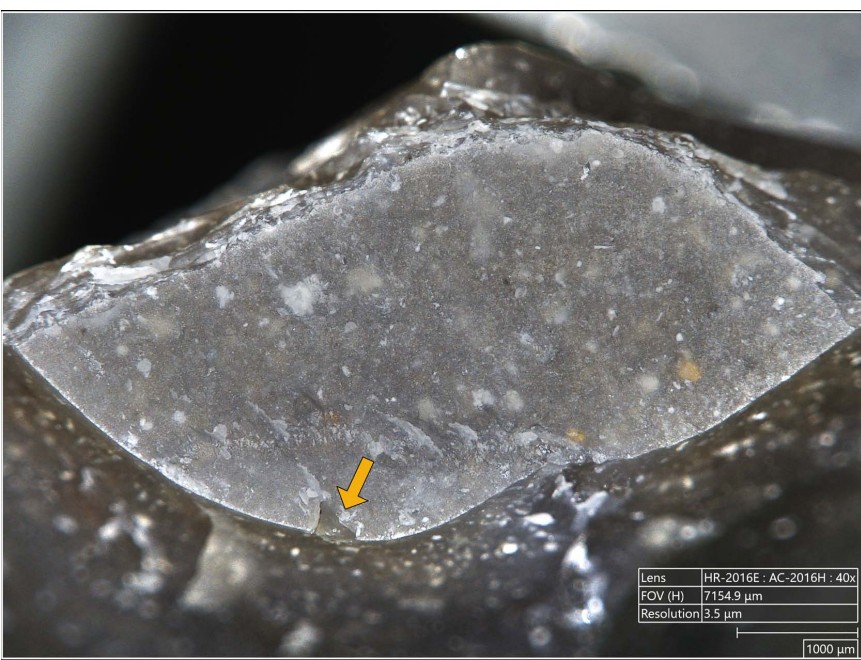

**Fig 27. Spontaneous scar on the posterior edge of a blade knapped by pressure with an antler CT.**

4.1.5.3. *Residues on the ventral face*: The counter-shocks that cause the formation of spontaneous scars can also lead to a deposit of residue on the ventral face of the blades (Fig 26A[5-6]). This phenomenon, rare with pressure (n = 2/38) and direct percussion (n = 1/56), was observed on several occasions on blades knapped by indirect percussion (n = 7/26), with no significant difference with regards to the CT (n = 4/11 for boxwood and 3/11 for antler), even if the four elongated flakes detached with bone do not have it. These residues are generally mineral (n = 8/10), confirming that they are the result of a contact between the core and the blade. The few cases of organic residues (2 cases) are linked to the use of indirect percussion and pressure and can be explained by an accidental contact of the ventral face and the CT, or with the wood device that held the core in the case of pressure knapping.

All MFAs and CTs taken together, these residues are mainly located on the proximal or proximo-mesial part of the blades (n = 5/10 and 1/10 respectively), but they may also appear on the mesial (2 cases) or the distal part (2 cases; Fig 26A[5-6]) with indirect percussion. The general shape of these residue deposits can be non-linear (n = 6/10), which is notably the case for organic residues, or linear (n = 4/10) which clearly indicates frictional movement of the blade against the core.

## 4.2. Identifying knapping techniques using hierarchical cluster analysis

Hierarchical cluster analysis was used to assess whether particular associations of micro- and macroscopic knapping traces, each of which is already known to have a link with the MFA and/or the CT, can help identify these two parameters of the knapping techniques used to produce the experimental blades and, if so, how effectively.

### 4.2.1. Identification of the MFA.
By grouping the blades into two clusters, a first distinction can be made between those extracted by percussion and those extracted by pressure. Cluster 1 contains 97.4% of the former (including 98.2% of the blades detached by direct percussion and 95.5% of the blades detached by indirect percussion) and 7.9% of the latter, while cluster 2 contains 92.10% of the latter and 2.6% of the former (Figs 28 and 29; Table 4; S4 Fig). A Fisher's exact test (p-value < 0.001), as well as a chi-square test (value = 94.4; df = 2; p-value < 0.001) and a Cramér'V (value = 0.902) confirm that the MFA parameter significantly determines the clustering. These results suggest that the transmission of kinetic energy by means of an impact, or a continuous pressure is a key element in the identification of MFAs. In the case of indirect percussion, even if the impact does not directly happen on the core, its consequences in

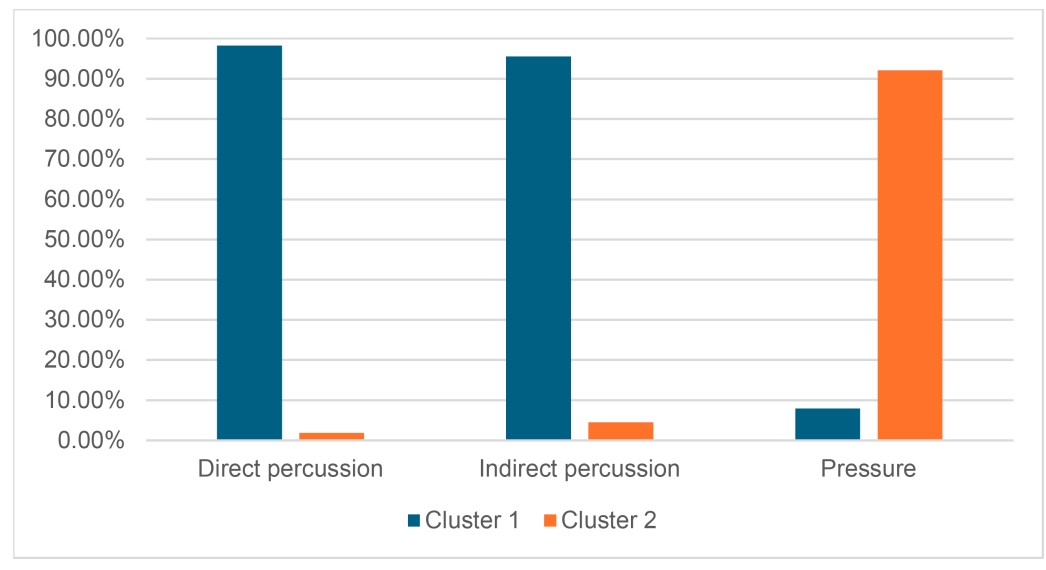

**Fig 28. Clustering of MFAs based on two clusters (in %).**

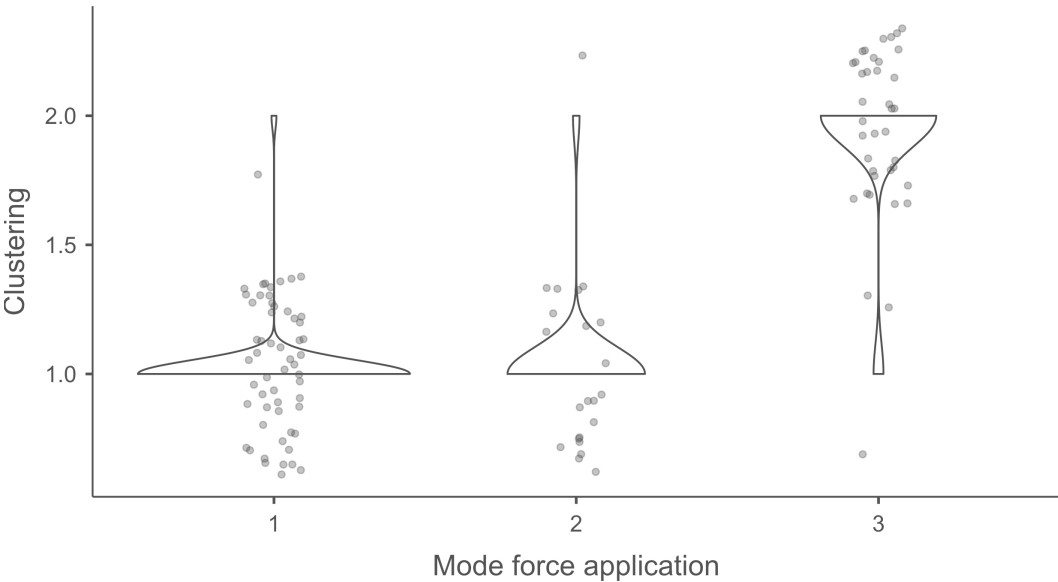

**Fig 29. Plot showing the clustering of MFAs based on two clusters (MFA 1: direct percussion; MFA 2: indirect percussion; MFA 3: pressure; each dot represents a blade).**

**Table 4. Results of the hierarchical cluster analysis applied to the identification of MFAs, regardless of the CTs used, based on two clusters.**

| MFAs | | Cluster 1 | Cluster 2 | Total |
|---|---|---|---|---|
| Direct percussion | Observed | 55 | 1 | 56 |
| | Expected | 38.1 | 17.86 | 56.0 |
| | % within row | 98.2% | 1.8% | 100.0% |
| Indirect percussion | Observed | 21 | 1 | 22 |
| | Expected | 15.0 | 7.02 | 22.0 |
| | % within row | 95.5% | 4.5% | 100.0% |
| Pressure | Observed | 3 | 35 | 38 |
| | Expected | 25.9 | 12.12 | 38.0 |
| | % within row | 7.9% | 92.1% | 100.0% |
| Total | Observed | 79 | 37 | 116 |
| | Expected | 79.0 | 37.00 | 116.0 |
| | % within row | 68.1% | 31.9% | 100.0% |

terms of knapping traces are probably similar to those of direct percussion, at least to a certain extent. In any case, these similarities have more influence on the development of these traces than the elements that differ between these two MFAs, including the precision of the gesture (much greater with indirect percussion because of the use of the punch) and maybe the amount of kinetic energy transmitted to the core (a punch probably does not transfer all the energy generated by the impact, unlike a hammer used in direct percussion).

Grouping the blades into three clusters results in the separation of 30.4% of the blades knapped by direct percussion, which are grouped in cluster 1, in which no blades knapped by indirect percussion or pressure appear. It is interesting to note that the blades in cluster 1 were extracted with all the CTs used, although those associated with sandstone are

better represented (6 blades detached with sandstone, 3 with quartzite, bone and boxwood, and 2 with antler; Figs 30–32; Tables 5 and 6; S5 Fig). Almost all the other blades knapped by direct percussion (67.9%) are associated with blades knapped by indirect percussion within cluster 2. A Fisher's exact test (p-value < 0.001; the conditions required for the use of a chi-square test were not met), confirms that the MFA parameter is linked with the clustering.

When the blades are grouped into 8 clusters, a split appears among the blades knapped by indirect percussion: 36.4% of them appear in cluster 4 (including 5 associated with boxwood and 3 with antler), with only one blade knapped by direct

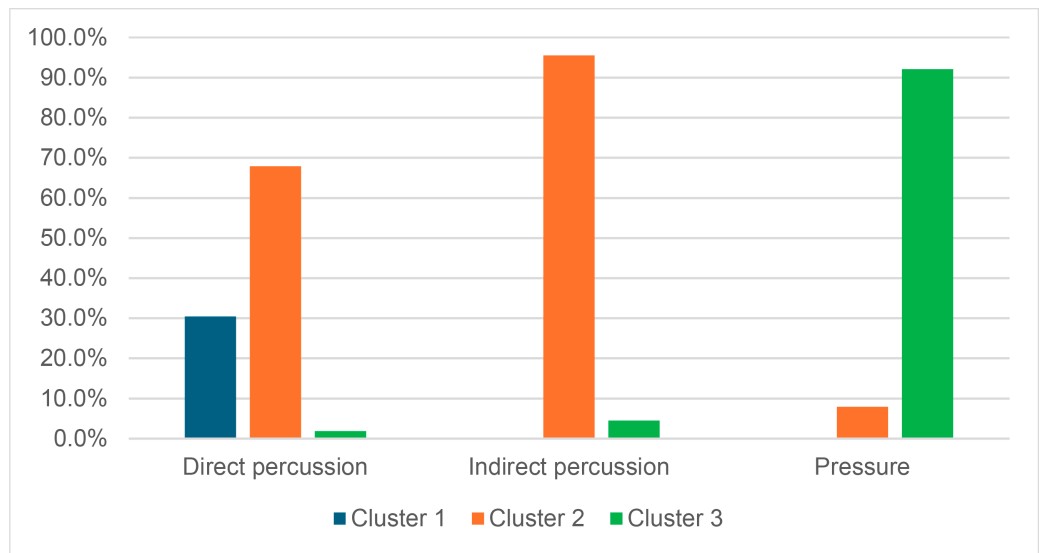

**Fig 30. Clustering of MFAs based on three clusters (in %).**

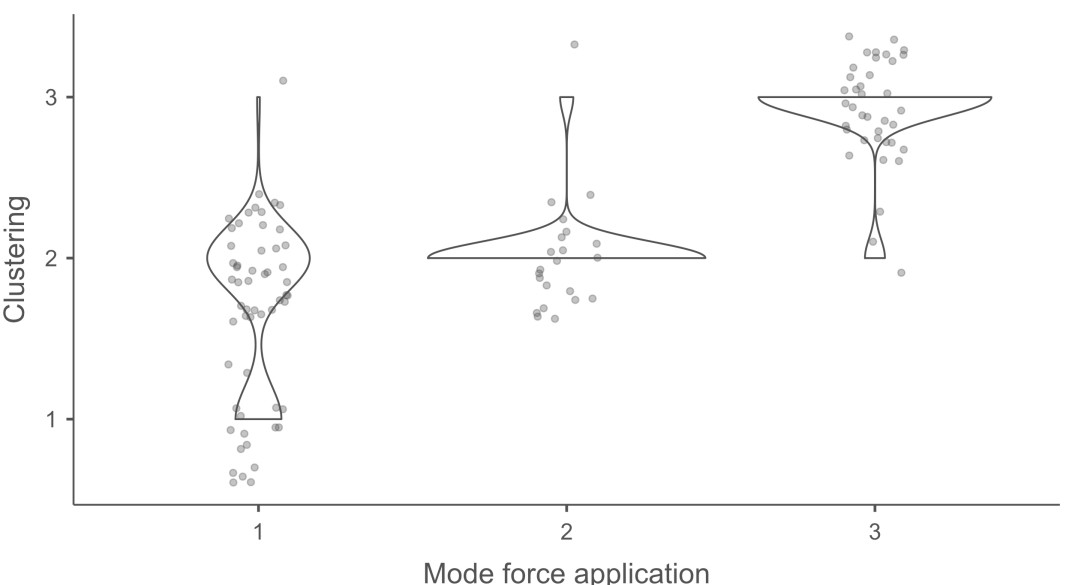

**Fig 31. Plot showing the clustering of MFAs based on three clusters (MFA 1: direct percussion; MFA 2: indirect percussion; MFA 3: pressure; each dot represents a blade).**

percussion and one knapped by pressure (Figs 33–35; Tables 7 and 8; S6 Fig; a Fisher's exact test was performed to verify the link between the MFA parameter and the clustering results but failed to provide any result). Nevertheless, most of the blades extracted by direct and indirect percussion remain grouped in the same clusters, mainly in the cluster 2 (26.8% of the blades knapped by direct percussion and 45.5% of the blades knapped by indirect percussion) and 3 (37.5% of the blades knapped by direct percussion and 9.1% of the blades knapped by indirect percussion). These results suggest that it is difficult to dissociate blades knapped by indirect percussion from blades knapped by direct percussion, and that these two MFAs are therefore difficult to distinguish.

**4.2.2. Identification of the CT used with direct percussion.** When the MFA is direct percussion, a grouping into two clusters is not informative because all the blades except one knapped with boxwood are grouped together. A grouping into three clusters provides more information (Figs 36 and 37; Table 9; S7 Fig). In this situation, most blades knapped with sandstone (90.9%), antler (90%), and to a lesser extent boxwood (54.5%) are grouped in cluster 1, while most blades

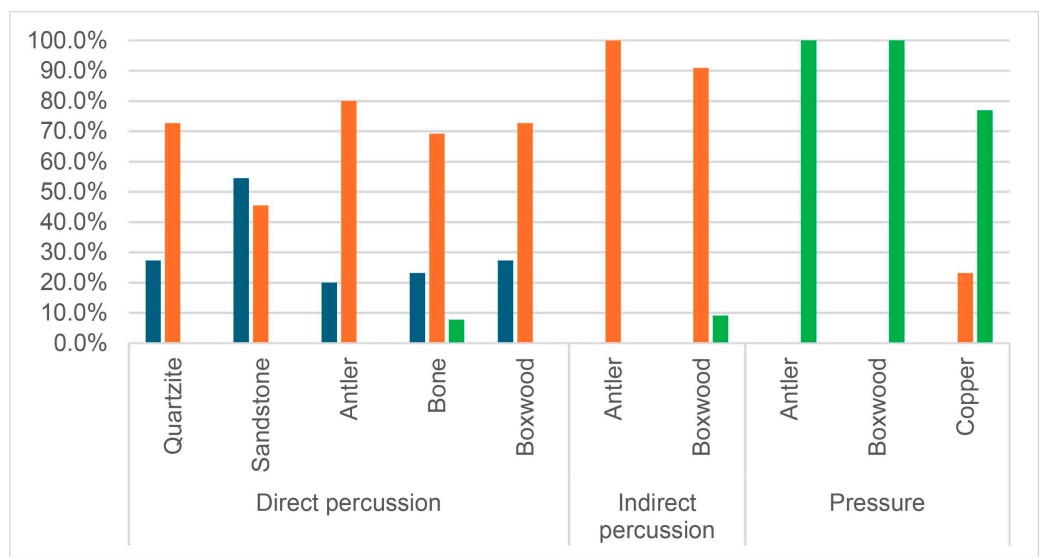

**Fig 32. Clustering of MFAs according to the CTs used based on three clusters (in %).**

**Table 5. Results of the hierarchical cluster analysis applied to the identification of MFAs, regardless of the CTs used, based on three clusters.**

| MFAs | | Cluster 1 | Cluster 2 | Cluster 3 | Total |
|---|---|---|---|---|---|
| Direct percussion | Observed | 17 | 38 | 1 | 56 |
| | Expected | 8.21 | 29.9 | 17.86 | 56.0 |
| | % within row | 30.4% | 67.9% | 1.8% | 100.0% |
| Indirect percussion | Observed | 0 | 21 | 1 | 22 |
| | Expected | 3.22 | 11.8 | 7.02 | 22.0 |
| | % within row | 0.0% | 95.5% | 4.5% | 100.0% |
| Pressure | Observed | 0 | 3 | 35 | 38 |
| | Expected | 5.57 | 20.3 | 12.12 | 38.0 |
| | % within row | 0.0% | 7.9% | 92.1% | 100.0% |
| Total | Observed | 17 | 62 | 37 | 116 |
| | Expected | 17.00 | 62.0 | 37.00 | 116.0 |
| | % within row | 14.7% | 53.4% | 31.9% | 100.0% |

**Table 6. Results of the hierarchical cluster analysis applied to the identification of MFAs, depending on the CTs used, based on three clusters.**

| MFAs | CTs | | Cluster 1 | Cluster 2 | Cluster 3 | Total |
|---|---|---|---|---|---|---|
| Direct percussion | Quartzite | Observed | 3 | 8 | 0 | 11 |
| | | Expected | 3.34 | 7.464 | 0.196 | 11.0 |
| | | % within row | 27.3% | 72.7% | 0.0% | 100.0% |
| | Sandstone | Observed | 6 | 5 | 0 | 11 |
| | | Expected | 3.34 | 7.464 | 0.196 | 11.0 |
| | | % within row | 54.5% | 45.5% | 0.0% | 100.0% |
| | Antler | Observed | 2 | 8 | 0 | 10 |
| | | Expected | 3.04 | 6.786 | 0.179 | 10.0 |
| | | % within row | 20.0% | 80.0% | 0.0% | 100.0% |
| | Bone | Observed | 3 | 9 | 1 | 13 |
| | | Expected | 3.95 | 8.821 | 0.232 | 13.0 |
| | | % within row | 23.1% | 69.2% | 7.7% | 100.0% |
| | Boxwood | Observed | 3 | 8 | 0 | 11 |
| | | Expected | 3.34 | 7.464 | 0.196 | 11.0 |
| | | % within row | 27.3% | 72.7% | 0.0% | 100.0% |
| Indirect percussion | Antler | Observed | 0 | 11 | 0 | 11 |
| | | Expected | 0.00 | 10.500 | 0.500 | 11.0 |
| | | % within row | 0.0% | 100.0% | 0.0% | 100.0% |
| | Boxwood | Observed | 0 | 10 | 1 | 11 |
| | | Expected | 0.00 | 10.500 | 0.500 | 11.0 |
| | | % within row | 0.0% | 90.9% | 9.1% | 100.0% |
| Pressure | Antler | Observed | 0 | 0 | 11 | 11 |
| | | Expected | 0.00 | 0.868 | 10.132 | 11.0 |
| | | % within row | 0.0% | 0.0% | 100.0% | 100.0% |
| | Boxwood | Observed | 0 | 0 | 14 | 14 |
| | | Expected | 0.00 | 1.105 | 12.895 | 14.0 |
| | | % within row | 0.0% | 0.0% | 100.0% | 100.0% |
| | Copper | Observed | 0 | 3 | 10 | 13 |
| | | Expected | 0.00 | 1.026 | 11.974 | 13.0 |
| | | % within row | 0.0% | 23.1% | 76.9% | 100.0% |
| Total | | Observed | 17 | 62 | 37 | 116 |
| | | Expected | 17.00 | 62.000 | 37.000 | 116.0 |
| | | % within row | 14.7% | 53.4% | 31.9% | 100.0% |

knapped with bone (92.3%) and quartzite (63.6%) appear in cluster 2. We can therefore see that most of the "soft" CTs, whether organic or mineral, are better represented in cluster 1, while the hardest CT – quartzite – is mainly represented in cluster 2. However, the high presence of bone and the significant presence of boxwood (36.4%) in cluster 2 make the interpretation of the cluster 2 population somewhat unclear. A Fisher's exact test (p-value < 0.001; the conditions required for the use of a chi-square test were not met) nonetheless confirms that the CT parameter is linked with the clustering.

A grouping in four clusters appears more instructive as it splits the cluster associated with the bone and the quartzite CTs seen in the previous situation (Figs 38 and 39; Table 10; S8 Fig; a Fisher's exact test was performed to assess the link between the CT parameter and the results of the cluster analysis, but it was not able to provide a result). Most of the blades knapped with bone (84.6%) are now grouped in a specific cluster (cluster 2), which also includes 27.3% of the blades knapped with boxwood and only 9.1% of the blades knapped with quartzite, while most of the blades knapped

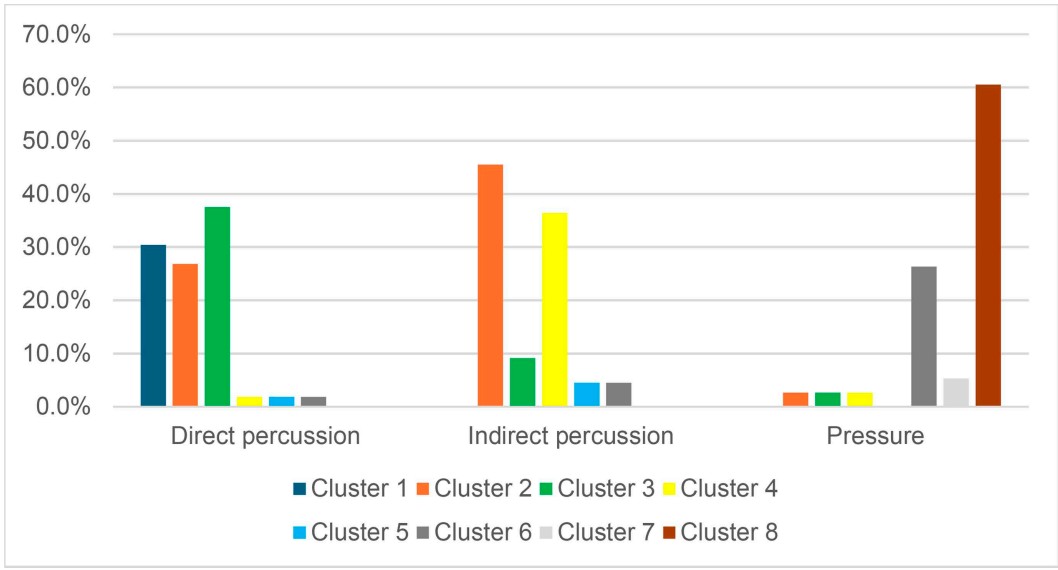

**Fig 33. Clustering of MFAs based on eight clusters (in %).**

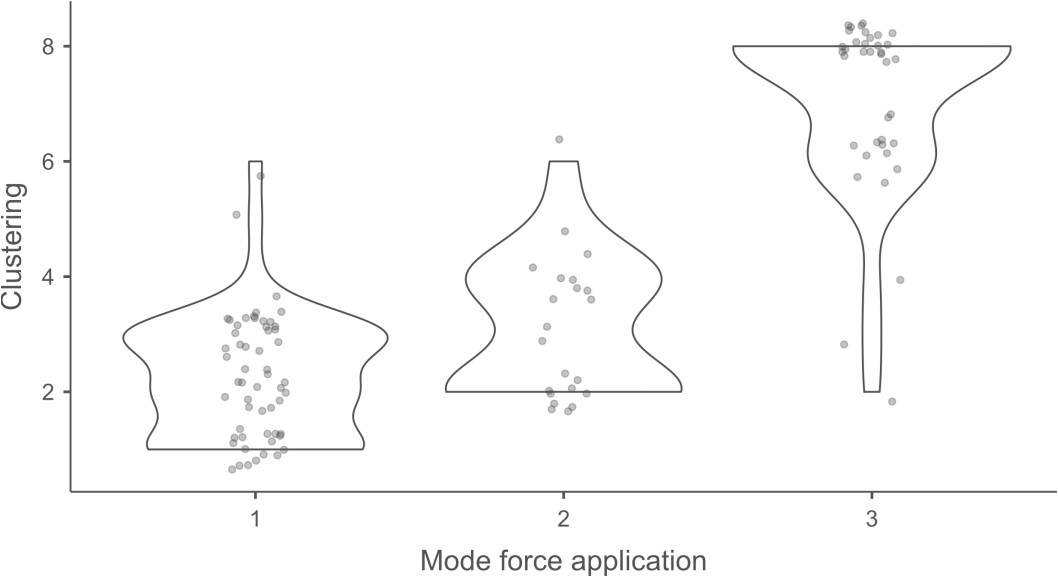

**Fig 34. Plot showing the clustering of MFAs based on eight clusters (MFA 1: direct percussion; MFA 2: indirect percussion; MFA 3: pressure; each dot represents a blade).**

with quartzite (54.5%) appear in another cluster (cluster 3), with a few other blades associated with various CTs (9.1% for sandstone and boxwood, 7.7% for bone). We hypothesize that the distinction between each of these clusters results from knapping traces patterns that are mainly influenced by the physical properties of the CTs, probably their respective hardnesses in particular. If this interpretation is correct, cluster 1 could be considered to indicate the use of a rather soft CT, while cluster 3 would indicate the use of a rather hard CT. Cluster 2, which is dominated by blades knapped with bone,

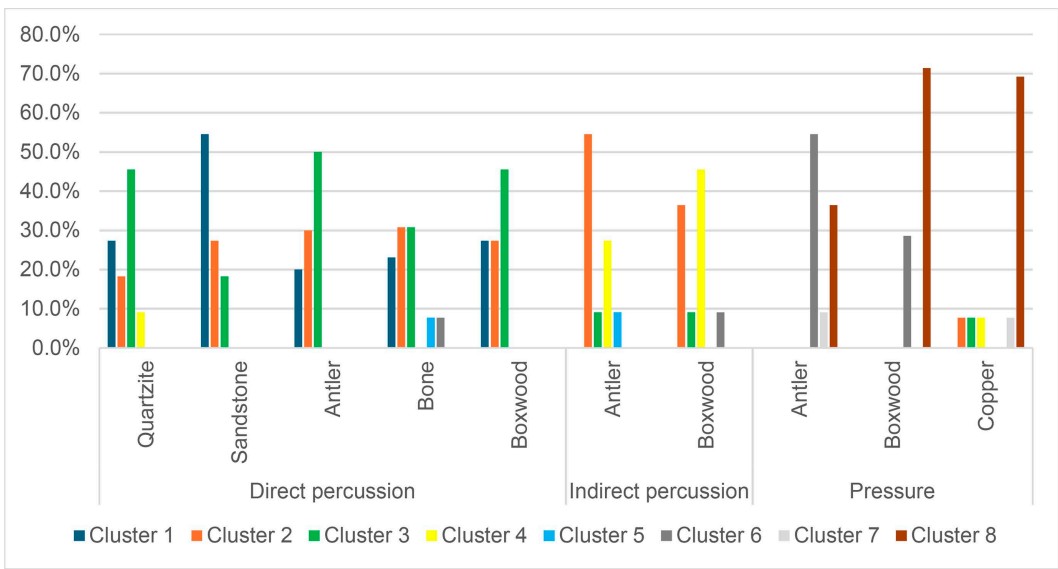

**Fig 35. Clustering of MFAs according to the CTs used based on eight clusters (in %).**

**Table 7. Results of the hierarchical cluster analysis applied to the identification of MFAs, regardless of the CTs used, based on eight clusters.**

| MFAs | | Cluster 1 | Cluster 2 | Cluster 3 | Cluster 4 | Cluster 5 | Cluster 6 | Cluster 7 | Cluster 8 | Total |
|---|---|---|---|---|---|---|---|---|---|---|
| Direct percussion | Observed | 17 | 15 | 21 | 1 | 1 | 1 | 0 | 0 | 56 |
| | Expected | 8.21 | 12.55 | 11.59 | 4.83 | 0.966 | 5.79 | 0.966 | 11.10 | 56.0 |
| | % within row | 30.4% | 26.8% | 37.5% | 1.8% | 1.8% | 1.8% | 0.0% | 0.0% | 100.0% |
| Indirect percussion | Observed | 0 | 10 | 2 | 8 | 1 | 1 | 0 | 0 | 22 |
| | Expected | 3.22 | 4.93 | 4.55 | 1.90 | 0.379 | 2.28 | 0.379 | 4.36 | 22.0 |
| | % within row | 0.0% | 45.5% | 9.1% | 36.4% | 4.5% | 4.5% | 0.0% | 0.0% | 100.0% |
| Pressure | Observed | 0 | 1 | 1 | 1 | 0 | 10 | 2 | 23 | 38 |
| | Expected | 5.57 | 8.52 | 7.86 | 3.28 | 0.655 | 3.93 | 0.655 | 7.53 | 38.0 |
| | % within row | 0.0% | 2.6% | 2.6% | 2.6% | 0.0% | 26.3% | 5.3% | 60.5% | 100.0% |
| Total | Observed | 17 | 26 | 24 | 10 | 2 | 12 | 2 | 23 | 116 |
| | Expected | 17.00 | 26.00 | 24.00 | 10.00 | 2.000 | 12.00 | 2.000 | 23.00 | 116.0 |
| | % within row | 14.7% | 22.4% | 20.7% | 8.6% | 1.7% | 10.3% | 1.7% | 19.8% | 100.0% |

being closer to cluster 3 than to cluster 1 (see dendrogram in S8 Fig), could potentially indicate that the bone CT (a cow fibula, the head of which was used as the percussive part) was somewhat harder than the other soft CTs we used.

**4.2.3. Identification of the CT used with indirect percussion.** Cluster analysis proves quite effective with indirect percussion, as a grouping into two clusters allows to distinguish the two CTs used with a satisfactory degree of precision. Each cluster includes the majority (81.8%) of blades knapped with one of these two CTs, and a few blades (18.2%) knapped with the other CT (Figs 40 and 41; Table 11; S9 Fig). A Fisher's exact test (p-value = 0.009), as well as a chi-square test (value = 8.91; df = 1; p-value = 0.003) and a Cramér'V (value = 0.636) confirm that the CT parameter is linked with the clustering.

**4.2.4. Identification of the CT used with pressure.** In the case of pressure knapping, an analysis based on two clusters gives results that allow to discriminate between the three CTs used with a relative efficiency, probably according

to their degree of hardness (Figs 42 and 43; Table 12; S10 Fig). Blades knapped with the hardest CTs, i.e., copper and antler, are mainly grouped in cluster 1 (84.6% and 63.6% respectively), which also includes 14.3% of blades knapped with boxwood. Conversely, cluster 2 includes 85.7% of the blades detached with boxwood, the softest CT, as well as part of the blades extracted with antler (36.4%) and copper (15.4%). A Fisher's exact test (p-value < 0.001), as well as a chi-square test (value = 14.1; df = 2; p-value < 0.001) and a Cramér'V (value = 0.610) confirm that the CT parameter is linked with the clustering results.

An analysis based on three clusters makes it possible to distinguish most of the blades knapped with copper (n = 9, i.e., 69.2%) which are grouped in cluster 1, in which there are no blades knapped with another CT (Figs 44 and 45; Table 13; S11 Fig). According to the dendrogram, this group is closer to cluster 2, which mainly includes blades knapped with antler, than to cluster 3, which is mainly associated with boxwood. If these clusters are influenced primarily by the degree

**Table 8. Results of the hierarchical cluster analysis applied to the identification of MFAs, depending on the CTs used, based on eight clusters.**

| MFAs | CTs | | Cluster 1 | Cluster 2 | Cluster 3 | Cluster 4 | Cluster 5 | Cluster 6 | Cluster 7 | Cluster 8 | Total |
|---|---|---|---|---|---|---|---|---|---|---|---|
| Direct percussion | Quartzite | Observed | 3 | 2 | 5 | 1 | 0 | 0 | 0 | 0 | 11 |
| | | Expected | 3.34 | 2.946 | 4.125 | 0.196 | 0.196 | 0.196 | 0.000 | 0.00 | 11.0 |
| | | % within row | 27.3% | 18.2% | 45.5% | 9.1% | 0.0% | 0.0% | 0.0% | 0.0% | 100.0% |
| | Sandstone | Observed | 6 | 3 | 2 | 0 | 0 | 0 | 0 | 0 | 11 |
| | | Expected | 3.34 | 2.946 | 4.125 | 0.196 | 0.196 | 0.196 | 0.000 | 0.00 | 11.0 |
| | | % within row | 54.5% | 27.3% | 18.2% | 0.0% | 0.0% | 0.0% | 0.0% | 0.0% | 100.0% |
| | Antler | Observed | 2 | 3 | 5 | 0 | 0 | 0 | 0 | 0 | 10 |
| | | Expected | 3.04 | 2.679 | 3.750 | 0.179 | 0.179 | 0.179 | 0.000 | 0.00 | 10.0 |
| | | % within row | 20.0% | 30.0% | 50.0% | 0.0% | 0.0% | 0.0% | 0.0% | 0.0% | 100.0% |
| | Bone | Observed | 3 | 4 | 4 | 0 | 1 | 1 | 0 | 0 | 13 |
| | | Expected | 3.95 | 3.482 | 4.875 | 0.232 | 0.232 | 0.232 | 0.000 | 0.00 | 13.0 |
| | | % within row | 23.1% | 30.8% | 30.8% | 0.0% | 7.7% | 7.7% | 0.0% | 0.0% | 100.0% |
| | Boxwood | Observed | 3 | 3 | 5 | 0 | 0 | 0 | 0 | 0 | 11 |
| | | Expected | 3.34 | 2.946 | 4.125 | 0.196 | 0.196 | 0.196 | 0.000 | 0.00 | 11.0 |
| | | % within row | 27.3% | 27.3% | 45.5% | 0.0% | 0.0% | 0.0% | 0.0% | 0.0% | 100.0% |
| Indirect percussion | Antler | Observed | 0 | 6 | 1 | 3 | 1 | 0 | 0 | 0 | 11 |
| | | Expected | 0.00 | 5.000 | 1.000 | 4.000 | 0.500 | 0.500 | 0.000 | 0.00 | 11.0 |
| | | % within row | 0.0% | 54.5% | 9.1% | 27.3% | 9.1% | 0.0% | 0.0% | 0.0% | 100.0% |
| | Boxwood | Observed | 0 | 4 | 1 | 5 | 0 | 1 | 0 | 0 | 11 |
| | | Expected | 0.00 | 5.000 | 1.000 | 4.000 | 0.500 | 0.500 | 0.000 | 0.00 | 11.0 |
| | | % within row | 0.0% | 36.4% | 9.1% | 45.5% | 0.0% | 9.1% | 0.0% | 0.0% | 100.0% |
| Pressure | Antler | Observed | 0 | 0 | 0 | 0 | 0 | 6 | 1 | 4 | 11 |
| | | Expected | 0.00 | 0.289 | 0.289 | 0.289 | 0.000 | 2.895 | 0.579 | 6.66 | 11.0 |
| | | % within row | 0.0% | 0.0% | 0.0% | 0.0% | 0.0% | 54.5% | 9.1% | 36.4% | 100.0% |
| | Boxwood | Observed | 0 | 0 | 0 | 0 | 0 | 4 | 0 | 10 | 14 |
| | | Expected | 0.00 | 0.368 | 0.368 | 0.368 | 0.000 | 3.684 | 0.737 | 8.47 | 14.0 |
| | | % within row | 0.0% | 0.0% | 0.0% | 0.0% | 0.0% | 28.6% | 0.0% | 71.4% | 100.0% |
| | Copper | Observed | 0 | 1 | 1 | 1 | 0 | 0 | 1 | 9 | 13 |
| | | Expected | 0.00 | 0.342 | 0.342 | 0.342 | 0.000 | 3.421 | 0.684 | 7.87 | 13.0 |
| | | % within row | 0.0% | 7.7% | 7.7% | 7.7% | 0.0% | 0.0% | 7.7% | 69.2% | 100.0% |
| Total | | Observed | 17 | 26 | 24 | 10 | 2 | 12 | 2 | 23 | 116 |
| | | Expected | 17.00 | 26.000 | 24.000 | 10.000 | 2.000 | 12.000 | 2.000 | 23.00 | 116.0 |
| | | % within row | 14.7% | 22.4% | 20.7% | 8.6% | 1.7% | 10.3% | 1.7% | 19.8% | 100.0% |

of hardness of the CTs used, cluster 1 could result from the fact that copper is the hardest of the three CTs used with pressure. A Fisher's exact test (p-value < 0.001; conditions were not met to perform a chi-square test), shows that the CT parameter is linked with the results of the cluster analysis.

**4.2.5. Identification of the CT without prior identification of the MFA.** A final cluster analysis was carried out to determine whether the nature of the CT could be identified when the MFA had not been determined beforehand.

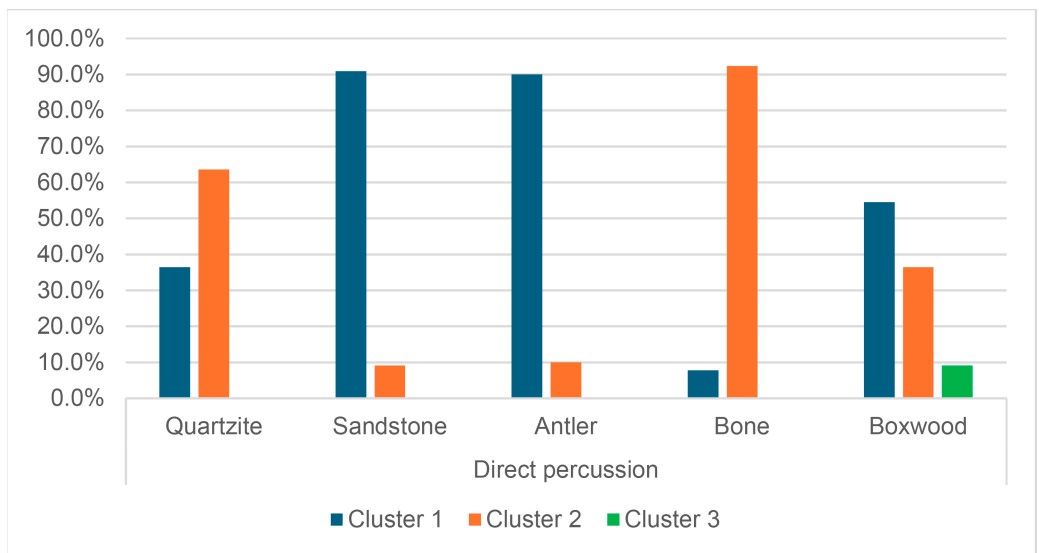

**Fig 36. Clustering of CTs associated with direct percussion based on three clusters (in %).**

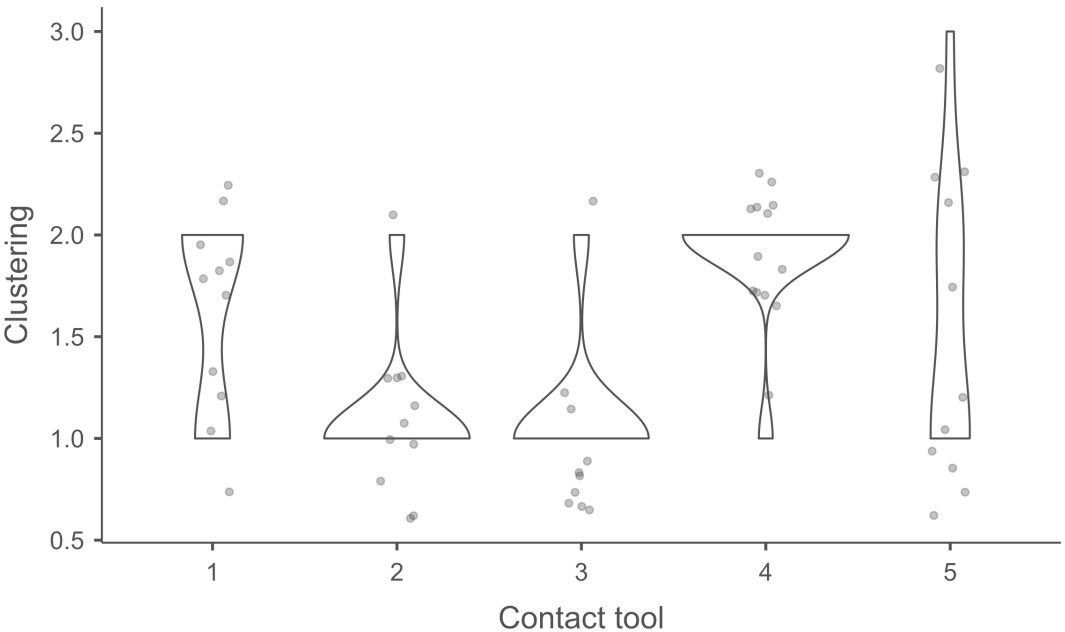

**Fig 37. Plot showing the clustering of CTs associated with direct percussion based on three clusters (CT 1: quartzite; CT 2: sandstone; CT 3: antler; CT 4: bone; CT 5: boxwood; each dot represents a blade).**

A grouping into two clusters associates all blades knapped with quartzite and sandstone, as well as almost all blades knapped with bone (92.3%) in cluster 1 (Figs 46 and 47; Table 14; S12 Fig). Antler and boxwood are also associated with this cluster (71.9% and 61.1% respectively), but unlike the previous three CTs, they are also well represented in cluster 2 (28.1% and 38.9% respectively). Only copper is predominantly represented in cluster 2 (84.6%) and to a lesser extent in cluster 1 (15.4%). The significance of these results is not obvious as far as the CTs are concerned, as copper,

Table 9. Results of the hierarchical cluster analysis applied to the identification of CTs used with direct percussion, based on three clusters.

| CTs | | Cluster 1 | Cluster 2 | Cluster 3 | Total |
|---|---|---|---|---|---|
| Quartzite | Observed | 4 | 7 | 0 | 11 |
| | Expected | 5.89 | 4.91 | 0.196 | 11.0 |
| | % within row | 36.4% | 63.6% | 0.0% | 100.0% |
| Sandstone | Observed | 10 | 1 | 0 | 11 |
| | Expected | 5.89 | 4.91 | 0.196 | 11.0 |
| | % within row | 90.9% | 9.1% | 0.0% | 100.0% |
| Antler | Observed | 9 | 1 | 0 | 10 |
| | Expected | 5.36 | 4.46 | 0.179 | 10.0 |
| | % within row | 90.0% | 10.0% | 0.0% | 100.0% |
| Bone | Observed | 1 | 12 | 0 | 13 |
| | Expected | 6.96 | 5.80 | 0.232 | 13.0 |
| | % within row | 7.7% | 92.3% | 0.0% | 100.0% |
| Boxwood | Observed | 6 | 4 | 1 | 11 |
| | Expected | 5.89 | 4.91 | 0.196 | 11.0 |
| | % within row | 54.5% | 36.4% | 9.1% | 100.0% |
| Total | Observed | 30 | 25 | 1 | 56 |
| | Expected | 30.00 | 25.00 | 1.000 | 56.0 |
| | % within row | 53.6% | 44.6% | 1.8% | 100.0% |

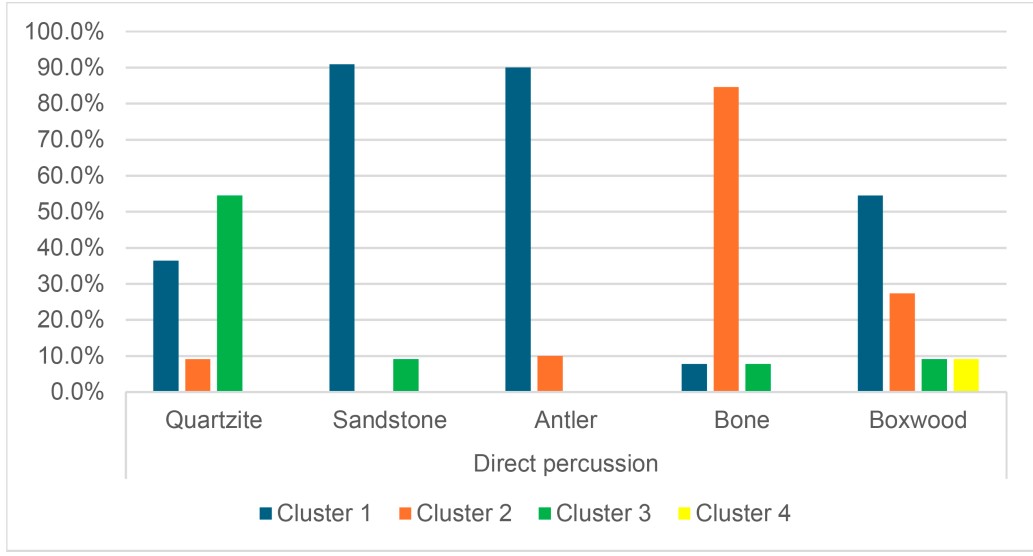

**Fig 38. Clustering of CTs associated with direct percussion based on four clusters (in %).**

for example, would be expected to show a clustering more similar to that of quartzite, sandstone, bone, or even antler, because it is harder than a material such as boxwood. This situation appears to be due to the composition of our reference collection, where most CTs (quartzite, sandstone, bone, and copper) were used in association with only one MFA,

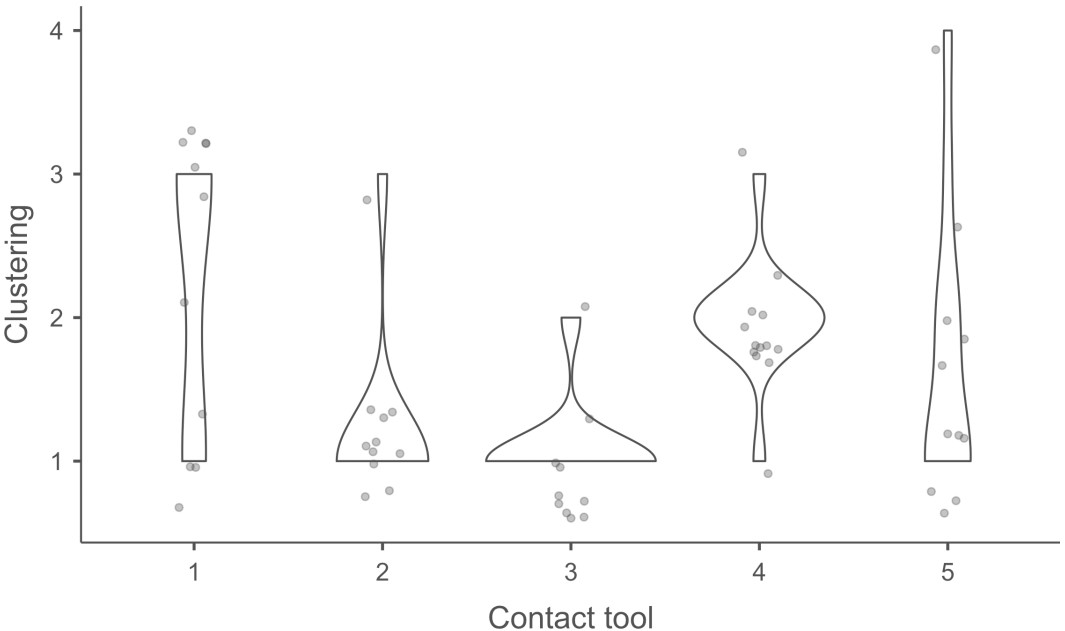

**Fig 39. Plot showing the clustering of CTs associated with direct percussion based on four clusters (CT 1: quartzite; CT 2: sandstone; CT 3: antler; CT 4: bone; CT 5: boxwood; each dot represents a blade).**

**Table 10. Results of the hierarchical cluster analysis applied to the identification of CTs used with direct percussion, based on four clusters.**

| CTs | | Cluster 1 | Cluster 2 | Cluster 3 | Cluster 4 | Total |
|---|---|---|---|---|---|---|
| Quartzite | Observed | 4 | 1 | 6 | 0 | 11 |
| | Expected | 5.89 | 3.14 | 1.77 | 0.196 | 11.0 |
| | % within row | 36.4% | 9.1% | 54.5% | 0.0% | 100.0% |
| Sandstone | Observed | 10 | 0 | 1 | 0 | 11 |
| | Expected | 5.89 | 3.14 | 1.77 | 0.196 | 11.0 |
| | % within row | 90.9% | 0.0% | 9.1% | 0.0% | 100.0% |
| Antler | Observed | 9 | 1 | 0 | 0 | 10 |
| | Expected | 5.36 | 2.86 | 1.61 | 0.179 | 10.0 |
| | % within row | 90.0% | 10.0% | 0.0% | 0.0% | 100.0% |
| Bone | Observed | 1 | 11 | 1 | 0 | 13 |
| | Expected | 6.96 | 3.71 | 2.09 | 0.232 | 13.0 |
| | % within row | 7.7% | 84.6% | 7.7% | 0.0% | 100.0% |
| Boxwood | Observed | 6 | 3 | 1 | 1 | 11 |
| | Expected | 5.89 | 3.14 | 1.77 | 0.196 | 11.0 |
| | % within row | 54.5% | 27.3% | 9.1% | 9.1% | 100.0% |
| Total | Observed | 30 | 16 | 9 | 1 | 56 |
| | Expected | 30.00 | 16.00 | 9.00 | 1.000 | 56.0 |
| | % within row | 53.6% | 28.6% | 16.1% | 1.8% | 100.0% |

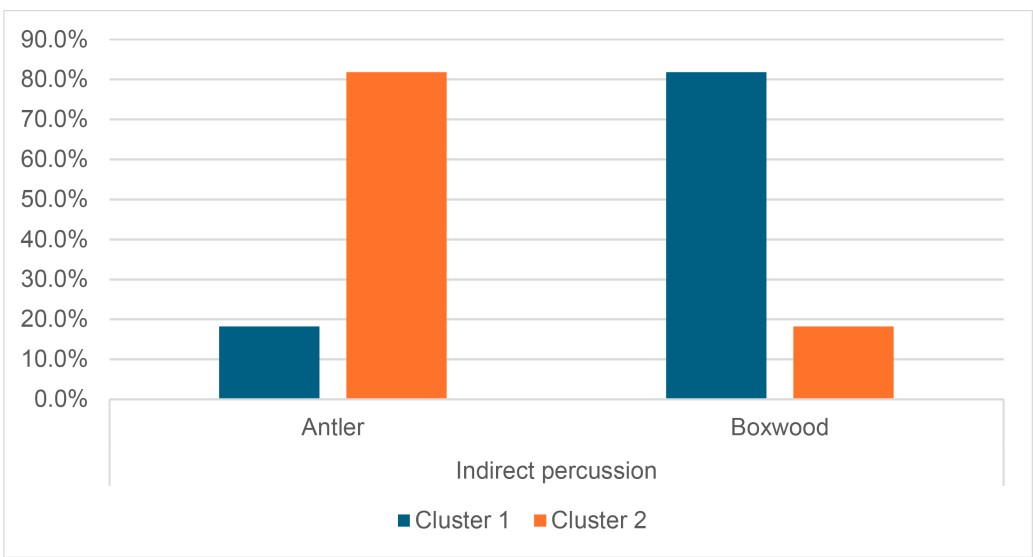

**Fig 40. Clustering of CTs associated with indirect percussion based on two clusters (in %).**

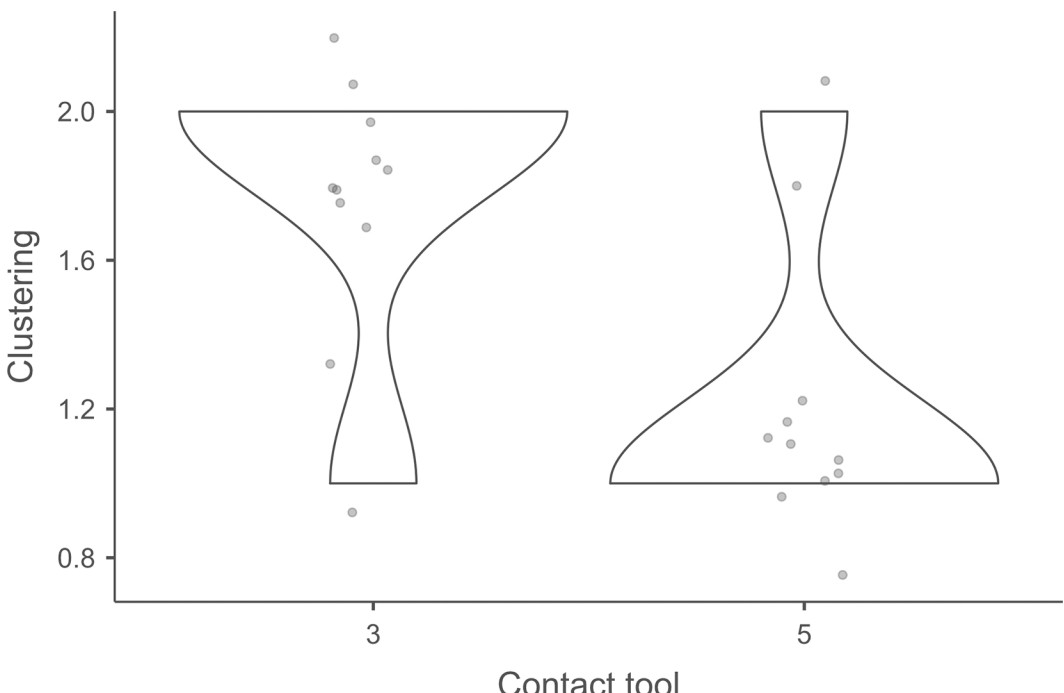

**Fig 41. Plot showing the clustering of CTs associated with indirect percussion based on two clusters (CT 3: antler; CT 5: boxwood; each dot corresponds to a blade).**

whereas only two CTs (antler and boxwood) were used with all MFAs. For this reason, the results of the clustering are mainly driven by the MFA parameter, as confirmed by a Fisher's exact test (p-value < 0.001), as well as by a chi-square test (value = 86.2, df = 2, p-value < 0.001) and a Cramér's V (value = 0.862), rather than by the nature of the CTs itself.

**Table 11. Results of the hierarchical cluster analysis applied to the identification of CTs used with indirect percussion, based on two clusters.**

| CTs | | Cluster 1 | Cluster 2 | Total |
|---|---|---|---|---|
| Antler | Observed | 2 | 9 | 11 |
| | Expected | 5.50 | 5.50 | 11.0 |
| | % within row | 18.2% | 81.8% | 100.0% |
| Boxwood | Observed | 9 | 2 | 11 |
| | Expected | 5.50 | 5.50 | 11.0 |
| | % within row | 81.8% | 18.2% | 100.0% |
| Total | Observed | 11 | 11 | 22 |
| | Expected | 11.00 | 11.00 | 22.0 |
| | % within row | 50.0% | 50.0% | 100.0% |

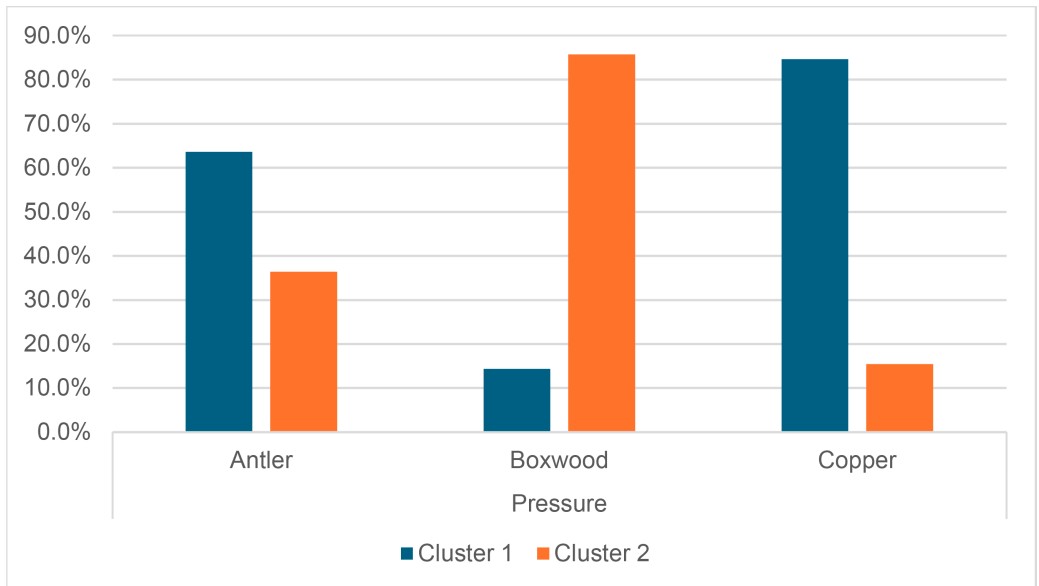

**Fig 42. Clustering of CTs associated with pressure based on two clusters (in %).**

To investigate this question further, a hierarchical cluster analysis was carried out on all the blades knapped with antler and boxwood with the same attributes used in the previous situation (Figs 48 and 49; Table 15; S13 Fig). The results of this analysis show that all blades detached by direct and indirect percussion are grouped in the same cluster (cluster 1), whereas the vast majority of blades knapped by pressure (72.7% in the case of antler and 100% in the case of boxwood) are grouped in the other cluster (cluster 2). A Fisher's exact test (p-value < 0.001), as well as a chi-square test (value = 55.9; df = 2; p-value < 0.001) and a Cramér'V (value = 0.907) confirm that the MFA parameter strongly determines the clustering, unlike the CT parameter (Fisher's exact test: p-value = 0.301; chi-square test: value = 1.49; d = 1; p-value = 0.222).

Although other combinations of MFAs and CTs will need to be added to our reference collection in the future (e.g., direct percussion with copper, indirect percussion with sandstone, etc.) to better assess this, we can hypothesise from these results that the MFA may have more influence than the CT on the knapping traces we examined, and that the identification of techniques may therefore prioritize the identification of the MFA over that of the CT.

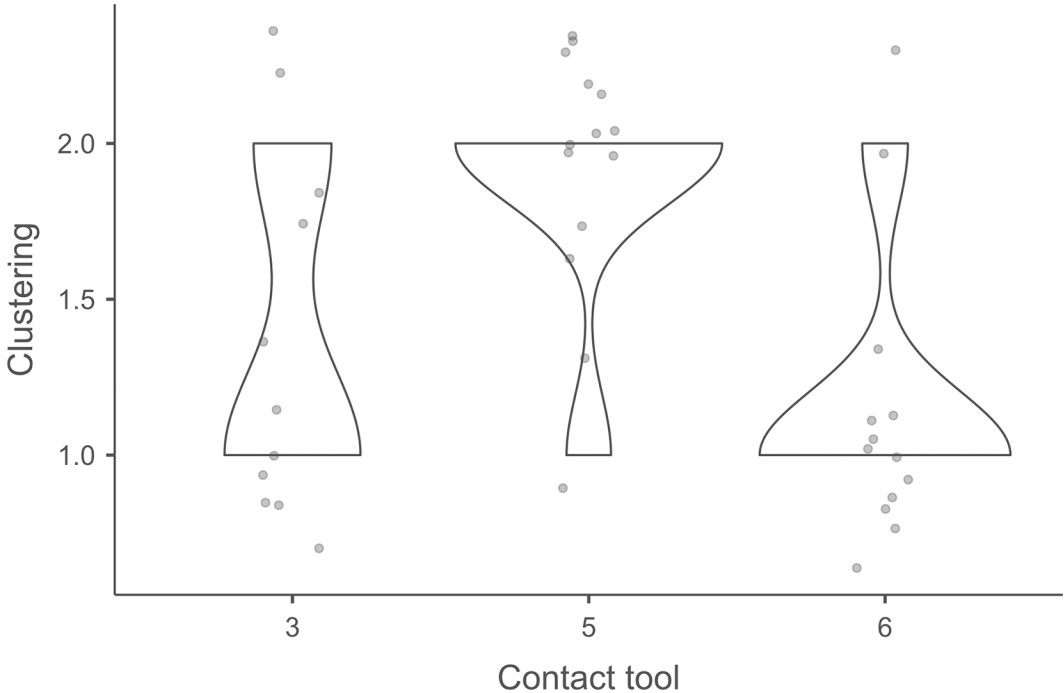

**Fig 43. Plot showing the clustering of CTs associated with pressure based on two clusters (CT 3: antler; CT 5: boxwood; CT 6: copper; each dot corresponds to a blade).**

**Table 12. Results of the hierarchical cluster analysis applied to the identification of CTs used with pressure, based on two clusters.**

| CTs | | Cluster 1 | Cluster 2 | Total |
|---|---|---|---|---|
| Antler | Observed | 7 | 4 | 11 |
| | Expected | 5.79 | 5.21 | 11.0 |
| | % within row | 63.6% | 36.4% | 100.0% |
| Boxwood | Observed | 2 | 12 | 14 |
| | Expected | 7.37 | 6.63 | 14.0 |
| | % within row | 14.3% | 85.7% | 100.0% |
| Copper | Observed | 11 | 2 | 13 |
| | Expected | 6.84 | 6.16 | 13.0 |
| | % within row | 84.6% | 15.4% | 100.0% |
| Total | Observed | 20 | 18 | 38 |
| | Expected | 20.00 | 18.00 | 38.0 |
| | % within row | 52.6% | 47.4% | 100.0% |

## 5. Discussion

While knapping methods can be reconstructed based on a technological analysis of the archaeological artefacts they produced (flakes, blades, bladelets, core tablets, cores, bifacial pieces, production waste, etc.), ideally supplemented by physical refits, the same cannot be said for knapping techniques. In order to identify the latter, it is indeed necessary to decipher the meaning of the traces present on the lithic artefacts, which require the prior creation of an experimental

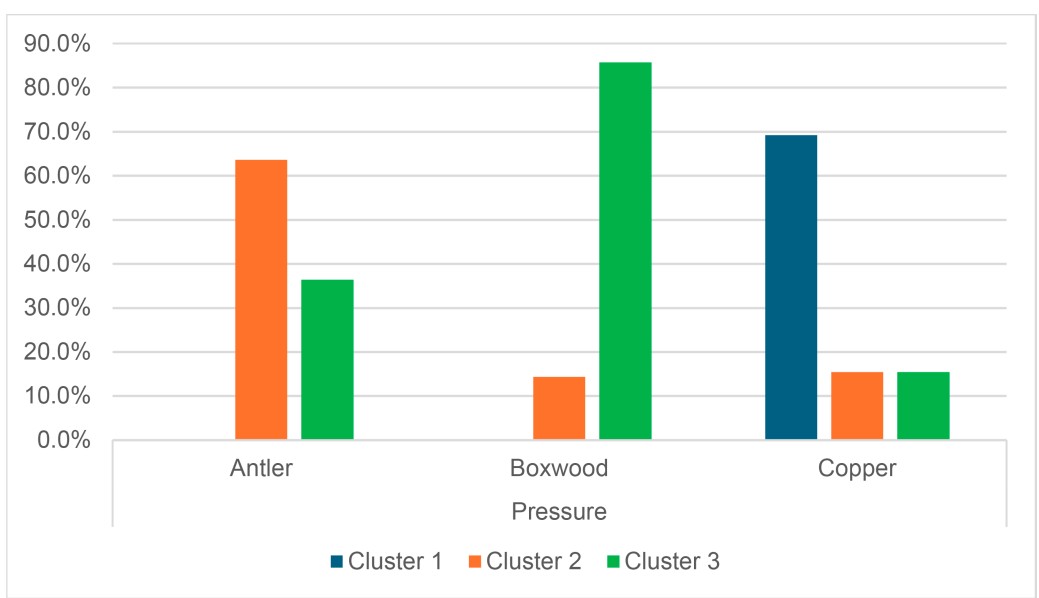

**Fig 44. Clustering of CTs associated with pressure based on three clusters (in %).**

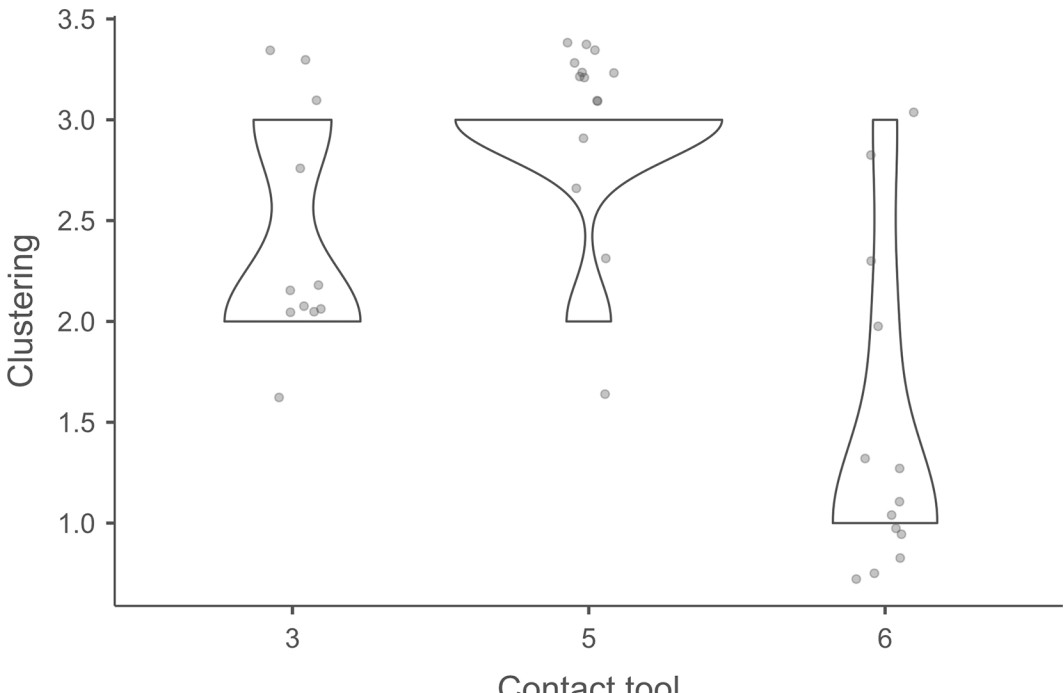

**Fig 45. Plot showing the clustering of CTs associated with pressure based on three clusters (CT 3: antler; CT 5: boxwood; CT 6: copper; each dot corresponds to a blade).**

**Table 13. Results of the hierarchical cluster analysis applied to the identification of CTs used with pressure, based on three clusters.**

| CTs | | Cluster 1 | Cluster 2 | Cluster 3 | Total |
|---|---|---|---|---|---|
| Antler | Observed | 0 | 7 | 4 | 11 |
| | Expected | 2.61 | 3.18 | 5.21 | 11.0 |
| | % within row | 0.0% | 63.6% | 36.4% | 100.0% |
| Boxwood | Observed | 0 | 2 | 12 | 14 |
| | Expected | 3.32 | 4.05 | 6.63 | 14.0 |
| | % within row | 0.0% | 14.3% | 85.7% | 100.0% |
| Copper | Observed | 9 | 2 | 2 | 13 |
| | Expected | 3.08 | 3.76 | 6.16 | 13.0 |
| | % within row | 69.2% | 15.4% | 15.4% | 100.0% |
| Total | Observed | 9 | 11 | 18 | 38 |
| | Expected | 9.00 | 11.00 | 18.00 | 38.0 |
| | % within row | 23.7% | 28.9% | 47.4% | 100.0% |

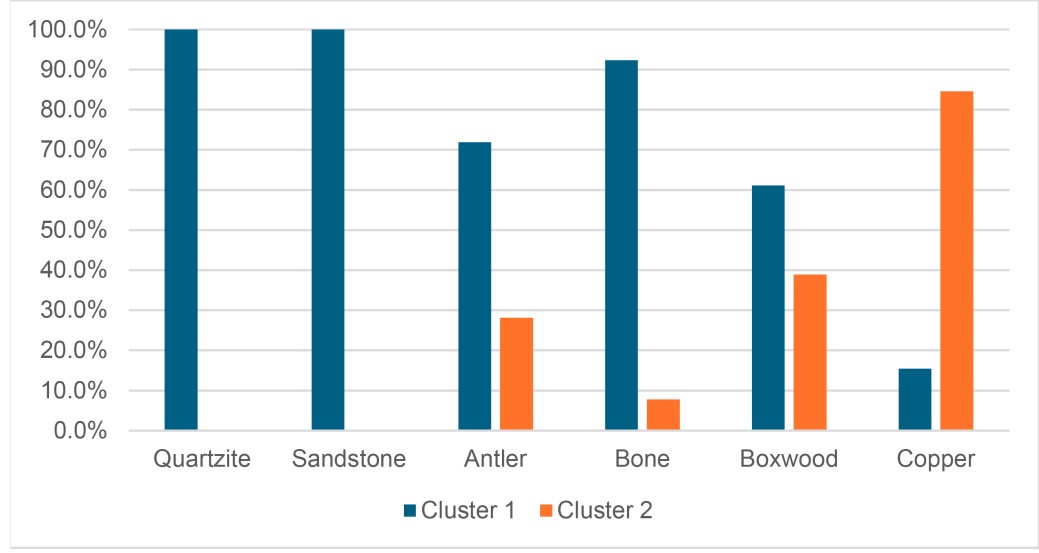

**Fig 46. Clustering of CTs based on two clusters (in %).**

collection according to specific conditions (raw material, type of production, type of striking/pressure platform, etc.) that reproduce the characteristics of the archaeological assemblage we intend to study as closely as possible [123].

To our knowledge, only one study to date has looked at the microscopic knapping traces associated with a wide range of techniques [158]. That study and this article show that the characteristics of these traces vary according to the MFA and the CT used and that they therefore represent an asset for identifying knapping techniques, in the same way as macroscopic traces. However, whether microscopic or macroscopic, no single type of trace can identify a particular technique with certainty. As an example, we can mention the bulb scar (*esquillement du bulbe* in French) which, when it was first recognised, was considered to be closely associated with direct soft stone percussion [123], before cases were also recorded with direct percussion with antler and boxwood [124]. In our experimental collection, bulb scars were mainly observed with direct percussion with antler (n = 5/10), quartzite (n = 5/11) and sandstone (n = 3/11), but discrete bulb scars were also observed sometimes with direct percussion with bone (n = 1/13), indirect percussion with antler and boxwood

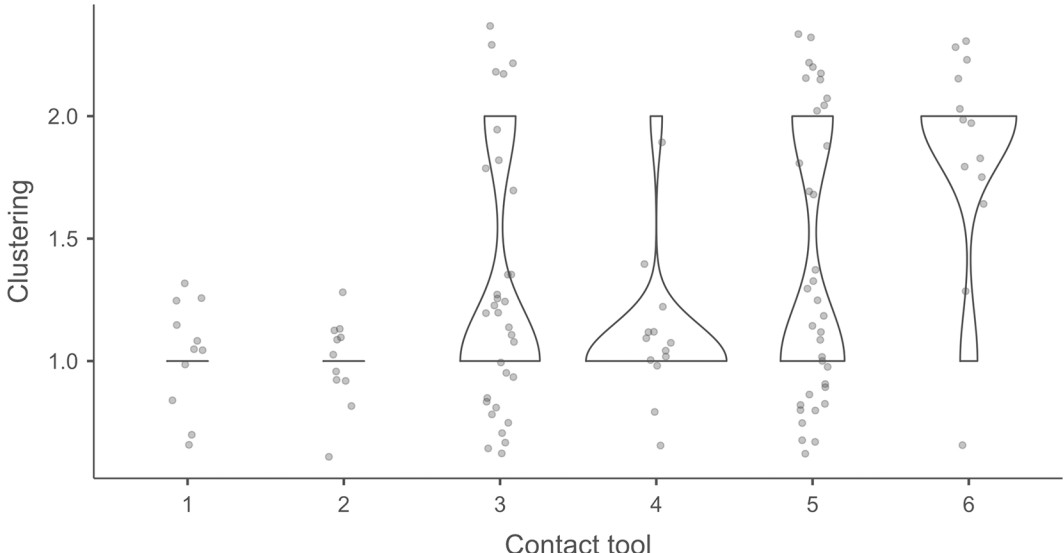

**Fig 47. Plot showing the clustering of CTs based on two clusters (CT 1: quartzite; CT 2: sandstone; CT 3: antler; CT 4: bone; CT 5: boxwood; CT 6: copper; each dot represents a blade).**

**Table 14. Results of the hierarchical cluster analysis applied to the identification of CTs, regardless of the MFAs used, based on two clusters.**

| CTs | | Cluster 1 | Cluster 2 | Total |
|---|---|---|---|---|
| Quartzite | Observed | 11 | 0 | 11 |
| | Expected | 7.68 | 3.32 | 11.0 |
| | % within row | 100.0% | 0.0% | 100.0% |
| Sandstone | Observed | 11 | 0 | 11 |
| | Expected | 7.68 | 3.32 | 11.0 |
| | % within row | 100.0% | 0.0% | 100.0% |
| Antler | Observed | 23 | 9 | 32 |
| | Expected | 22.34 | 9.66 | 32.0 |
| | % within row | 71.9% | 28.1% | 100.0% |
| Bone | Observed | 12 | 1 | 13 |
| | Expected | 9.08 | 3.92 | 13.0 |
| | % within row | 92.3% | 7.7% | 100.0% |
| Boxwood | Observed | 22 | 14 | 36 |
| | Expected | 25.14 | 10.86 | 36.0 |
| | % within row | 61.1% | 38.9% | 100.0% |
| Copper | Observed | 2 | 11 | 13 |
| | Expected | 9.08 | 3.92 | 13.0 |
| | % within row | 15.4% | 84.6% | 100.0% |
| Total | Observed | 81 | 35 | 116 |
| | Expected | 81.00 | 35.00 | 116.0 |
| | % within row | 69.8% | 30.2% | 100.0% |

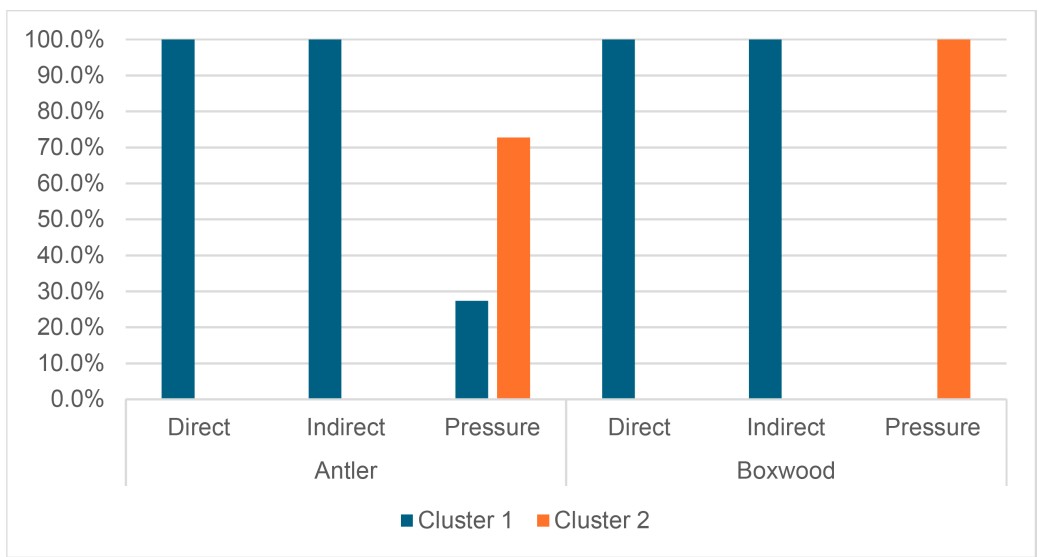

**Fig 48. Clustering of antler and boxwood CTs, depending on the MFAs used, based on three clusters (in %).**

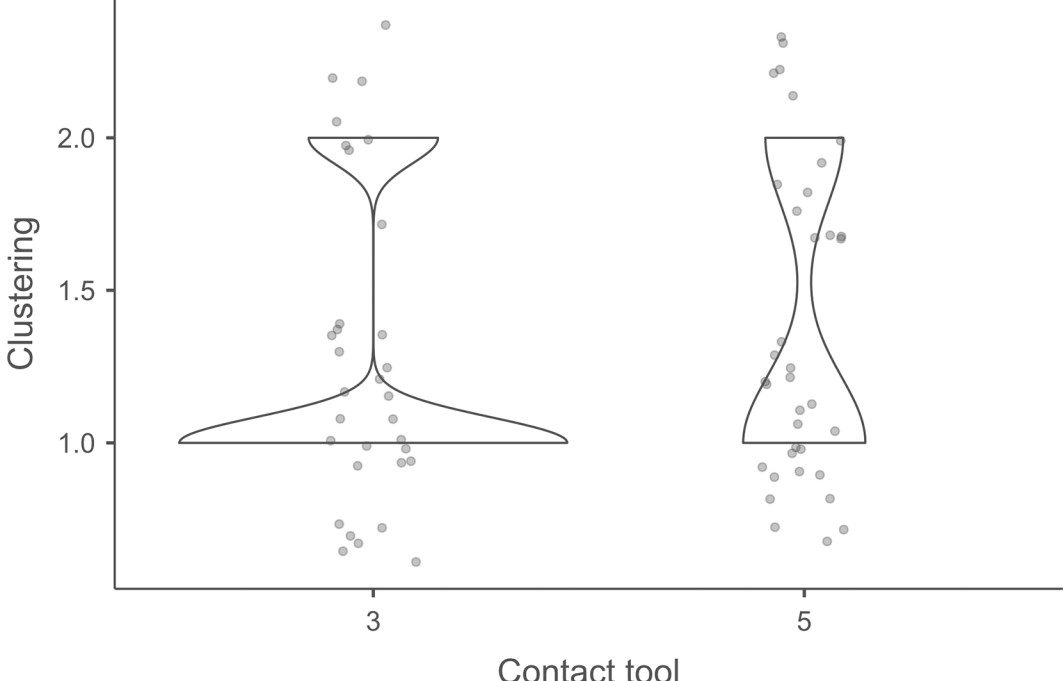

**Fig 49. Plot showing the clustering of antler and boxwood CTs, depending on the MFAs used, based on three clusters (CT 3: antler; CT 5: boxwood; each dot represents a blade).**

**Table 15. Results of the hierarchical cluster analysis applied to the identification of antler and boxwood CTs, depending on the MFAs used, based on two clusters.**

| CTs | | Cluster 1 | Cluster 2 | Total |
|---|---|---|---|---|
| Antler | Observed | 24 | 8 | 32 |
| | Expected | 21.6 | 10.4 | 32.0 |
| | % within row | 75.0 % | 25.0 % | 100.0 % |
| Boxwood | Observed | 22 | 14 | 36 |
| | Expected | 24.4 | 11.6 | 36.0 |
| | % within row | 61.1 % | 38.9 % | 100.0 % |
| Total | Observed | 46 | 22 | 68 |
| | Expected | 46.0 | 22.0 | 68.0 |
| | % within row | 67.6 % | 32.4 % | 100.0 % |

(n = 1/11 in both cases), and pressure with antler (n = 1/11). As a consequence, several researchers have suggested that the identification of techniques should be based on 1) the frequency of appearance of certain *combinations* of traces, and 2) the examination of large samples (e.g., [121]: 57, [176]: 21]), and we share their point of view.

The microscopic traces we examined (residues, cracks, polishes, incisions, and spontaneous scars) provide useful information for identifying the knapping techniques used in the past, notably because most of them, when present on the platforms of the blades, represent direct consequences of the physical interaction between the CT and the striking/pressure platform, which is not the case for several macroscopic traces and features frequently considered in technological studies (e.g., bulb size, lip, platform size, external platform angle, morphology, dimensions and mass of the blades, etc. [158]: 55]).

The very frequent presence of linear polishes associated with incisions can be a useful indicator of the use of percussion (direct or indirect), as these traces are less frequent when blade production is made by pressure. Conversely, the total absence of polishes, linear or non-linear, suggests that the friction between the CT and the striking/pressure platform was not sufficiently abrasive and/or was exerted with sufficient force to cause attrition of the flint microtopography. Such a phenomenon is compatible with pressure knapping using a rather soft CT. The formation of spontaneous scars is linked to the behaviour of the blade immediately after detachment, which could itself be influenced by the MFA used and the way in which the core is held by the knapper. Finally, the residues present on the platform of the blades are obviously very useful for identifying knapping techniques, since analysis of these deposits of material can make it possible to identify precisely the nature of the CT used.

However, the analysis of microscopic traces can face several obstacles, because archaeological reality is obviously different from an experiment carried out under laboratory conditions. This is why it may be necessary to adapt the analysis of knapping traces to the archaeological material under study. The preservation of residues, particularly organic residues, depends very much, for example, on the environment in which the lithic artefacts were buried (e.g., [177]). Although we have observed that these residues can adhere strongly to the platform of blades, unfavourable taphonomy can degrade them to the point where they disappear. The conservation of the lithic artefacts may also hinder or prevent the analysis of the polishes, for example if a deep patina has developed on the artefacts. Moreover, although we have (intentionally) studied blades with a flat striking/pressure platform, we suspect that the formation of polish somewhat differs if the CT comes into contact with the ridges of a platform that has undergone specific preparation (e.g., dihedral, spur, faceted platforms, etc.). In such cases, more or less pronounced (depending on the hardness of the CT and the force applied by the knapper) micro-crushing of these ridges may potentially predominate and obscure polish formation [158]: 55].

In the case of cracks, it is the nature of the knapped rock that can pose a problem. Observation of these traces is greatly facilitated if the rock is translucent, because by placing a light source in front of the dorsal or ventral face of the

blade, it is generally possible to observe cracks located even in the centre of the platform (see Fig 21b). However, this solution becomes ineffective if the rock is opaque. In addition, spontaneous scars observed on the edges of blades can potentially be confused with scars caused by phenomena other than knapping, such as use, hafting, trampling, taphonomy, or post-excavation handling, especially if their characteristics are not taken into account in sufficient detail. At least some of these limitations can be mitigated, however, if the blades examined have not been used as tools or tool blanks, if they come from a well-preserved site, and if they have been handled with care after being removed from the archaeological layer. The case of the spontaneous scars that sometimes form on the posterior edge of the platform is somewhat different, as this edge rarely has a functional role, and it is therefore less likely that these scars result from the use of the blade. In addition, the posterior edge is located at the junction between the platform and the ventral face of the blade, which articulate with each other at a rather open angle, which makes this edge more difficult to chip and therefore taphonomic phenomena are probably less likely to affect it.

In addition to the potential limitations of the archaeological material, the observation of microscopic traces also requires appropriate means of observation, which vary depending on the type of trace one wishes to examine. A stereomicroscope allows for detailed observation of cracks and spontaneous scars as well as the detection of possible residues to some extent, but it is necessary to use a reflected-light microscope to observe polishes and the incisions that are often associated with them, while different techniques, such as reflected-light microscopy, SEM-EDS, GC-MS, or FTIR, can provide finer characterization of residues. The use of these instruments and the subsequent recording of data mean that analysis takes longer than in the case of purely macroscopic observation.

While the study of microscopic traces therefore comes with several constraints, we feel that these are largely offset by the data provided by low and high magnification instruments, and by the usefulness of this data for identifying knapping techniques. In our view (see also [155,158]), microscopic traces should therefore be considered more often in the future – in combination with macroscopic traces –, as the former extend and therefore complement the information derived from the latter.

Furthermore, as J. Pelegrin underlined [125: 40], a certain knowledge of the technical principles of knapping, and more generally of fracture mechanics, is also required to be able to link the traces observed to the factors that trigger their appearance. The case of the lip is a good example. The appearance of this feature if often associated with the use of an organic CT or, more generally, a soft CT, particularly when the MFA is direct percussion (e.g., [123,137,141]), but an explanation for this phenomenon is rarely provided. It is important to stress here that lip formation is not directly dependent on the use of a soft CT, since hard stone or even steel can produce lipped flakes as has been demonstrated on several occasions [59,72]. The presence of a lip is only evidence of a fracture that was bending initiated (and is therefore not conchoidal [65]). In terms of knapping, bending initiation is the result of a gesture characterised by a dominant tensile component and a comparatively more discreet compressive component. In the case of direct percussion, such an imbalance corresponds to what is often called a "tangential" gesture (e.g., [149]). This type of gesture can be performed with both hard and soft CTs, but is more often required with the latter, as causing a conchoidal fracture – i.e., by means of a gesture characterised by a dominant compressive component and a comparatively weaker tensile component ("internal" gesture; e.g., [149]; in our opinion, however, "internal" seems to refer mainly to the distance between the edge of the platform and the contact point and could be opposed to "marginal", while the term "tangential", on the other hand, refers to the kinematics of the gesture and the way in which the force is applied by the knapper, and would be more appropriately opposed to "compressive"; from this perspective, the application of force can involve a compressive/inward or tangential/outward gesture and an internal or marginal contact point) – with a CT that is significantly softer than the knapped material is more difficult, and can potentially damage the CT if the force applied to compensate for this lack of hardness is excessive. Conversely, a hard CT allows a certain flexibility of use in that it can theoretically be used to cause bending initiated fractures as well as conchoidal fractures. All other things being equal, the "efficiency range" of a CT – i.e., the set of conditions under which a raw material can be successfully fractured – therefore varies according to the hardness of the CT. It can be

pointed out that, depending on their levels of know-how and personal experiences, knappers undoubtedly perceive this efficiency range to various degrees. In other words, we can distinguish between the *physical* efficiency range of the CT (what is physically possible) and the *perceived* efficiency range (what is possible according to the knapper). In any case, the influence of idiosyncratic knapping behaviour probably explains why the appearance of the lip is more or less frequent depending on the individual [140]. It would be desirable in the future to better integrate the field of fracture mechanics into the study of knapping techniques, to better understand the conditions of formation of certain features.

In this study, we followed the advice mentioned above to base the identification of knapping techniques on combinations of traces. To do this, we chose to apply hierarchical cluster analysis, which is an excellent statistical tool for detecting, in a large dataset, specific patterns of attributes that result from the use of particular knapping techniques. However, because a technique is the product of the interaction of several parameters (MFA, CT and body behaviour; Tixier 1967), each of which influences these attributes to varying degrees, it is essential to separate their identification, using adapted lists of attributes that enable one of these variables to be targeted, while minimising the influence of the other as much as possible. Based on this study, it also appears that the order in which the MFA and the CT are identified is important. Although the composition of our experimental collection does not currently allow us to extend our conclusions to all the CTs we have used, the data for antler and boxwood indicate that the MFA is likely the most influential of these two parameters and should therefore be identified first. We also deduce from this, by way of hypothesis, that the presence (direct and indirect percussion) or absence (pressure) of impact in the creation of a fracture has more influence than the nature of the CT used. Furthermore, although we have focused here on the MFA and the CT, as in most studies on knapping techniques (but see Clément 2022), the influence of the body behaviour should not be underestimated as it is clear that this parameter influences multiple knapping traces. For example, when several unsuccessful percussions or pressures attempts are performed before a blade is finally detached, it can be expected that linear polishes, if present, will have various orientations; the location of residues could be more dispersed; network cracks could be more frequent, etc. Consequently, future experiments will certainly benefit from taking greater account of the body behaviour than we have done in this work.

Based on the results of the cluster analyses performed for the MFAs, identifying the use of percussion (direct or indirect) or pressure appears as the easiest task. Because they probably induce similar consequences in terms of fracture mechanics, direct percussion and indirect percussion are a little more difficult to distinguish and it may be necessary to adjust the number of clusters to be able to identify them. Once the MFA has been determined, the cluster analyses we carried out enabled us to distinguish – for each MFA – between rather hard and rather soft CTs. However, we remain cautious as to whether it is possible to further characterise the CT in an archaeological context based on the analysis of knapping traces and related attributes, including the distinction between the use of a mineral CT and an organic CT. Moreover, some experimental results have shown that very soft stones can lead to the formation of trace combinations compatible with the use of an organic CT [6], which reinforces our scepticism. In our opinion, however, the best (if not the only?) way to refine the identification of the CT is to find and analyse the residues it may have left on the platform of the blades. As we have seen, these residues can be highly adhesive, particularly when they are organic and related to the use of direct percussion, which suggests that they could be frequently detected in an archaeological context.

Our clustering analyses were carried out based on specific lists of attributes, which are statistically related to the MFA or the CT, in order to assess the degree of precision that can potentially be achieved in the identification of knapping techniques from the analysis of macroscopic and microscopic knapping traces. Consequently, if the lists of attributes that we have used in the cluster analyses are applied to other lithic assemblages, we believe that they should be conceived exclusively as sources of inspiration and not as 'magic recipes' that would enable knapping techniques to be identified with great precision. Indeed, these lists of attributes do not constitute a method in the strict sense of the term for identifying techniques, which could be applied to archaeological assemblages. The aim of this study was to establish the theoretical bases on which such a method should be developed, i.e., a method based on the analysis of both macroscopic and

microscopic knapping traces which should be able to target and identify each of the three main parameters involved (MFA, CT, and body behaviour) according to the traces they most influence. Although we decided to focus on knapping traces in this study, it is clear to us that such a method would need to include other features that could prove equally useful, such as blade morphology for the recognition of pressure knapping for example [39,117,125,128] (see S3 and S4 Tables). It could also be interesting in the future to quantify other aspects not taken into account here, such as the length of linear polishes (to better assess the extent of the friction between the CT and the striking/pressure platform), or the distribution area of the polishes on the platform (to better assess the extent of the contact area, which certainly depends in part on the hardness of the CT). Finally, we can anticipate that certain variables will require this method to be adjusted depending on the archaeological assemblage studied. Raw materials particularly come to mind, since we are working with flint and it is already known that certain knapping features, such as the *contre-col*, are specific to certain rocks but are not observed with this material [148,149].

Another point worth returning to is the frequent presence of spontaneous scars along the lateral edges of the blades and the posterior edge of their platforms, and of residues on their ventral or dorsal faces, that are all linked to the movement of the blade immediately after its detachment. This movement is largely defined by the conditions under which the blade and the core are held, conditions that themselves derive from the third parameter of knapping techniques: the body behaviour [1] – at least for techniques that do not use a device to hold the core (direct and indirect percussion *a priori*, but also modes 1a, 1b and 2 of pressure [129]). In other words, in an archaeological context, examining these traces could in theory enable reconstructing the way in which the core was held during knapping and therefore at least one aspect of the body behaviour. In practice, however, this task will undoubtedly be complicated by the fact that these traces, when present on archaeological artefacts, can have multiple origins besides knapping (use, hafting, trampling, taphonomy, etc.).

More generally, understanding the influence of body behaviour on the formation of knapping traces appears as a major challenge for future research, which will require studying the inter-individual variability of this behaviour and therefore working with several knappers. Moreover, the body behaviour is a fairly general concept which encompasses several parameters in addition to the way the core is held, such as the kinetic energy of the knapping gesture, the kinematics of the gesture which determines the angle of incidence of the CT at the moment of fracture initiation, and the body position (e.g., sitting on a chair, sitting on the floor, etc.), the respective influences of which should be evaluated as J. Tixier has emphasized [176: 15]. In the future, we will also have to test combinations of MFAs and CTs that are not yet included in our experimental collection (e.g., direct percussion with copper, indirect percussion with sandstone, etc.) in order to extend the study of the knapping traces, but also to validate the predominant influence of the MFA on the latter, which we can only suspect for the moment based on the blades knapped with antler and boxwood CTs.

## 6. Conclusion

When knapping, the technique used to cause the detachment of a flake leads to the formation of microscopic traces, or at least traces that require the use of optical instruments to be finely characterised (residues, cracks, polishes, incisions and spontaneous scars). These traces have been largely ignored in lithic technology until now, although they can sometimes show significant variations depending on the technique used, just like the traces that can be characterised with the naked eye (e.g., bulb, lip, percussion cone, external platform angle, blade morphology, etc.), and can therefore contribute to the recognition of prehistoric knapping techniques. From this point of view, micro and macroscopic knapping traces are complementary.

This complementarity can be put to good use in identifying these techniques by means of hierarchical clustering analyses, which are highly effective in identifying specific patterns of attribute combinations. However, it is imperative to proceed in stages and first identify the MFA and then the CT used, using lists of appropriate attributes, i.e., attributes that are closely correlated with these two variables. Our results suggest that it is relatively easy to identify sets of blades produced by percussion and pressure in this way. It is also possible to distinguish between direct percussion and indirect

percussion, although this appears to be less obvious, probably because these two MFAs behave in a similar way since the fracture is created in both cases following an impact. It is also possible to distinguish between rather hard *versus* rather soft CTs, regardless of the MFA used, but the characterisation of CTs can only go so far with this type of analysis. The identification and analysis of knapping residues, which can adhere strongly to the flint, is the best option for obtaining more precise information on this aspect.

While the MFA and CT can be identified, the body behaviour is very difficult to perceive from the knapping traces, even if at least one of its aspects (how the core is held) can be comprehended, albeit in an experimental context. Analysing the effect of body behaviour on knapping traces will necessarily involve looking at the inter-individual variability of those traces. On this subject too, it would undoubtedly be advisable to opt for a multiscale approach, drawing on both the technological approach and the traceological approach (e.g., [87]).

## Supporting information

**S1 Table. List of studies reviewed in detail and their main parameters ('1' means that the parameter is present in the study, '0' means that the parameter is absent, 'NA' means that the parameter is not applicable).** Raw materials highlighted in red were heat-treated; raw material highlighted in beige is quartzite; raw material highlighted in yellow is resinite opal; raw material highlighted in brown is volcanic tuff; raw material highlighted in blue was knapped only by pressure.
(XLSX)

**S2 Table. List of the 185 attributes used in the sample of studies analysed, with an indication of the usefulness of these attributes for identifying techniques.** The correspondence with the simplified attributes used in Supplementary 3 is also indicated. The names of studies in black indicate studies on flint or chert; the names of studies in red indicate studies on obsidian or glass; the names of studies in purple indicate studies on flint or chert and obsidian or glass; the names of studies in green indicate studies on quartzite; the names of studies in blue indicate studies on all materials.
(XLSX)

**S3 Table. List of the 82 simplified attributes defined on the basis of the sample of studies analysed, with an indication of the usefulness of these attributes for identifying techniques.** The names of studies in black indicate studies on flint or chert; the names of studies in red indicate studies on obsidian or glass; the names of studies in blue indicate studies on all materials (quartzite is excluded because the number of studies is too small). Attributes unanimously considered useful are highlighted in green; attributes unanimously considered non useful are highlighted in orange; attributes with contradictory results are highlighted in blue; attributes whose results seem contradictory due to the grouping of attributes are highlighted in grey.
(XLSX)

**S4 Table. Count of the cases of convergence and divergence observed for the simplified attributes that appear in at least three different publications ('1' means that the observation is positive, '0' means that the observation is negative).** Attributes unanimously considered useful are highlighted in green; attributes unanimously considered non useful are highlighted in orange; attributes with contradictory results are highlighted in blue.
(XLSX)

**S5 Table. Data from the analysis of macroscopic and microscopic knapping traces observed on the experimental blades.**
(XLSX)

**S6 Table. List of the 151 attributes used in the study, indicating whether they were also included in the hierarchical cluster analyses.** For the attributes included in the latter, the p-values of the Fisher's exact tests and the p-values

of the Kruskal-Wallis tests are provided, as well as an indication of the use of the attributes in each of the five situations tested (identification of the MFA regardless of the CT used, identification of the CT regardless of the MFA used, identification of the CT when the MFA is direct percussion, identification of the CT when the MFA is indirect percussion, and identification of the CT when the MFA is pressure).
(XLSX)

**S7 Table. Coded data used to perform hierarchical cluster analyses.**
(XLSX)

**S1 Text. Description of the attributes used to document the microscopic knapping traces considered in the visual analysis of the experimental blades.**
(DOCX)

**S1 Fig. Histograms describing several aspects of the sample of studies examined in the section *2***. *Review of the knapping features used in the literature and their diagnostic value.*
(DOCX)

**S2 Fig. Extract from the high-speed video recording of the detachment of the blade bearing the identification number Exp122-78, immediately after the fracture was created.** Note that the knapper has positioned the core so that the striking platform is inclined at around 30° to the ground. The way he holds the core allows the proximo-mesial part of the blade (yellow circle) to be free of any constraint. This part can therefore tilt forward after the blade has been extracted, but as its distal part is still held firmly between the core and the knapper's thigh, the blade returns towards the core and hits it. This specific position creates spontaneous scars with intact denticles on the proximo-mesial part of the blade, and spontaneous scars with crushed denticles on its distal part.
(TIFF)

**S3 Fig. Extract from the high-speed video recording of the detachment of the blade bearing the identification number Exp122-91, immediately after the fracture was created.** Note that the knapper has positioned the core so that the striking platform is almost parallel to the ground. In this case, the entire length of the flaking surface is firmly maintained against the knapper's thigh and only the proximal part of the blade (yellow circle) can barely move away from the core. In this position, there is no significant friction between the blade and the core, so that spontaneous scars show only intact denticles.
(TIFF)

**S4 Fig. Dendrogram of the hierarchical cluster analysis applied to the identification of MFAs, regardless of the CTs used, based on two clusters.**
(TIFF)

**S5 Fig. Dendrogram of the hierarchical cluster analysis applied to the identification of MFAs, regardless of the CTs used, based on three clusters.**
(TIFF)

**S6 Fig. Dendrogram of the hierarchical cluster analysis applied to the identification of MFAs, regardless of the CTs used, based on eight clusters.**
(TIFF)

**S7 Fig. Dendrogram of the hierarchical cluster analysis applied to the identification of CTs used with direct percussion, based on three clusters.**
(TIFF)

**S8 Fig. Dendrogram of the hierarchical cluster analysis applied to the identification of CTs used with direct percussion, based on four clusters.**
(TIFF)

**S9 Fig. Dendrogram of the hierarchical cluster analysis applied to the identification of CTs used with indirect percussion, based on two clusters.**
(TIFF)

**S10 Fig. Dendrogram of the hierarchical cluster analysis applied to the identification of CTs used with pressure, based on two clusters.**
(TIFF)

**S11 Fig. Dendrogram of the hierarchical cluster analysis applied to the identification of CTs used with pressure, based on three clusters.**
(TIFF)

**S12 Fig. Dendrogram of the hierarchical cluster analysis applied to the identification of CTs, regardless of the MFAs used, based on two clusters.**
(TIFF)

**S13 Fig. Dendrogram of the hierarchical cluster analysis applied to the identification of antler and boxwood CTs, regardless of the MFAs used, based on two clusters.**
(TIFF)

**S1 Video. High-speed video recording (16,000 fps) of the detachment of the blade bearing the identification number Exp122-78 by indirect percussion with an antler punch.** Note that at 00:00:21, the video documents the detachment of a tiny flake on the mesial part of the right edge of the blade, due to the blade's counter-shock against the core. The negative of this flake corresponds to a spontaneous scar. The video also shows that after impact, the wooden hammer shakes slightly as it absorbs some of the kinetic energy of the knapping gesture. The quality of this video has been reduced to comply with PLoS ONE's guidelines regarding supporting information; the original, higher-quality video can be provided by the authors upon request.
(MP4)

**S2 Video. High-speed video recording (16,000 fps) of the detachment of the blade bearing the identification number Exp122-91 by indirect percussion with a boxwood punch.** The video very clearly shows that after impact, the wooden hammer shakes as it absorbs some of the kinetic energy of the knapping gesture. The quality of this video has been reduced to comply with PLoS ONE's guidelines regarding supporting information; the original, higher-quality video can be provided by the authors upon request.
(MP4)

## Acknowledgments

Our sincere thanks go to Christian Lepers for producing the experimental collection. We are grateful to Philippe Pirson, the Société de Recherche préhistorique en Hainaut (Society for Prehistoric Research in Hainaut) and the OMYA quarry for their help in collecting the flint blocks used in the experiment. We also express our gratitude to Cécile Ménager and Kim Redman for providing us with their unpublished theses, Dries Cnuts for performing the SEM-EDS analyses of several residues, as well as the TraceoLab team for helpful discussions and support, and especially Lena Asryan, Dries Cnuts, Justin Coppe, Solène Escarguel, Noora Taipale and Sonja Tomasso. This article is published with the support of the University Foundation of Belgium (AS-0613), to whom we extend our gratitude. Finally, we thank the two anonymous reviewers for their constructive comments, which have helped improve our manuscript.

## Author contributions

**Conceptualization:** Olivier Touzé, Veerle Rots.

**Data curation:** Olivier Touzé, Veerle Rots.

**Formal analysis:** Olivier Touzé.

**Funding acquisition:** Olivier Touzé, Veerle Rots.

**Investigation:** Olivier Touzé.

**Methodology:** Olivier Touzé, Veerle Rots.

**Project administration:** Veerle Rots.

**Supervision:** Veerle Rots.

**Visualization:** Olivier Touzé.

**Writing – original draft:** Olivier Touzé.

**Writing – review & editing:** Olivier Touzé, Veerle Rots.

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
