## [Decision Letter · Decision Letter 0]

30 Apr 2025

Dear Dr. Touzé,

We look forward to receiving your revised manuscript.

Kind regards,

Marco Peresani

Academic Editor

PLOS ONE

Reviewers' comments:

Reviewer's Responses to Questions

**Comments to the Author**

1. Is the manuscript technically sound, and do the data support the conclusions?

Reviewer #1: No

Reviewer #2: Yes

2. Has the statistical analysis been performed appropriately and rigorously?

Reviewer #1: No

Reviewer #2: Yes

3. Have the authors made all data underlying the findings in their manuscript fully available?

Reviewer #1: Yes

Reviewer #2: Yes

4. Is the manuscript presented in an intelligible fashion and written in standard English?

Reviewer #1: Yes

Reviewer #2: Yes

Reviewer #1: The manuscript contributes to the extensively researched topic of knapping techniques and the question of whether these can be identified archaeologically based on features of the knapped blanks. The originality of this work lies in the authors' approach, which envisions the combined use of a wide range of macroscopic and microscopic features for technique identification. This is particularly relevant since, thus far, only a few works have explored the microscopic traces left by different techniques, despite their informative potential. However, there are significant flaws in both the conceptualisation of the work and the presentation of results that must be addressed before the paper can be published.

Major Issues

Experimental Design and Conceptualisation: One major issue is the lack of a theoretical framework that connects the features observed on the blades to the MFAs and CTs, particularly concerning the microtraces. This aspect is crucial because the experiment is conducted under “realistic” conditions, and many variables, such as those related to body behaviour, are uncontrolled. This lack of control prevents isolating correlations arising from factors not directly related to the techniques from causal ones. The authors should clarify why they chose this approach and outline any measures taken to mitigate the impact of confounding variables. Features that may be influenced by uncontrolled variables should not be included in the analysis.

There are several examples demonstrating how this approach might have affected the experimental results and introduced confusion in data treatment and discussion. One example concerns the number and type of cracks (network cracks). As suggested by the authors in lines 766-767, 808-810, and 1120-1122, these features appear to be associated with the number of removal attempts and/or the shape of the CT active area, rather than the CT type. Consequently, the observed correlation between CT type and these features is merely apparent and should not be regarded as significant.

I see similar problems with post-fracture traces, such as spontaneous scars on the edges. The development of these traces seems linked to how the core was held or supported, not to the knapping technique used, as also noted by the authors on Page 49. In this case, there is a risk of emphasising a connection between scars and CTs when what we are observing is the knapper adjusting his body behaviour according to different shapes and sizes of cores and punches, as is also visible in Supplementary Information 7. Why, indeed, should the knapper adopt a different position when using an antler or boxwood CT?

A similar reasoning applies to lines 1021-1026 and lines 1046-1054: the orientation of linear polishes and the location of the residues, respectively.

In all these cases, there is no causal relationship between the observed features and CT type. All of these features seem related to body behaviour or other factors that were not controlled for and were not considered in the experimental design, rather than CT or MFA. Nevertheless, they are included in the analysis. The authors should focus solely on attributes that can be unequivocally linked to CT and/or MFA.

Experiment description: Some additional details on how the experiment was conducted would also be useful, such as the number of strikes for each blank removal, the position of the knapper, and a description of the hammers, punches, and crutches used to extract the blades (how many hammers of each type were involved? What were their sizes and shapes? etc.)

Statistics: I am not an expert in multivariate statistics, but I think there are a few points where the authors should double-check the methods and/or provide a more extensive explanation of their choices. It is unclear to me why the attributes considered in the hierarchical cluster analysis are selected based on the analysis of single variables (ANOVA and Chi-square tests). Is it not circular to perform the cluster analysis only on significantly different attributes?

Furthermore, I have some concerns regarding the tests conducted on single variables. In many cases, especially when crossing CTs and MFAs, the samples are small, falling below the recommended threshold for the Chi-square test. For these small samples, a Fisher's exact test would be better suited. Similarly, for testing the variability between groups of values that the authors have no reason to assume are sampled from distributions with similar shape and scale and (possibly) only different medians, the non-parametrical Kruskal-Wallis test is more appropriate than ANOVA.

Another issue pertains to the coding of nominal attributes. From what I understand, cluster analysis can be used with continuous, ordered, or binary variables. When using nominal values, they should be transformed in a way that allows the algorithm to compute the distances between the observations. Here, however, nominal categorical variables are converted into ordered ones, even though there is no established order among the states of every attribute. For example, the lateralization of oblique cracks could be “left, " “right, " “both sides,” or “N/A." These states are coded 1-2-3-0, but, in this way, the result is that the distance between “right” (2) and “both” (3) is 1, and the distance between “left” (1) and “both” (3) is 2. But the case “left” is not more different than “right” from the case “both”. This should be checked and, if necessary, corrected.

Results: In many cases, the authors claim to observe a correlation between the CT type and the examined features. However, these claims are not clearly supported by the data. The experimental samples are small, and I'm uncertain whether the differences in the observed traces would be considered significant by a Fisher exact test. I recommend conducting such a statistical test before correlating traces with technical traits. Examples include (but are not limited to) lines 659-662, 682-684, 762-764, 787-788, 792-795, and 789-808.

The results of the cluster analysis are described as effective. However, I think that the accuracy of the cluster analysis and its effectiveness in reconstructing aspects of knapping techniques would be more robust if evaluated statistically (e.g., Fisher's exact test or Chi-square test when the sample size permits). Following the cluster analysis, it would be important to discuss which combination of attributes proved to be the most relevant for tracking different aspects of knapping techniques.

Suggestions: To strengthen the paper's results, I recommend developing a solid theoretical framework that justifies the choice of attributes used in the analysis, i.e. why and how do I expect a feature to change with a change in CT and MFA? Features that are clearly linked to body behaviour or other factors (e.g., number and specific types of cracks, edge scars, etc.) should be excluded from the analysis.

Furthermore, the authors should use statistical tools appropriate to the experimental sample size (e.g., Fisher exact test instead of Chi-square) and clarify why selecting attributes for cluster analysis in advance through such tests would not be circular reasoning. The method for coding nominal attributes should also be checked. Lastly, the attributes that proved most useful in differentiating between clusters should be discussed, and statistics might also be applied to the clustering results.

It appears that better results are obtained when comparing larger experimental samples, such as when different MFAs are compared independently by the CT used. Perhaps the authors could focus solely on such more specific aspects that could yield more robust results or seek to increase the sample size by grouping different types of CTs (e.g., comparing organic vs. inorganic CTs).

Minor points

As one of the paper's goals is to further characterise microscopic traces by building on the previous works of Rots, it would be beneficial to include a summary of those works, as not every reader may be familiar with them.

The paper is already lengthy and rich in information. I believe some of the figures provided are not essential; for instance, Figures 1, 2, 4, 5, 6 can be omitted. Lines 468-473 can also be excluded, as this information is not pertinent to the current paper.

It is unfortunate that the authors chose not to include the paper by Magnani et al. (2014) in their sample of studies. This work presents interesting results relevant to the topic at hand and would have contributed to enlarging the sample of papers based on controlled experiments.

In Table 1, it would be helpful to see how many of the blades were complete.

Lines 815-816: The text states that the grain of the flint can influence polish formation. Why is this discussed only in relation to polishes? Does this also pertain to other variables considered?

Lines 1500-1506: In this instance, why is antler classified as a hard material (similar to copper), while in other cases, it is grouped with wood and bone (soft CTs)?

Reviewer #2: Identification of knapping techniques has been an important issue in the study of lithic production technology since the definition by J. Tixier. As many researchers agree, it is clear that the criteria of identification must be clarified by experimental analysis. However, different views have been presented on how to experimentally determine the identification criteria. Some French researchers have argued that the identification of knapping techniques can be based on differences in the morphological features of lithic artifacts that can be observed with the naked eye. However, other researchers have pointed out that the identification criteria proposed there are vague and not necessarily valid indicators. There was a need to present a quantitative analysis method by focusing on traces that more directly reflect differences in knapping techniques.

This paper is an attempt to find a useful indicator for identifying the knapping techniques, using the traces observed in macro- or micro-scorpics as a clue. The paper presents an extremely interesting argument from the following points.

First, the experiment was conducted by setting up a comprehensive experimental procedure that combined two parameters: the mode of force application and the contact tool. The fact that it is now possible to discuss which parameters allow for more reliable identification is an important achievement. Second, the microscopic traces on the platforms of the blades such as residues, cracks, polishes, inclusions, and spontaneous scars represent direct consequences of the physical interaction between the contact tools and the striking/pressure platform. These provide important evidence that reflects directly the processes of contact. The results of the experiments presented here can be used by other researchers to disprove the issues. Third, the analytical results, including microscopic traces, are presented quantitatively, allowing a probabilistic evaluation of the validity of the identification criteria. Last, it is an important result of this paper to show that the observation of residues is also effective in identifying the knapping techniques.

This paper can be evaluated as reaching the level required by the journal and will attract the attention of more researchers. Therefore, I recommend publication of this paper.

**Do you want your identity to be public for this peer review?** For information about this choice, including consent withdrawal, please see our Privacy Policy

Reviewer #1: No

Reviewer #2: No

---

## [Author Response · Author response to Decision Letter 1]

10 Jul 2025

[all the information below can be found in the attached file entitled "Response to Reviewers"]

Dear Dr. Peresani,

We have received your letter and the reports from the two reviewers, whom we sincerely thank for their evaluations. We would particularly like to express our gratitude to Reviewer 1, whose comments have enabled us to improve the manuscript, particularly with regard to statistical aspects.

Please find below the reviewers' comments (in blue), followed by our response (in black).

We have also ensured that the database (file S5 Table) is included as requested by PLoS ONE, and we also added the coded data used to perform hierarchical cluster analyses (file S7 Table). In addition, the file entitled “Manuscript” has been adapted in accordance with PLoS ONE formatting guidelines.

We hope that the revised manuscript will meet the expectations of PLoS ONE and that it will be deemed suitable for publication. We remain at your disposal should further adjustments be necessary.

Kind regards,

Olivier Touzé and Veerle Rots

1. REVIEWER #1

The manuscript contributes to the extensively researched topic of knapping techniques and the question of whether these can be identified archaeologically based on features of the knapped blanks. The originality of this work lies in the authors' approach, which envisions the combined use of a wide range of macroscopic and microscopic features for technique identification. This is particularly relevant since, thus far, only a few works have explored the microscopic traces left by different techniques, despite their informative potential. However, there are significant flaws in both the conceptualisation of the work and the presentation of results that must be addressed before the paper can be published.

1.1. Major Issues

1.1.1. Experimental Design and Conceptualisation

Comment 1

One major issue is the lack of a theoretical framework that connects the features observed on the blades to the MFAs and CTs, particularly concerning the microtraces. This aspect is crucial because the experiment is conducted under “realistic” conditions, and many variables, such as those related to body behaviour, are uncontrolled. This lack of control prevents isolating correlations arising from factors not directly related to the techniques from causal ones. The authors should clarify why they chose this approach and outline any measures taken to mitigate the impact of confounding variables. Features that may be influenced by uncontrolled variables should not be included in the analysis.

Developing a strict theoretical framework to connect the attributes considered with the three parameters that make up a technique (MFA, CT and body behaviour) is complex, because these parameters are not independent: on the contrary, they interact with each other, so that two or three of them can influence an attribute to varying degrees. For example, we have indicated (lines 1699-1719) that the formation of the lip depends primarily on the gesture of the knapper (= body behaviour), but that this gesture can itself be conditioned by the nature of the CT used: a CT that is significantly softer than flint, such as a boxwood hammer, will encourage the knapper to adopt a tangential gesture (i.e. with a strong tensile component and a weak compression component), which in turn will induce a bending-initiated fracture and therefore the creation of a lipped blade.

Regarding microscopic traces more specifically, we have nevertheless regularly mentioned the main factors that we believe influence their formation. For example:

• Linear polishes and incisions seem to be linked to the friction induced by the MFA used (greater in percussion than in pressure), to the abrasiveness of the CT, and to the force of the gesture (e.g. lines 1634-1639).

• Cracks seem to be mainly related to the degree of hardness of the CT (e.g. line 761-762).

• Network cracks could be the result of successive percussions in the same area (= body behaviour), or to a CT with an irregular surface (e.g. lines 808-810).

• Spontaneous scars are primarily related to the way the knapper holds the core (= body behaviour), or to the MFA used (e.g. lines 1640-1642).

From our point of view, developing a more in-depth theoretical framework would, however, comes with a significant risk, as most of the time we lack the figures to establish how each of the parameters of a technique influences the attributes used.

We also chose not to control the body behaviour and to work under “realistic” conditions, essentially because the main, long-term, goal of our research program is to examine the inter-individual variability in blade production. We therefore needed to give the participants enough freedom to act as they usually do during the experiment, even though they always had to use specific MFAs and CTs. Besides, controlling the body behaviour of a knapper is a particularly difficult task. For example, it does not seem possible to control effectively aspects such as the force of the gesture, or the angle at which the CT initiates the fracture, without the use of an automated device.

However, we filmed each blade detachment using a high-speed camera (this information is given in section 3.1.1. Protocol of the experiment, lines 435-437). The aim of these recordings is to obtain quantitative data on at least two aspects of the body behaviour: the kinetic energy and the angle of incidence, to examine their links with certain types of knapping traces. We have chosen to present these data in a separate article (to be published later), for two main reasons:

• Firstly, the main aim of this article is to show, on the basis of the MFA and the CT, i.e. the two parameters that have always been taken into account in the literature related to the identification of knapping techniques (the body behaviour being generally ignored), the interest there is in considering the identification of techniques by proceeding in stages, i.e. by considering separately the identification of the three parameters involved.

• Secondly, the body behaviour is a complex parameter because it encompasses several sub-parameters (kinetic energy, angle of incidence, the way the core is held, the position of the body), something of which J. Tixier was perfectly aware. From our point of view, examining the influence of the body behaviour therefore requires dedicated studies.

Comment 2

There are several examples demonstrating how this approach might have affected the experimental results and introduced confusion in data treatment and discussion. One example concerns the number and type of cracks (network cracks). As suggested by the authors in lines 766-767, 808-810, and 1120-1122, these features appear to be associated with the number of removal attempts and/or the shape of the CT active area, rather than the CT type. Consequently, the observed correlation between CT type and these features is merely apparent and should not be regarded as significant.

In section 4.1. The microscopic knapping traces of the article, we have chosen to describe in depth the microscopic knapping traces observed, as it seemed to us that such a description was still lacking in the literature of which we are aware. For this reason, we thought it would be useful to include in the analysis characteristics that may be influenced primarily by the body behaviour, such as the number of cracks observed on the platforms of the blades, as this can also be observed on archaeological material. However, we have not claimed or implied that there is necessarily a significant correlation between the number of cracks and the CT used. As Reviewer 1 rightly points out, we have been careful on the contrary to point out on various occasions that this element must depend in part on the number of percussions/pressures carried out by the knapper. It is also for this reason that the number and distribution of cracks on the platforms were excluded from the cluster analyses, as indicated in Supplementary 5.

However, as we did not record the number of percussions/pressures ultimately leading to a blade detachment, we are unable to assess precisely how this aspect may have influenced a feature such as network cracks (or others, see Comment 4 below). Based on the data at our disposal, we can only assess how this feature is, or is not, correlated with the MFA and the CT used, bearing in mind that the body behaviour certainly affects it too to some extent. We have added a comment on this aspect in the section Discussion to underline that aspects related to the body behaviour, such as the number of percussions/pressure attempts, will need to be looked at more closely in the future.

Comment 3

I see similar problems with post-fracture traces, such as spontaneous scars on the edges. The development of these traces seems linked to how the core was held or supported, not to the knapping technique used, as also noted by the authors on Page 49. In this case, there is a risk of emphasising a connection between scars and CTs when what we are observing is the knapper adjusting his body behaviour according to different shapes and sizes of cores and punches, as is also visible in Supplementary Information 7. Why, indeed, should the knapper adopt a different position when using an antler or boxwood CT?

We have not claimed that there is a causal link between the formation of spontaneous scars and the CT used. On the contrary, we stated that ‘The way in which the core and the blade are held also influences the position and lateralization of the scars, but this highly depends on the morphology of the core and how the knapper adapts to it’ (lines 1226-1228). Moreover, we took care to detail in section 4.1.5.1. Spontaneous scars on the edges how the way the core is held influences the formation of these scars.

Therefore, we agree with Reviewer 1 that spontaneous scars are mainly influenced by the way the core is held, that is to say by one of the aspects related to the body behaviour. However, in the case of our experimental collection, some characteristics of the spontaneous scars, such as their location on the dorsal and/or the ventral surface of the blades, is also related to the MFA, since the way cores were held when using percussion (direct or indirect) was significantly different than when using pressure.

Therefore, when we reproduced the cluster analyses, we chose to keep the attributes associated with the spontaneous scars. The situation here is somewhat like that of the lip, in the sense that several parameters are simultaneously influencing the same attribute. As we explained in lines 1658-1687, the formation of the lip is mainly linked to the tensile and compression components involved in the creation of the fracture, which depend directly on the knapping gesture, i.e. the body behaviour. However, this does not rule out the fact that another parameter of the technique, in this case the CT, also plays a key role, as its hardness encourages the knapper to use certain types of gesture, especially if the CT is softer that the knapped material.

Comment 4

A similar reasoning applies to lines 1021-1026 and lines 1046-1054: the orientation of linear polishes and the location of the residues, respectively.

We agree with Reviewer 1. With regard to the orientations of the linear polishes, we have in fact specified that they seem “to indicate that the knapper had a little more difficulty extracting blades with the boxwood punch” (lines 1025-1026), which clearly means that they relate to the body behaviour. Similarly, in lines 1046-1054, we have stated the location of the residues is influenced by the body behaviour (“the knapper placed the CT in different places until he found the ideal one”; “The frequent scattering of copper residues, and the less frequent scattering of organic residues can also be the result of successive attempts to detach a blade from the same area of the pressure platform”), and to lesser extent to the nature of the CT (“the copper CT leaves residues more easily than the organic CTs”).

As mentioned above, we did not record the number of percussions/pressures leading to a blade detachment, and so we are unable to assess how this aspect of the body behaviour may have influenced certain features (although the case of linear polishes with secant orientations is obvious, and this is why the orientation of the linear polishes was not included in the cluster analyses).

Comment 5

In all these cases, there is no causal relationship between the observed features and CT type. All of these features seem related to body behaviour or other factors that were not controlled for and were not considered in the experimental design, rather than CT or MFA. Nevertheless, they are included in the analysis. The authors should focus solely on attributes that can be unequivocally linked to CT and/or MFA.

Given the small number of descriptions of microscopic knapping traces published to date, as well as the virtual absence of any systematic description of these traces using a wide range of techniques, we felt it essential to present such a description. This is the subject of section 4.1. The microscopic knapping traces, the aim of which is to make these traces better known, particularly to lithic technology specialists. From our point of view, selecting only traces influenced mainly by the CT or the MFA would reduce the interest of both this section and the article.

We added a sentence in the Material and method part (see section 3.2.1. Visual analysis of the experimental collection) to briefly explain why we chose to propose a detailed description of the microscopic knapping traces observed on our experimental set of blades.

1.1.2. Experiment description

Some additional details on how the experiment was conducted would also be useful, such as the number of strikes for each blank removal, the position of the knapper, and a description of the hammers, punches, and crutches used to extract the blades (how many hammers of each type were involved? What were their sizes and shapes? etc.).

We have added these details in section 3.1.1. Protocol of the experiment (with the exception of the number of hammers used, which was already mentioned in lines 427-428). Most of the information relating to the tools has been brought together in a new table (Table 2).

1.1.3. Statistics

Comment 1

I am not an expert in multivariate statistics, but I think there are a few points where the authors should double-check the methods and/or provide a more extensive explanation of their choices. It is unclear to me why the attributes considered in the hierarchical cluster analysis are selected based on the analysis of single variables (ANOVA and Chi-square tests). Is it not circular to perform the cluster analysis only on significantly different attributes?

The statistical tests indicate that the MFA and the CT do not have the same influence on the characteristics of the knapping traces observed. Depending on the case, the link may be strong, weak or absent. This apparently simple result suggests that the identification of techniques needs to be rethought from a slightly different angle to that adopted to date: we may achieve more robust identifications if we try and identify separately each of the parameters making up a technique, based on the most relevant attributes. From our point of view, this is the main lesson of this study.

We understand the doubts raised by Reviewer 1 regarding our cluster analyses, but we do not believe that we have used circular reasoning. As indicated in lines 555-557, the objective of these analyses is to determine whether knapping techniques can be identified and, if so, to what extent. In other words, with what degree of precision. Selecting attributes that are statistically related to MFA or CT maximises the probability of obtaining a result, but it does not predict the exact nature of that result. For example, the selection of attributes linked to the MFA does not allow us to know whether the three MFAs used will be recognised in the same way. And in fact, the analyses showed that this was not the case, since direct percussion and indirect percussion proved more difficult to distinguish than percussion (direct + indirect) vs pressure.

We agree nonetheless with Reviewer 1

---

## [Editor Report · Decision Letter 1]

23 Jul 2025

When the hammer drops: Identification of knapping techniques in blade production based on a multi-scale study of knapping traces

PONE-D-25-08995R1

Dear Dr. Touzé,

We’re pleased to inform you that your manuscript has been judged scientifically suitable for publication and will be formally accepted for publication once it meets all outstanding technical requirements.

Kind regards,

Marco Peresani

Academic Editor

PLOS ONE
---

## [Editor Report · Acceptance letter]

PONE-D-25-08995R1

PLOS ONE

Dear Dr. Touzé,

I'm pleased to inform you that your manuscript has been deemed suitable for publication in PLOS ONE. Congratulations! Your manuscript is now being handed over to our production team.

Kind regards,

on behalf of

Dr. Marco Peresani

Academic Editor

PLOS ONE